# DART: Distribution Shift-Aware Prediction Refinement for Test-Time Adaptation

## ABSTRACT

Test-time adaptation (TTA) enables models to adapt to test domains using only unlabeled test data, addressing the challenge of distribution shift during test time. However, existing TTA methods mainly focus on input distribution shifts, often neglecting class distribution shifts. In this work, we first reveal that existing methods can suffer from performance degradation when encountering class distribution shifts. We also show that there exist class-wise confusion patterns observed across different input distribution shifts. Based on these observations, we introduce a novel TTA method, named *Distribution shift-Aware prediction Refinement for Test-time adaptation (DART)*, which refines the predictions made by the trained classifiers by focusing on class-wise confusion patterns. DART trains a distribution shift-aware module during intermediate time by exposing several batches with diverse class distributions using the training dataset. This module is then used during test time to detect and correct class distribution shifts, significantly improving pseudo-label accuracy for test data. This improvement leads to enhanced performance in existing TTA methods, making DART a valuable plug-in tool. Extensive experiments on CIFAR, PACS, ImageNet, and digit benchmarks demonstrate DART's ability to correct inaccurate predictions caused by test-time distribution shifts, resulting in significant performance gains for TTA methods.

## 1 INTRODUCTION

Deep learning has achieved remarkable success across various domains, including image classification (Krizhevsky et al., 2012; Simonyan & Zisserman, 2014; Radford et al., 2021) and natural language processing (Vaswani et al., 2017; Devlin et al., 2018). However, some recent findings have shown that when a substantial shift occurs between the training and test data distributions, the performance of trained models on test data often deteriorates considerably (Saenko et al., 2010; Taori et al., 2020; Mendonca et al., 2020). Test-time adaptation (TTA) methods have emerged as a prominent solution to mitigate the performance drop resulting from distribution shifts. TTA methods (Wang et al., 2020; Goyal et al., 2022; Boudiaf et al., 2022; Zhao et al., 2022; Jang et al., 2022) enable trained models to adapt to the test domain using only unlabeled test data, effectively addressing the challenge of distribution shift during test time.

In TTA methods, two primary branches exist: normalization-based and entropy minimization-based approaches. Normalization-based TTA techniques (Nado et al., 2020; Schneider et al., 2020) address the challenge by adjusting Batch Normalization (BN) (Ioffe & Szegedy, 2015) statistics using statistics obtained from the test domain. On the other hand, entropy minimization-based TTA methods (Lee, 2013; Liang et al., 2020; Wang et al., 2020; Goyal et al., 2022; Jang et al., 2022) adapt pre-trained models by leveraging predictions generated by the model itself on unlabeled test data, treating them as pseudo labels.

While these TTA techniques have proven effective against various test-time distribution shifts, including image corruptions, recent research (Gong et al., 2022; Zhou et al., 2023) has revealed that significant performance degradation can occur when the class distribution of the test domain differs from that of the training domain, in addition to the shift in input distribution. To assess the impact of class distribution shifts on existing TTA methods, we first benchmark the BNAdapt (Schneider et al., 2020) method on long-tailed test sets when the model is trained on balanced training sets. We observe substantial drops in test accuracy even after applying the BNAdapt, especially as the

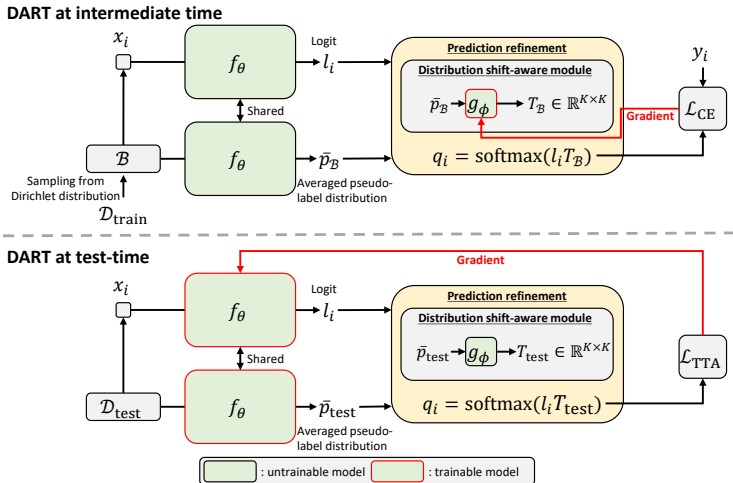

Figure 1: **Overview of DART**. *(Top)* At intermediate time, the period between the training and test times, DART trains a distribution shift-aware module $g_\phi$ to detect and correct the class distribution shifts. By sampling the training data from Dirichlet distributions, we generate batches $\mathcal{B}$ with diverse class distributions during the intermediate time. The distribution shift-aware module takes the averaged pseudo label distribution of $\mathcal{B}$ and outputs a square matrix $T_\mathcal{B}$ of size $K$ (class numbers) for prediction modification. Since the label of the training data is available, we optimize $g_\phi$ to minimize the cross-entropy loss of $\mathcal{B}$ while the pre-trained model $f_\theta$ is frozen. *(Bottom)* At test-time, we fine-tune the pre-trained $f_\theta$ using the refined predictions by $g_\phi$. We can compute the square matrix $T_{\text{test}}$ and modify the predictions of test data $x \in \mathcal{D}_{\text{test}}$ using $g_\phi$ since $g_\phi$ does not require any label for generating the square matrix. Thus, DART can be used in conjunction with existing TTA methods.

imbalance ratio between classes increases (Table 1). This shows the challenge in TTA when facing label distribution shift in addition to input distribution shift during test time. We further examine the misclassification (confusion) patterns between classes under various input distribution shifts, represented by eight distinct image corruption patterns. An interesting observation is that consistent class-wise confusion patterns occur across different input corruption patterns (Fig. 2).

Motivated by such observations, we propose a novel test-time adaptation method, named *Distribution shift-Aware prediction Refinement for Test-time adaptation (DART)* as a solution to address test-time distribution shifts in both input data and label distributions. DART aims to correctly modify predictions made by trained classifiers by focusing on class-wise confusion patterns that arise due to label-distribution shifts. Our key insight is that the model can learn how to adjust inaccurate predictions due to label distribution shifts by experiencing several batches with diverse class distributions using the labeled training dataset before the start of test time. DART trains a distribution shift-aware module during an *intermediate time*, situated between the end of training and the start of testing, by exposing multiple batches composed of labeled training data with diverse class distributions, sampled from the Dirichlet distribution. The module then outputs a square matrix of the class dimension that will be multiplied to logit vector of the network for prediction refinement. Since the distribution shift-aware module only requires a pseudo-label distribution as an input, it can be readily employed during test time by providing the estimated pseudo-label distribution for the test data generated by the pre-trained model, as depicted in Fig. 1.

We evaluate the effectiveness of DART on several standard test-time adaptation benchmarks, including CIFAR-10/100C, ImageNet-C, CIFAR-10.1, PACS, and digit classification (SVHN →MNIST/USPS/MNIST-M). Our results consistently demonstrate that DART enhances prediction accuracy across most benchmarks involving test-time distribution shifts in both input data and label distributions and it contributes to the improved performance of existing TTA methods. Specifically, DART achieves substantial improvements in test accuracy, enhancing the BNAdapt method by 5.52% and 16.44% on CIFAR-10C-LT under class imbalance ratios of $\rho = 10$ and 100, respectively. Furthermore, our extensive ablation studies demonstrate the pivotal role played by the prediction refinement scheme in DART's performance enhancement.

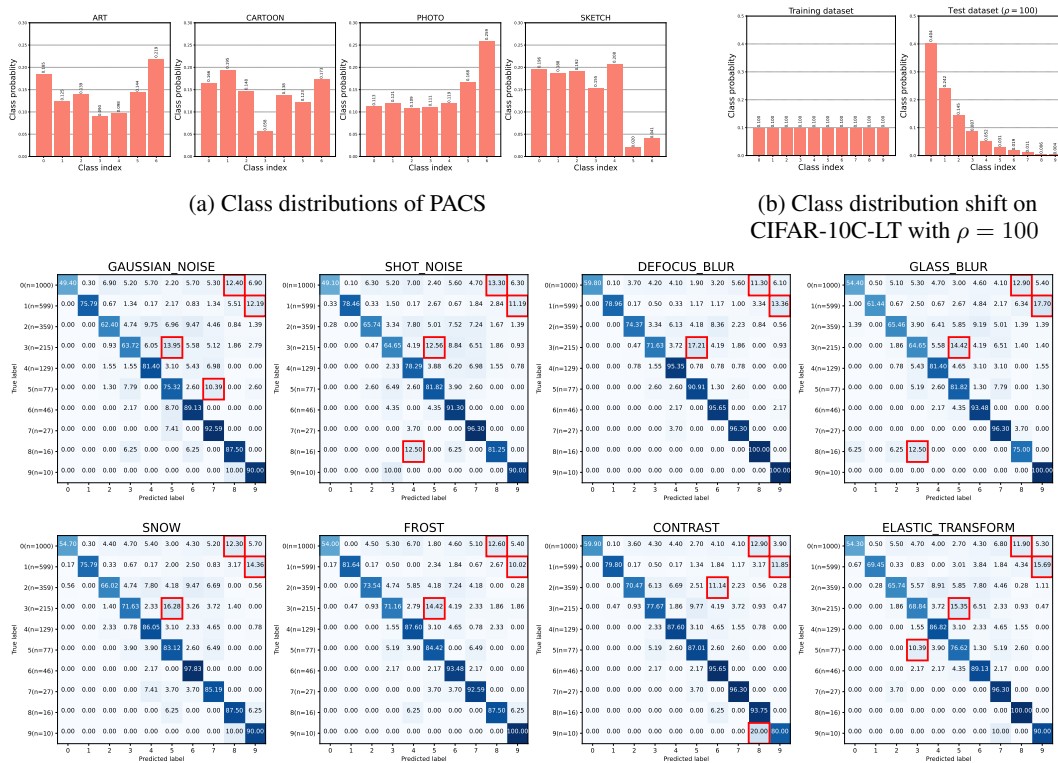

(a) Class distributions of PACS

(b) Class distribution shift on CIFAR-10C-LT with $\rho = 100$

(c) Confusion matrices from BNAdapt model for 8 different corruption types from CIFAR-10C-LT ($\rho = 100$)

Figure 2: **Test-time class distribution shifts**. (a) The class distribution of four different domains in PACS. (b) *(left)* The class distribution of training and *(right)* test distributions of CIFAR-10C-LT with the class imbalance ratio $\rho$ of 100. We evaluate the robustness of classifiers trained on CIFAR-10, which has a balanced class distribution, on CIFAR-10C-LT, which has a long-tailed class distribution. (c) Confusion matrices of BNAdapt on CIFAR-10C-LT for eight different types of corruptions. We mark the cases where the confusion rate exceeds 10% with red squares. We can observe notable accuracy degradation in classes with large amounts of data (*e.g.,* class 0 and 1), and similar confusing patterns regardless of the corruption types under class distribution shifts.

## 2 PROBLEM SETUP AND MOTIVATION

We consider a $K$-class classification problem under test-time distribution shift. During training time, a classifier $f_\theta$ is trained using a labeled training dataset $\mathcal{D} = \{(x_i, y_i)\}_{i=1}^{n^{\text{train}}}$, drawn from a training distribution $P_{XY}$ over $\mathcal{X} \times \{0, \ldots, K-1\}$. However, during test time, the classifier may encounter test data $\mathcal{D}_{\text{test}} = \{(x_i', y_i')\}_{i=1}^{n^{\text{test}}}$ drawn from a test distribution $P_{XY}^{\text{test}} \neq P_{XY}^{\text{train}}$. This shift in distribution significantly degrades the classification performance of the trained classifier on the test data (Wang et al., 2020). To address this issue, many test-time adaptation (TTA) methods aim to adapt the trained model to the test domain using only unlabeled test data. While many existing TTA methods have predominantly focused on covariate shifts, where the input data distributions change between the training and test data (e.g., due to image style transfer or image corruption), we focus on a problem setup where both covariate and label distribution shifts occur during the test time.

### 2.1 MOTIVATION FOR PREDICTION REFINEMENT SCHEME

**Impact of label distribution shifts on existing TTA methods** We first examine the impact of label distribution shifts on BNAdapt on CIFAR-10C. We evaluate the performances of the trained classifier on the CIFAR-10C-LT test set, when this classifier was initially trained with CIFAR-10 (Krizhevsky & Hinton, 2009). CIFAR-10C (Hendrycks & Dietterich, 2019) is a benchmark designed to evaluate the robustness of models trained on clean CIFAR-10 data against 15 predefined types of

corruptions, including Gaussian noise. To create a class-imbalanced dataset, CIFAR-10C-LT, which exhibits a long-tailed class distribution, as depicted in Fig. 2b, we set the number of images per class to decrease exponentially as the class index increases. Specifically, we set the number of samples for class $k$ as $n_k = n(1/\rho)^{k/(K-1)}$, where $\rho$ denotes the class imbalance ratio.

In Table 1, we compare the performances of NoAdapt, which makes no modifications to the trained classifier, and BNAdapt, which updates the Batch Normalization (BN) statistics with those of the test domain on CIFAR-10C-LT for different class imbalance ratios $\rho$ set to 1, 10, and 100.

We observe a decline in performance for BNAdapt as the class imbalance ratio increases, while the performance of NoAdapt remains consistent regardless of class imbalance. When $\rho = 100$, BNAdapt exhibits even worse performance than NoAdapt. This shows that in the presence of class distribution shifts, correcting BN statistics without accounting for the class distribution shift significantly degrades TTA performance.

Table 1: Average accuracy (%) on CIFAR-10C-LT with several class imbalance ratios $\rho$.

| Method | CIFAR-10C-LT | | |
| --- | --- | --- | --- |
| | $\rho = 1$ | $\rho = 10$ | $\rho = 100$ |
| NoAdapt | 71.68±0.00 | 71.28±0.08 | 71.13±0.17 |
| BNAdapt | 85.24±0.08 | 79.01±0.07 | 66.90±0.16 |
| Oracle | 85.53±0.05 | 85.97±0.18 | 87.77±0.07 |

In Figure 2c, we present confusion matrices between classes, where each entry $(i, j)$ represents the fraction of samples from the $i$-th class classified into the $j$-th class. These matrices are generated across eight different types of image corruption patterns for CIFAR-10C-LT, where $\rho = 100$. We observe significant accuracy degradation in head classes (with smaller class index $k$), for which the fraction of samples increase the most during test time. Additionally, we notice that the confusion patterns tend to be consistent across different corruption types when the label distribution shift is fixed. For instance, frequent class-wise confusion patterns include 0 to 8 (airplane → ship), 1 to 9 (automobile → truck), and 3 to 5 (cat → dog). In Appendix C, we provide a theoretical analysis demonstrating that such a class-wise confusion pattern occurs when facing label distribution shifts by using a toy example of four-class Gaussian mixture distribution. This observation raises the question of how to effectively learn and utilize such a confusion pattern between classes to correctly modify the predictions of the classifier *during test time* where only unlabeled test data is available.

**An oracle's attempt for prediction refinement**   To answer this challenging question, we begin with a simpler but relevant question: Can we prevent the performance degradation of BNAdapt by refining model predictions through the multiplication of a *distribution shift-aware* square matrix $T \in \mathbb{R}^{K \times K}$ with the model outputs? This type of refinement scheme is commonly employed to adjust model outputs trained with label-noise datasets when specific class-wise confusion patterns, as seen in Figure 2c, exist (Natarajan et al., 2013; Patrini et al., 2017; Zhu et al., 2021). However, it has not been explored in the context of test-time adaptation, where performance degradation is caused by distribution shifts rather than label noise. To evaluate the effectiveness of multiplying a square matrix with the model's output, we first consider an Oracle method using the *labeled* test data to find a desirable $T_{\text{oracle}} \in \mathbb{R}^{K \times K}$. We define $T_{\text{oracle}}$ as a solution that minimizes the cross-entropy loss between the modified softmax probability and the ground truth labels of the test data: $T_{\text{oracle}} = \arg\min_{T \in \mathbb{R}^{K \times K}} \mathbb{E}_{(x,y) \in \mathcal{D}_{\text{test}}}[\text{CE}(\text{softmax}(f_\theta(x)T), y)]$, and we find $T_{\text{oracle}}$ by gradient descent. The last row of Table 1 presents the test accuracy achievable with $T_{\text{oracle}}$ when applied to the output of the BNAdapt model. Remarkably, simply multiplying the output with $T_{\text{oracle}}$ reverses the performance degradation caused by class distribution shifts. Having confirmed the effectiveness of refining the output by multiplying with $T_{\text{oracle}}$, the remaining question is how to obtain such a square matrix $T$ without access to test data labels.

## 3   DISTRIBUTION SHIFT-AWARE PREDICTION REFINEMENT FOR TTA

We introduce a distribution shift-aware module that can detect test-time class distribution shifts and output a square matrix to refine the predictions of the trained classifiers. Our core idea is that if the module experiences various batches with diverse class distributions before the test time, it can develop the ability to refine inaccurate predictions resulting from label distribution shifts. Based on this insight, we train a distribution shift-aware module $g_\phi$ during the intermediate time, in between the training and testing times, by exposing several batches with diverse class distributions using the

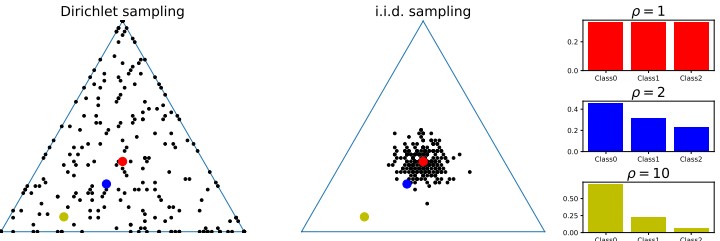

Figure 3: **Example of the Dirichlet distribution sampling**. IID (i.i.d.) sampling denotes standard uniform sampling. The black dots indicate the class distribution of the sampled batches. The red, blue, yellow dots represent the class distributions of different class imbalance ratios $\rho$, namely 1,2, and 10, respectively. By employing the Dirichlet distribution for batch sampling, we can expose the model to numerous batches with diverse class distributions during the intermediate time, thereby enabling it to learn how to mitigate performance degradation caused by class distribution shifts.

training datasets. During the test time, $g_\phi$ takes an averaged pseudo-label distribution for test data as input to detect class distribution shifts, and generates a square matrix of size $K$ to refine predictions.

**Dataset for intermediate time**   During the intermediate time, we assume that the labeled training dataset $\mathcal{D}$ is available while the test dataset $\mathcal{D}_{\text{test}}$ remains unavailable, as is common in previous settings (Choi et al., 2022; Lim et al., 2022; Park et al., 2023). For example, we use CIFAR-10 dataset during the intermediate time on CIFAR-10C-LT benchmark. In cases where the training dataset exhibits imbalanced class distribution, as seen in datasets like SVHN or PACS, the imbalanced class distribution can inadvertently influence the training of $g_\phi$. To mitigate this, we create a class-balanced intermediate dataset $\mathcal{D}_{\text{int}}$ by uniformly sampling data from each class.

**Training of $g_\phi$**   To create batches with diverse class distributions during the intermediate time, we employ a Dirichlet distribution (Yurochkin et al., 2019; Gong et al., 2022). Batches sampled through i.i.d. sampling tend to have class distributions, resembling a uniform distribution. In contrast, batches sampled using the Dirichlet distribution exhibit a wide range of class distributions, including long-tailed distributions as illustrated in Figure 3. The training objective of $g_\phi$ for a batch $\mathcal{B} \subset \mathcal{D}_{\text{int}}$ is formulated as follows:

$$\mathcal{L}(\phi) = \mathbb{E}_{(x,y)\in\mathcal{B}}[\text{CE}(\text{softmax}(f_\theta(x)g_\phi(\bar{p})), y] \quad \text{where} \quad \bar{p} = \frac{1}{|\mathcal{B}|}\sum_{(x,y)\in\mathcal{B}}\text{softmax}(f_\theta(x)). \quad (1)$$

Here, $\bar{p}$ represents the averaged pseudo-label distribution for batch $\mathcal{B}$, and CE denotes the standard cross-entropy loss. During the intermediate time, $g_\phi$ is optimized to minimize the cross-entropy loss between the modified softmax probability and the ground truth labels of the training samples. During this time, the parameters of the trained classifier $f_\theta$ are not updated, but the batch statistics in the classifiers are updated as in BNAdapt (Schneider et al., 2020). For each pre-trained model, we train $g_\phi$ only once, regardless of the number of test domains.

**Utilizing $g_\phi$ at test-time**   Since the distribution shift-aware module $g_\phi$ only requires the averaged pseudo label distributions generated by the trained classifier as input, it can be employed effectively at test time when only unlabeled test data is available. During test time, $g_\phi$ takes the pseudo label distribution $\hat{p}$ averaged over the test dataset $\mathcal{D}_{\text{test}}$, and generates a square matrix $T_{\text{test}} \in \mathbb{R}^{K \times K}$,

$$T_{\text{test}} = g_\phi(\hat{p}) \quad \text{where} \quad \hat{p} = \frac{1}{|\mathcal{D}_{\text{test}}|}\sum_{\hat{x}\in\mathcal{D}_{\text{test}}}\text{softmax}(f_\theta(\hat{x})). \quad (2)$$

We note that $g_\phi$ remains frozen and does not update its parameters during test time. We obtain the square matrix $T_{\text{test}}$ at the start of the test phase and utilize it throughout the test time. By multiplying the classifier output with $T_{\text{test}}$, we can effectively enhance the accuracy of pseudo-labels. Furthermore, as illustrated in Figure 1 (bottom), our method can be integrated as a plug-in with any test-time adaptation (TTA) methods that rely on pseudo labels obtained from the classifier. Specifically, we can adapt the pre-trained classifier $f_\theta$ by using the modified output $\text{softmax}(f_\theta(\cdot)T_{\text{test}})$. For example, in the case of TENT (Wang et al., 2020), we adapt the classifier $f_\theta$ using a training objective $\mathcal{L}_{\text{TENT}}(\theta) = \mathbb{E}_{\hat{x}\in\mathcal{D}_{\text{test}}}[-\sum_k \text{softmax}(f_\theta(\hat{x})T_{\text{test}})_k \log \text{softmax}(f_\theta(\hat{x})T_{\text{test}})_k]$.

Table 2: Average accuracy (%) on CIFAR-10C/10.1-LT, digit classification, and PACS. **Bold** indicates the best performance for each benchmark.

| Method | CIFAR-10C-LT | | CIFAR-10.1-LT | | Digit | PACS |
|--------|--------------|--------------|---------------|---------------|-------|------|
| | $\rho = 10$ | $\rho = 100$ | $\rho = 10$ | $\rho = 100$ | | |
| NoAdapt | 71.28±0.08 | 71.13±0.17 | 87.13±0.48 | 86.64±0.97 | 58.45±0.00 | 60.65±0.00 |
| BNAdapt | 79.01±0.07 | 66.90±0.16 | 77.37±0.45 | 64.43±0.97 | 61.10±0.20 | 72.08±0.11 |
| BNAdapt+ours | 84.53±0.20 | 83.34±0.20 | 85.81±0.65 | 80.64±2.12 | 62.60±0.35 | 75.33±0.09 |
| TENT | 83.02±0.19 | 70.49±0.43 | 78.23±0.52 | 64.53±1.53 | 63.59±0.19 | 74.53±0.97 |
| TENT+ours | **85.13±0.31** | 88.56±0.13 | 86.88±0.78 | 82.32±1.60 | 64.85±0.44 | 80.98±1.19 |
| PL | 83.09±0.28 | 69.63±0.46 | 78.51±0.38 | 64.38±1.09 | 63.25±0.23 | 70.56±0.75 |
| PL+ours | 84.50±0.39 | 87.88±0.07 | 86.02±0.93 | 82.16±1.44 | 64.60±0.33 | 80.12±0.49 |
| DELTA | 82.41±0.59 | 69.88±1.47 | 78.57±2.42 | 64.79±1.61 | 64.64±0.23 | 77.60±0.87 |
| DELTA+ours | 84.46±0.30 | **89.25±0.33** | **87.19±0.35** | 84.25±1.54 | **65.83±0.34** | **83.46±1.33** |
| NOTE | 80.72±0.23 | 79.49±0.41 | 82.94±1.95 | 81.55±1.59 | 63.85±0.41 | 67.84±0.56 |
| NOTE+ours | 80.79±0.14 | 85.38±0.29 | 84.67±0.83 | **87.25±1.01** | 64.15±0.29 | 68.76±0.90 |
| LAME | 80.50±0.06 | 70.40±0.25 | 78.79±1.00 | 67.68±2.58 | 63.97±0.14 | 65.43±0.27 |
| LAME+ours | 82.27±0.20 | 79.66±0.12 | 82.60±0.76 | 77.34±1.31 | 64.54±0.56 | 76.50±0.11 |
| ODS | 82.01±0.28 | 78.32±0.34 | 81.83±1.77 | 77.74±1.86 | 65.50±0.26 | 64.25±0.13 |
| ODS+ours | 83.14±0.15 | 81.58±0.17 | 82.85±1.12 | 79.32±1.48 | 65.81±0.23 | 65.66±0.75 |

## 4 EXPERIMENTS

### 4.1 EXPERIMENTAL SETUP

**Benchmarks** We consider two types of input data distribution shifts: synthetic and natural distribution shifts, each characterized by its generation process. Synthetic distribution shifts are artificially created through data augmentation techniques including image corruption using Gaussian noise. In contrast, natural distribution shifts arise from changes in image style, for instance, from artistic to photographic styles. We evaluate synthetic distribution shifts on CIFAR-10/100C and ImageNet-C (Hendrycks & Dietterich, 2019), and natural distribution shifts on CIFAR-10.1 (Recht et al., 2018), digit classification (SVHN → MNIST/USPS/MNIST-M) and PACS (Li et al., 2017) benchmarks. For synthetic distribution shifts, we apply 15 different types of common corruptions, each at the highest severity level (*i.e.* level 5). To evaluate the impact of class distribution shifts in CIFAR benchmarks, we introduce test datasets with long-tailed class distributions, as described in Section 2.1. For ImageNet-C, we create a new test set with online label distribution shifts following the approach in Niu et al. (2023). This new test set comprises $K$ subsets, each characterized by a class distribution $[p_1, p_2, \ldots, p_K]$, where $p_k = p_{max}$ and $p_i = p_{min} = (1 - p_{max})/(K - 1)$ for $i \neq k$, where $K$ is the number of classes in ImageNet-C, which is 1,000. Let $\alpha = p_{max}/p_{min}$ represent the imbalance ratio. Each subset consists of ImageNet-C test images sampled according to the aforementioned class distribution. Additionally, we shuffle the subsets to prevent predictions based on their order. Conversely, for digit classification and PACS benchmarks, we utilize the original datasets, as these benchmarks inherently include label distribution shifts across domains. More details for benchmarks are available in Appendix A.

**Baselines** We compare DART with the following baselines: (1) BNAdapt (Schneider et al., 2020) corrects the batch statistics using the test data; (2) TENT (Wang et al., 2020) fine-tunes parameters in BN layer of the trained classifier to minimize the prediction entropy of test data; (3) PL (Lee, 2013) fine-tunes the trained classifier using confident pseudo-labeled test samples; (4) NOTE (Gong et al., 2022) adapts the classifiers while mitigating the effects of non-i.i.d test data streams through instance-aware BN and prediction-balanced reservoir sampling; (5) DELTA (Zhao et al., 2022) adapts the classifiers while addressing issues related to incorrect BN statistics and prediction bias by employing test-time batch renormalization and dynamic online reweighting; (6) ODS (Zhou et al., 2023) estimates the label distribution of test data through Laplacian-regularized maximum likelihood estimation and adapts the trained model by assigning high and low weights to infrequent and frequent classes, respectively; (7) LAME (Boudiaf et al., 2022) modifies the prediction

Table 3: Average accuracy (%) on CIFAR-100C and ImageNet-C under several class imbalance ratios $\rho$ and $\alpha$. As $\rho$ and $\alpha$ increase, the severity of the class distribution shifts is intensified.

| | CIFAR-100C-LT | | ImageNet-C online imbalanced | | |
| | $\rho = 10$ | $\rho = 100$ | $\alpha = 1000$ | $\alpha = 2000$ | $\alpha = 5000$ |
|---|---|---|---|---|---|
| NoAdapt | 41.04±0.17 | 40.71±0.24 | 18.15±0.06 | 18.16±0.01 | 18.16±0.04 |
| BNAdapt | 58.33±0.15 | 55.25±0.13 | 19.85±0.10 | 14.11±0.04 | 8.48±0.06 |
| BNAdapt+ours | 59.79±0.18 | 59.74±0.16 | 25.18±0.75 | 20.48±0.80 | 14.82±0.78 |
| TENT | 61.32±0.20 | 58.21±0.40 | 22.49±0.15 | 13.52±0.11 | 6.61±0.08 |
| TENT+ours | 62.54±0.28 | 63.79±0.22 | 26.18±0.88 | 18.51±0.98 | 11.17±0.77 |

by Laplacian-regularized maximum likelihood estimation considering nearest neighbor information in the embedding space of classifiers. More details about baselines are available in Appendix A.

**Experimental details**  We use ResNet-18/26 (He et al., 2016) as the backbone networks for the digit and CIFAR, and ResNet-50 for PACS and ImageNet-C benchmarks, respectively. During the intermediate time, we use a 2-layer MLP (Haykin, 1998) with a hidden dimension of 1,000 for the distribution shift aware module $g_\phi$. We train $g_\phi$ using the labeled dataset from the training domain with an intermediate batch size of 50/200 for 100 epochs for ImageNet-C and other benchmarks. We ensure that fine-tuning layers, optimizers, and hyperparameters remain consistent with those introduced in each baseline for a fair comparison. Implementation details for pre-training, intermediate-time training, and test-time adaptation are described in Appendix A.

## 4.2 EXPERIMENTAL RESULTS

**DART-applied TTA methods**  In Table 2, we present and compare the experimental results for the original vs. DART-applied TTA methods across CIFAR-10C-LT, CIFAR-10.1-LT, digit classification, and PACS benchmarks. We can observe that DART consistently improves the performance of existing TTA methods for both synthetic and natural distribution shifts. In particular, the performance gain achieved by DART becomes more significant as the class imbalance ratio $\rho$ increases. For instance, on CIFAR-10C-LT with class imbalance ratios $\rho = 10$ and 100, DART boosts the test accuracy of BNAdapt by 5.52% and 16.44%, respectively. We can see that there is no single dominant TTA method that outperforms all the other baselines across all benchmarks, as previously observed in Zhao et al. (2023). In experiments on CIFAR-10.1, many TTA methods that rely on BNAdapt do not improve the performance of the pre-trained model, as demonstrated in Zhao et al. (2023). However, DART efficiently mitigates the accuracy degradation caused by test-time distribution shifts in this scenario and outperforms NoAdapt when combined with either DELTA for $\rho = 10$ or NOTE for $\rho = 100$. NOTE and ODS, which construct prediction-balanced batches using memory for adaptation, assume that the pre-trained model has been trained on a balanced training dataset. Consequently, these methods exhibit poor performance on PACS benchmarks, where training class distributions are significantly imbalanced. Detailed experimental results and additional findings, including DART's performance on balanced test dataset, are reported in Appendix F.

**Comparison between $T_{\text{oracle}}$ and $T_{\text{test}}$**  To verify the effectiveness of $g_\phi$, we compare the output $T_{\text{test}}$ of the distribution-shift aware module and $T_{\text{oracle}}$ obtained using the labeled test data as in Section 2.1 for the case of CIFAR-10C-LT with Gaussian noise ($\rho = 100$) in Figure 4. We can observe that the distribution shift module can indeed provide a good estimate $T_{\text{test}}$ that closely resembles $T_{\text{oracle}}$ even without access to the ground truth labels of the test data.

**DART on large-scale datasets**  We proceed to evaluate the effectiveness of DART on large-scale datasets. As the number of classes increases, the output dimension of $g_\phi$ also increases, becoming more challenging to learn and generate a higher-dimensional square matrix $T$ for large-scale datasets. To address this challenge, we modify the distribution shift-aware module to produce $T$ with some entries fixed to 0 for the large-scale datasets. For CIFAR-100C, we first analyze class-wise confusion patterns, similar to Figure 2c, using an augmented training dataset. Then we set the entries where class-wise confusion never occurred to 0 when training the distribution-shift aware module to generate $T$. For example, when using a speckle-noised augmentation for the CIFAR-

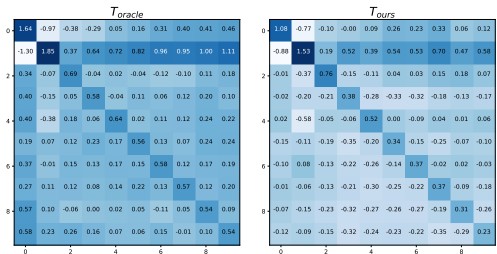

Figure 4: Comparison of $T_{\text{oracle}}$ and $T_{\text{test}}$ on CIFAR-10C-LT ($\rho = 100$) with Gaussian noise.

| Method | CIFAR-10C-LT | |
|---|---|---|
| | $\rho = 10$ | $\rho = 100$ |
| BNAdapt | 79.01±0.07 | 66.90±0.16 |
| BNAdapt+ours | 84.53±0.20 | 83.34±0.20 |
| BNAdapt+ours (diag) | 83.94±0.15 | 76.41±0.21 |
| BNAdapt+ours (online) | 83.54±0.76 | 82.57±0.49 |

Table 4: Ablation studies to evaluate the effectiveness of two variants of DART: *(online)* We obtain $T$ using only the first test batch. *(diag)* We set all off-diagonal elements of $T$ to 0.

100 dataset, we can set 7,400 entries of $T \in \mathbb{R}^{100 \times 100}$ to 0. This noise type is not used when testing the model with CIFAR-100C test sets. On the other hand, for ImageNet-C, given the large number of classes (1,000), we set all off-diagonal entries to 0. In Table 3, we summarize the experimental results comparing the original TTA methods with DART-applied methods on CIFAR-100C and ImageNet-C. DART consistently improves the test accuracy of BNAdapt, achieving a 1.46% improvement for $\rho = 10$ and a 4.49% improvement for $\rho = 100$ on CIFAR-100C, respectively. Moreover, DART achieves a performance gain of about 6% for all imbalance ratios on ImageNet-C.

## 4.3 ABLATION STUDIES

**DART for online TTA**   Some TTA works (Wang et al., 2020; Iwasawa & Matsuo, 2021; Jang et al., 2022) focus on an online approach where each test data sample is encountered only once during test time. To adapt DART for this online TTA scenario, we modify it to take the averaged pseudo label distribution of the first test batch to output $T$, which is then used throughout the test time. This differs from the original DART, which takes the averaged pseudo label distribution of the entire test dataset. We summarize the experimental results of this variant of DART for online TTA in Table 4 (last row). The results indicate that this online variant of DART performs similarly to the original DART but with a slight decrease in performance.

**Effects of diagonal/off-diagonal entires of $T$**   To assess the importance of both the diagonal and off-diagonal entries of the square matrix $T$, we consider a variant of DART in which all off-diagonal entries are set to 0. The experimental results presented in Table 4 on CIFAR-10C-LT and Table 16 on CIFAR-10.1-LT in Appendix show that this variant achieves performance improvements of 4.93/9.51% on CIFAR-10C-LT with $\rho$ values of 10/100 and 6.34/9.81% on CIFAR-10.1-LT with $\rho$ values of 10/100, respectively. However, this variant exhibits accuracy decreases of 6.94/6.45% on CIFAR-10C/10.1-LT with $\rho = 100$ compared to the original DART, respectively. These results suggest that both the diagonal and off-diagonal entries in the matrix $T$ play important roles in improving TTA performance, and removing the off-diagonal entries can lead to decreased performance in certain scenarios. More experimental results using these two variants can be found in Appendix F.

**Effects of Dirichlet sampling and prediction modification scheme of DART**   We consider three variants of DART, named DART v1-3, which involve changes to either the sampling strategy or the prediction modification scheme. For sampling strategy, we consider a scenario where the module experiences only three types of batches during the intermediate time: uniform, long-tailed with a class imbalance ratio $\rho = 20$, and inversely long-tailed class distributions. For prediction modification scheme, we modify $g_\phi$ to generate the parameters for a part of the model, including affine parameters for the output of the feature extractor and the weight difference for the classifier weights, inspired by the label shift adapter (LSA) (Park et al., 2023). For this case, the output dimension of $g_\phi$ gets larger since it is proportional to the feature dimension of the trained model. In Table 5, we report the performance of BNAdapt combined with these DART variants on CIFAR-10C-LT. DART v1 shows worse performances on both $\rho = 10$ and 100 compared to DART. This suggests the challenge in training $g_\phi$ to generate a high-dimensional output for prediction modification and demonstrates the benefit of refining the prediction output by simply multiplying the square matrix. DART v2 shows worse performance compared to DART for the $\rho = 100$ case, which is more severely imbalanced than the class distribution experienced during the intermediate time ($\rho = 20$). This observation shows the benefit of Dirichlet sampling. Lastly, DART v3 exhibits worse performance than DART

Table 5: Ablation studies to evaluate the effects of the Dirichlet sampling and prediction modification scheme of DART. We consider three variants of DART, which replace each component with the ones used in LSA. We report the performance of BNAdapt combined with DART variants on CIFAR-10C-LT.

| Method | Sampling strategy for int. time | | $g_\phi$ output | | Test acc. (%) | |
|---|---|---|---|---|---|---|
| | Dirichlet | Unif&LT ($\rho = 20$) | Square matrix | Parameters of a model | $\rho = 10$ | $\rho = 100$ |
| DART | ✓ | | ✓ | | 84.53±0.20 | 83.34±0.20 |
| DART v1 | ✓ | | | ✓ | 84.00±0.18 | 79.18±0.11 |
| DART v2 | | ✓ | ✓ | | 84.74±0.06 | 81.72±0.18 |
| DART v3 | | ✓ | | ✓ | 85.18±0.30 | 82.29±0.41 |

for $\rho = 100$ similar to DART v2. These experiments demonstrate that both the prediction refinement scheme and the sampling strategy contribute to the effectiveness and scalability of DART.

Due to space limitation, we present other ablation studies in Appendix E. Throughout these additional experiments, we confirm that (1) obtaining a square matrix $T$ by using only confident pseudo-labeled test samples during test time results in worse performance compared to DART, and (2) using the fixed $T$ generated by DART during test time is more effective than attempting to update $T$ through iterative or gradient-based methods.

## 5 RELATED WORKS

**TTA method utilizing intermediate time** Some recent works (Choi et al., 2022; Lim et al., 2022) have explored methods to prepare unknown test-time distribution shifts by leveraging the training dataset at the intermediate time. For instance, LSA (Park et al., 2023) involves exposing the model to several batches with three types of class distributions during the intermediate time: the training class distribution, a uniform distribution, and the inversely imbalanced training distribution. LSA primarily focuses on adjusting model parameters. Specifically, it trains a label shift adapter to produce affine parameters for the output of the feature extractor and weight difference for the classifier weights. In contrast, DART exposes the distribution shift-aware module to a more diverse range of class distributions during the intermediate time through Dirichlet sampling. DART's main objective is to correct predictions with a specific focus on class-wise confusion patterns. It uses a square matrix to modify predictions directly, without necessarily adjusting model parameters. In Section 4.3, the effectiveness and scalability of DART compared to LSA are demonstrated.

**TTA methods considering class-wise relationships** Some TTA methods (Iwasawa & Matsuo, 2021; Kang et al., 2023; Zhang et al., 2023) consider the class-wise relationship as domain-invariant information and aim to preserve it during test time. The method in (Kang et al., 2023) stores the class-wise relationship of the training domain and tries to minimize the difference between the class-wise relationships of the training and test domains. CRS (Zhang et al., 2023) estimates the class-wise relationships using the last linear layer of the trained models and embeds the source-domain class relationship in contrastive learning. While these methods utilize class-wise relationships to prevent their deterioration during test-time adaptation, DART takes a different approach by focusing on directly modifying the predictions. DART considers class-wise confusion patterns to refine predictions, effectively addressing performance degradation due to distribution shifts, without explicitly enforcing preservation of class-wise relationships. More related works are reviewed in Appendix B.

## 6 CONCLUSION

We proposed DART, a method designed to mitigate the impact of test-time class distribution shifts including both covariate and label distribution shifts, by taking class-wise confusion patterns into account. DART achieves this by training a distribution-shift aware module during the intermediate time to refine the predictions of pre-trained classifiers. Our experimental results demonstrate the effectiveness of DART across benchmarks that include both synthetic and natural distribution shifts. We expect that our method can be integrated with various TTA techniques in future applications, enhancing the robustness and accuracy of models when facing test-time distribution shifts.

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

# A    IMPLEMENTATION DETAILS

## A.1    DETAILS ABOUT DATASET

We consider two types of input data distribution shifts: synthetic and natural distribution shifts. The synthetic and natural distribution shifts differ in their generation process. Synthetic distribution shift is artificially generated by data augmentation schemes including image corruption like Gaussian noise and glass blur. On the other hand, the natural distribution shift occurs due to changes in image style transfer, for example, the domain is shifted from artistic to photographic styles.

For the synthetic distribution shift, we first test on CIFAR-10/100C, which is created by applying 15 types of common image corruptions (e.g. Gaussian noise and impulse noise) to the clean CIFAR-10/100 test dataset. We test on the highest severity (*i.e.*, level-5). CIFAR-10/100C is composed of 10,000 generic images of size 32 by 32 from 10/100 classes, respectively. The class distributions of the original CIFAR-10/100C are balanced. Thus, to change the label distributions between training and test domains, we consider CIFAR-10/100C-LT, which have long-tailed class distributions, as described in Section 2.1. Then, we test on ImageNet-C, which is composed of generic images of size 224 by 224 from 1,000 classes. The samples of ImageNet-C are created by applying the same image corruptions of CIFAR-10/100C. We test on the highest severity (*i.e.*, level-5). To change the label distributions between training and test domains, we can construct a new test set, named ImageNet-C-LT similar to CIFAR-10/100C-LT. However, unlike CIFAR-10/100C-LT, each test batch of ImageNet-C-LT does not have imbalanced class distributions, since the test batch size for the ImageNet-C is set to be smaller than the number of classes, *e.g.,* 32 or 64. Thus we consider a new test set for ImageNet-C by the online-label distribution shift setup, described in SAR, which is composed of $K$ subsets, whose $K$ is the number of classes of ImageNet-C. We assume a class distribution of the $k$-th subset as $[p_1, p_2, \ldots, p_K]$, where $p_k = p_{\max}$ and $p_i = p_{\min} = (1 - p_{\max})/(K - 1)$ for $i \neq k$. Let $\alpha = p_{\max}/p_{\min}$ represent the imbalance ratio. Each subset consists of 1,000 samples from the ImageNet-C test set based on the above class distribution. Thus, the new test set for ImageNet-C is composed of 100,000 samples. Additionally, we shuffle the subsets to prevent predictions based on their order.

For the natural distribution shift, we test on CIFAR-10.1-LT, digit classification, and PACS benchmarks. CIFAR-10.1 (Recht et al., 2018) is a newly collected test dataset for CIFAR-10 from the Tiny-Images dataset (Torralba et al., 2008), and is known to exhibit a distribution shift from CIFAR-10 due to differences in data collection process and timing. Since the CIFAR-10.1 has a balanced class distribution, we construct a test set having a long-tailed class distribution, named CIFAR-10.1-LT, similar to CIFAR-10/100C-LT. The digit classification benchmark consists of one training dataset (SVHN (Netzer et al., 2011)) and three test datasets MNIST (Deng, 2012), USPS (Hull, 1994), and MNIST-M (Ganin et al., 2016)). These four datasets have different styles of digit images. SVHN is composed of 73,257/26,032 training/test images, and MNIST/USPS/MNIST-M are composed of 10,000/2,007/10,000 test images from 10 classes, respectively. All digit datasets have class imbalance as illustrated in Figure 5. SVHN is composed of colored real-world digit images. MNIST and USPS are composed of handwritten digits. MNIST-M is generated by combining MNIST digits and BSDS500 (Arbelaez et al., 2010) backgrounds. PACS benchmark consists of samples from seven classes including dogs and elephants in four domains: photo, art, cartoon, and sketch. In PACS, we test the robustness of classifiers across 12 different scenarios, each using the four domains as training and test domains, respectively. The data generation/collection process of the digit classification and PACS benchmarks is different across domains, resulting in differently imbalanced class distribution, as illustrated in Figure 2.

## A.2    DETAILS ABOUT PRE-TRAINING

We use ResNet-18 for digit classification, ResNet-26 for CIFAR datasets, and ResNet-50 for PACS and ImageNet-C as backbone networks. We use publicly released trained models and codes for a fair comparison. Specifically, for CIFAR-10/100 [1], we train the model with 200 epochs, batch size 200, SGD optimizer, learning rate 0.1, momentum 0.9, and weight decay 0.0005. For PACS, we use released pre-trained models of TTAB (Zhao et al., 2023) [2]. For ImageNet-C, we use the released

---

[1] https://github.com/locuslab/tta_conjugate
[2] https://github.com/LINs-lab/ttab

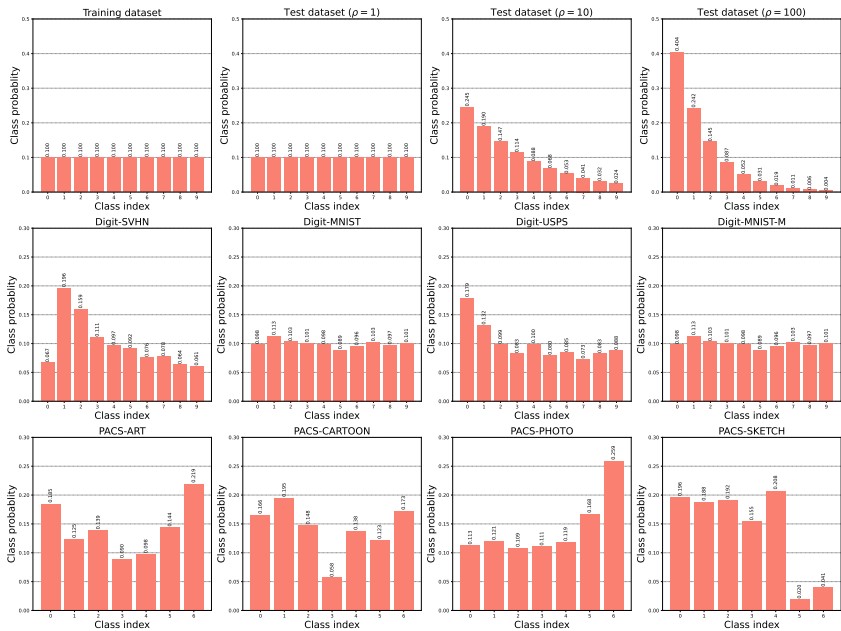

Figure 5: Class distribution of all benchmarks.

pre-trained models in the PyTorch library (Paszke et al., 2019) as described in Niu et al. (2023). For digit classification, we train the model with 50 epochs, batch size 256, SGD optimizer, learning rate 0.01, and weight decay 0.0005 with cosine annealing.

## A.3 DETAILS ABOUT INTERMEDIATE TIME TRAINING

We use a 2-layer MLP (Haykin, 1998) for the distribution shift aware module $g_\phi$. $g_\phi$ is composed of two fully connected layers and ReLU (Agarap, 2018). The hidden dimension of the distribution shift-aware module is set to 1,000. During the intermediate time, we train the $g_\phi$ by experiencing several batches with diverse class distributions using the labeled training dataset. For digit classification benchmark, we train $g_\phi$ with SGD optimizer (Ruder, 2016), a learning rate of 0.001, and cosine annealing for 100 epochs. For other benchmarks, we train $g_\phi$ with Adam optimizer (Kingma & Ba, 2014), a learning rate of 0.001, and cosine annealing for 100 epochs. To make intermediate batches having diverse class distributions, we use Dirichlet sampling with two hyperparameters, the Dirichlet sampling concentration parameter $\delta$, and the number of chunks $N_{\mathrm{dir}}$. As these two hyperparameters increase, the class distributions of intermediate batches become similar to the uniform. $\delta$ is set to 0.001 for ImageNet-C, 10 for digit classification, and 1 for other benchmarks. $N_{\mathrm{dir}}$ is set to 2000 for ImageNet-C, and to the value obtained by dividing the intermediate dataset size by the intermediate batch size for other benchmarks, e.g. 250 for CIFAR-10C-LT. The intermediate batch size is set to 50 for ImageNet-C and 200 for other benchmarks.

We use the labeled dataset in the training domain to train the distribution shift-aware module. Specifically, on CIFAR benchmarks and PACS, there is no auxiliary dataset in the training domain and we use the training dataset as an intermediate dataset. On the other hand, in the digit classification benchmark, we use the SVHN test dataset as an intermediate dataset. For ImageNet-C, the intermediate dataset is a subset of the ImageNet training dataset, composed of 50 samples randomly selected from each of the classes.

## A.4 DETAILS ABOUT TEST-TIME ADAPTATION METHODS

For a fair comparison, we fine-tune the Batch Normalization (BN) layer parameters unless otherwise specified. We use the Adam optimizer with a learning rate of 0.001 for all TTA methods in all experiments, except on ImageNet-C, following the approach in TENT (Wang et al., 2020). We set

the test batch size to 32/64/200 for PACS, ImageNet-C, and the other benchmarks. We run on 4 different random seeds for the intermediate-time and test-time training (0,1,2, and 3).

**BNAdapt (Schneider et al., 2020)** BNAdapt does not update the parameters in the trained model, but it corrects the BN statistics using the BN statistics computed in the test domain in exponential moving average with momentum 0.1.

**TENT (Wang et al., 2020)** TENT replaces the BN statistics of the trained classifier with the BN statistics computed in each test batch during test time. TENT only optimizes the BN layer parameters to minimize the prediction entropy of the test data.

**PL (Lee, 2013)** PL regards the test data with confident predictions as reliable pseudo-labeled data and fine-tunes the BN layer parameters to minimize cross-entropy loss using these pseudo-labeled data. We set the confidence threshold to 0.9 for filtering out test data with unconfident predictions.

**NOTE (Gong et al., 2022)** NOTE aims to mitigate the negative effects of non-i.i.d stream during test time by instance-aware BN (IABN) and prediction-balanced reservoir sampling (PBRS). IABN first detects whether a sample is out-of-distribution or not, by comparing the instance normalization (IN) and BN statistics for each sample. For in-distribution samples, IABN uses the standard BN statistics, while for out-of-distribution samples, it corrects the BN statistics using the IN statistics. We set the hyperparameter to determine the level of detecting out-of-distribution samples to 4 as used in NOTE (Gong et al., 2022). Due to non-i.i.d stream, class distribution within each batch is highly imbalanced. Thus, PBRS stores an equal number of predicted test data for each class and does test-time adaptation using the stored data in memory. We set the memory size the same as the batch size, for example, 200 for CIFAR benchmarks. NOTE and ODS create prediction-balanced batches and utilize them for adaptation. Thus, the batches for adaptation in ODS and NOTE have different class distributions from the test dataset, unlike other baselines including TENT. Therefore, in NOTE and ODS, DART is used exclusively to enhance the prediction accuracy of the examples stored in memory.

**DELTA (Zhao et al., 2022)** DELTA aims to alleviate the negative effects such as wrong BN statistics and prediction bias by test-time batch renormalization (TBR) and dynamic online reweighting (DOT). Since the BN statistics computed in the test batch are mostly inaccurate, TBR corrects the BN statistics with renormalization using test-time moving averaged BN statistics with a factor of 0.95. DOT computes the class prediction frequency in exponential moving average with a factor of 0.95 during test time and uses the estimated class prediction frequency to assign low/high weights to frequent/infrequent classes, respectively.

**LAME (Boudiaf et al., 2022)** LAME modifies the prediction by Laplacian regularized maximum likelihood estimation considering nearest neighbor information in the embedding space of the trained classifier. We compute the similarity among samples for the nearest neighbor information with k-NN with $k = 5$.

**ODS (Zhou et al., 2023)** ODS estimates label distribution of test data using the refined label distribution by LAME and adapts the trained classifiers using IABN and PBRS like NOTE, while assigning high/low weights on infrequent/frequent classes, respectively. Thus, we use the same hyperparameters used in LAME and NOTE.

**LSA (Park et al., 2023)** LSA estimates the label distribution of test data and produces an affine layer for feature representation for test data and parameter perturbation for the last linear layer for trained classifiers by taking the estimated label distribution. During intermediate time, the LSA is trained to output affine parameters for the feature representation $\gamma \in \mathbb{R}^{1 \times d}$ and $\beta \in \mathbb{R}^{1 \times d}$ and parameter perturbations $\Delta W \in \mathbb{R}^{d \times C}$ and $\Delta b \in \mathbb{R}^{1 \times C}$ for the last linear classifier weighted by $W$ and $b$ by taking ground truth label distribution. Specifically, when $z \in \mathbb{R}^d$ is a feature representation for a test data $x$, the refined prediction $\hat{y}$ for $(x, y) \in \mathcal{D}$ is

$$\hat{y} = (\gamma z + \beta)(W + \Delta W) + (b + \Delta b). \tag{3}$$

The LSA trains a label shift adapter to match the refined prediction $\hat{y}$ and the ground truth label $y$ using the logit adjusted loss. At test time, the LSA estimates the pseudo-label distribution in an online manner similar to DELTA,

$$\hat{q}_t = \alpha \bar{y}_t + (1 - \alpha)\hat{q}_{t-1}, \tag{4}$$

where $\hat{q}_t$ is the estimated test label distribution at time $t$, $\alpha$ is momentum hyperparameter, and $\bar{y}_t$ is the averaged model prediction of test batch at time $t$. $\alpha$ is set to 0.1. Then, the label shift adapter takes the estimated test label distribution $\hat{q}$ as an input during the test time. Similar to DART, LSA is a plug-in method that can be used in any existing entropy-minimization TTA methods.

The label shift adapter structure is a 2-layer MLP with hidden dimension 100. During intermediate time, LSA originally experiences several batches having three types of class distributions (forward, uniform, and backward class distributions) to train the label shift adapter. Forward/ backward indicates a class distribution that is same/inverse order of the label distribution of the training dataset, respectively. If the training class distribution is uniform, then LSA can only experience uniformity during the intermediate time.

## A.5 DART ON LARGE-SCALE BENCHMARK

As the number of classes $K$ increases, the output dimension of $g_\phi$ also increases as $K^2$. For instance, in CIFAR-100C-LT, the output dimension of $g_\phi$ is 10,000. The high output dimension makes it hard to learn and generate good square matrix $T$. To address it, we modify the module $g_\phi$ to produce $T$ with some entries fixed to 0 for the large-scale datasets. For CIFAR-100C, we first analyze class-wise confusion patterns using an augmented training dataset. Then we set the entries where class-wise confusion never occurred to 0 when training the distribution-shift aware module to generate $T$. For example, when we use a speckle noise augmentation of severity level 1, we can set 7,400 entries of $T$ to 0. We note that the noise type is not used when testing the model with CIFAR-100C test set. On the other hand, for ImageNet-C, we set the off-diagonal entries to 0 since the number of classes is huge.

During test time, $g_\phi$ takes the averaged pseudo label distribution over the test dataset to output a square matrix $T$ of size $K$. This is because in benchmarks like CIFAR-10C-LT, the class distribution of each test batch is similar to the one of the whole test dataset. However, for the online label distribution shift setup on ImageNet-C, the class distributions within test batches are different. Thus we compute the square matrix $T$ for each test batch.

CPL (Goyal et al., 2022) finds that TTA results can vary significantly when prediction confidence changes although pseudo-label accuracy is the same. DART on the large-scale benchmarks shows a similar phenomenon. Therefore, we perform TTA with normalization to maintain the prediction confidence on CIFAR-100C-LT. For example, in the case of TENT (Wang et al., 2020), we adapt the classifier $f_\theta$ using a training objective $\mathcal{L}_{\text{TENT}}(\theta) = \mathbb{E}_{\hat{x} \in \mathcal{D}_{\text{test}}}[-\sum_k \text{softmax}(\|f_\theta(\hat{x})\|_2 \frac{f_\theta(\hat{x})T_{\text{test}}}{\|f_\theta(\hat{x})T_{\text{test}}\|_2})_k \log \text{softmax}(\|f_\theta(\hat{x})\|_2 \frac{f_\theta(\hat{x})T_{\text{test}}}{\|f_\theta(\hat{x})T_{\text{test}}\|_2})_k]$. On the other hand, DART using the normalization shows similar performance compared to the original DART on CIFAR-10. For example, TENT+DART with the normalization achieves the test accuracy of 85.63±0.19 on CIFAR-10C-LT ($\rho = 10$).

## A.6 RUNTIME

We conduct experiments on RTX A6000. It takes about 2 hours to train $g_\phi$ during intermediate time for CIFAR-10C-LT. We train $g_\phi$ only once for each pre-trained classifier. Since we train a 2-layer MLP during the intermediate time, it requires a shorter training time compared to pre-training.

## B MORE RELATED WORKS

### B.1 TTA METHOD UTILIZING INTERMEDIATE TIME

Some recent works (Choi et al., 2022; Lim et al., 2022; Park et al., 2023) try to prepare an unknown test-time distribution shift by utilizing the training dataset at the time after the training phase and before the test time, called intermediate time. SWR (Choi et al., 2022) and TTN (Lim et al., 2022)

compute the importance of each layer in the trained model during intermediate time and prevent the important layers from significantly changing during test time. SWR and TTN compute the importance of each layer by computing cosine similarity between gradient vectors of training data and its augmented data. TTN additionally updates the importance with subsequent optimization using cross-entropy. Layers with lower importance are encouraged to change significantly during test time, while layers with higher importance are constrained to change minimally. On the other hand, our method DART trains a distribution shift aware module during intermediate time by experiencing several batches with diverse class distributions and learning how to modify the predictions generated by pre-trained classifiers to mitigate the negative effects caused by the class distribution shift of each batch.

### B.2 TTA METHODS CONSIDERING SAMPLE-WISE RELATIONSHIPS

Some recent works (Boudiaf et al., 2022; Iwasawa & Matsuo, 2021; Jang et al., 2022) focus on prediction modification using the nearest neighbor information based on the idea that nearest neighbors in the embedding space of the trained classifier share the same label. T3A (Iwasawa & Matsuo, 2021) replaces the last linear layer of the trained classifier with the prototypical classifier, which predicts the label of test data to the nearest prototype representing each class in the embedding space. LAME (Boudiaf et al., 2022) modifies the prediction of test data by Laplacian-regularized maximum likelihood estimation considering clustering information.

### B.3 LOSS CORRECTION METHODS FOR LEARNING WITH LABEL NOISE

In learning with label noise (LLN), it is assumed that there exists a noise transition matrix $T$, which determines the label-flipping probability of a sample from one class to other classes. For LLN, two main strategies have been widely used in estimating $T$: 1) using anchor points (Xia et al., 2019; Yao et al., 2020), which are defined as the training examples that belong to a particular class almost surely, and 2) using the clusterability of nearest neighbors of a training example belonging to the same true label class (Zhu et al., 2021). LLN uses the empirical pseudo label distribution of the anchor points or nearest neighbors to estimate $T$.

For TTA, on the other hand, the misclassification occurs not based on a fixed label-flipping pattern, but from the combination of covariate shift and label distribution shift. To adjust the pre-trained model against the covariate shifts, most TTA methods apply the BN adaptation, which updates the Batch Norm statistics using the test batches. However, when there exists label distribution shift in addition to the covariate shift, since the updated BN statistics follows the test label distribution, it induces bias in the classier (by pulling the decision boundary closer to the head classes and pushing the boundary farther from the tail classes as in Appendix C). Thus, the resulting class-wise confusion pattern depends not only on the class-wise relationship in the embedding space but also on the classifier bias originated from the label distribution shift and the updated BN statistics. Such a classifier bias has not been a problem for LLN, where we don't modify the BN statistics of the classifier at the test time.

Our proposed method, DART, focuses on this new class-wise confusion pattern, and is built upon the idea that if the module experiences various batches with diverse class distributions before the test time, it can develop the ability to refine inaccurate predictions resulting from label distribution shifts. Based on this intuition, we train a distribution shift-aware module during the intermediate time, by exposing several batches with diverse class distributions using the training datasets. As described in Equation (1) of the manuscript, the module is trained using the labeled training dataset to output a square matrix of the class dimension for prediction refinement. In this process, the module takes the averaged pseudo-label distribution as an input to learn the class-wise confusion pattern of the BN-adapted classifier depending on the label distribution shift.

## C   MOTIVATING TOY EXAMPLE

To understand the effects of test-time class distribution shift, we consider a four-class Gaussian mixture distribution with mean centering similar to batch normalization. Let the distribution of class $i$ is $\mathcal{N}(\mu_i, \sigma^2 I_2)$ at training time for $i = 1, 2, 3$, and 4, where $\mu_i \in \mathbb{R}^2$ is the mean of each class distribution. We set the mean of each class as $\mu_1 = (d, \beta d), \mu_2 = (-d, \beta d), \mu_3 = (d, -\beta d)$, and

$\mu_4 = (-d, -\beta d)$, where $\beta$ controls the distances between the classes, and we assume that $\beta > 1$ without loss of generality. Moreover, we assume that the four classes have the same prior probability at training time, *i.e.*, $p_{\text{tr}}(y = i) = 1/4, i = 1, 2, 3$, and 4. Since the class priors for the training data are equal, the Bayes classifier $f_{\text{tr}}$ predicts $x$ to the class $i$ when

$$p_{\text{tr}}(x|y = i) > p_{\text{tr}}(x|y = j), \quad j \neq i \tag{5}$$

due to Bayes' rule. Then, we have

$$f_{\text{tr}}(x) = \begin{cases} 1, & \text{if } x_1 > 0, x_2 > 0; \\ 2, & \text{if } x_1 < 0, x_2 > 0; \\ 3, & \text{if } x_1 > 0, x_2 < 0; \\ 4, & \text{if } x_1 < 0, x_2 < 0.. \end{cases} \tag{6}$$

At the test time, we assume that the class distribution is imbalanced, similar to the long-tailed distribution mainly discussed in the manuscript, as

$$p_{\text{te}}(y = 1) = p, \tag{7}$$
$$p_{\text{te}}(y = 2) = 1/4, \tag{8}$$
$$p_{\text{te}}(y = 3) = 1/4, \tag{9}$$
$$p_{\text{te}}(y = 4) = 1/2 - p. \tag{10}$$

Without loss of generality, we set $1/4 < p < 1/2$. Due to the mean centering, the distribution of class $i$ is shifted to $\mathcal{N}(\mu_i', \sigma^2 I_2)$, where $\mu_i'$ is the shifted class mean as follows:

$$\mu_1' = ((3/2 - 2p)d, (3/2 - 2p)\beta d), \tag{11}$$
$$\mu_2' = ((-1/2 - 2p)d, (3/2 - 2p)\beta d), \tag{12}$$
$$\mu_3' = ((3/2 - 2p)d, (-1/2 - 2p)\beta d), \tag{13}$$
$$\mu_4' = ((-1/2 - 2p)d, (-1/2 - 2p)\beta d). \tag{14}$$

Then, the probability that the samples from class 1 is wrongly classified to class 2 can be computed as

$$\Pr[f_{\text{tr}}(x) = 2|y = 1] = \Pr_{x=(x_1,x_2)\ \mathcal{N}(\mu_1',\sigma^2 I_2)}[x_1 < 0, x_2 > 0] \tag{15}$$

$$= \Phi\left(-\frac{(3/2 - 2p)d}{\sigma}\right)\left\{1 - \Phi\left(-\frac{(3/2 - 2p)\beta d}{\sigma}\right)\right\}, \tag{16}$$

where $\Phi$ is the standard normal cumulative density function. Similarly, the probability that the samples from class 2 is wrongly classified to class 1 can be computed as

$$\Pr[f_{\text{tr}}(x) = 1|y = 2] = \left\{1 - \Phi\left(-\frac{(-1/2 - 2p)d}{\sigma}\right)\right\}\left\{1 - \Phi\left(-\frac{(3/2 - 2p)\beta d}{\sigma}\right)\right\}. \tag{17}$$

Since $1/4 < p < 1/2$, we have $\Pr[f_{\text{tr}}(x) = 2|y = 1] > \Pr[f_{\text{tr}}(x) = 1|y = 2]$. With similar computations, we can obtain $\Pr[f_{\text{tr}}(x) = i|y = 1] > \Pr[f_{\text{tr}}(x) = 1|y = i], \forall i = 2, 3$, and 4. In other words, the probability that the samples from the class of a larger number of samples are confused to the rest of classes is greater than the inverse direction.

The probability that samples from class 1 are wrongly classified by $f_{\text{tr}}$ as class 1,2, and 3 can be calculated as follows:

$$\Pr[f_{\text{tr}}(x) = 2|y = 1] = \Phi\left(-\frac{(3/2 - 2p)d}{\sigma}\right)\left\{1 - \Phi\left(-\frac{(3/2 - 2p)\beta d}{\sigma}\right)\right\}, \tag{18}$$

$$\Pr[f_{\text{tr}}(x) = 3|y = 1] = \Phi\left(-\frac{(3/2 - 2p)\beta d}{\sigma}\right)\left\{1 - \Phi\left(-\frac{(3/2 - 2p)d}{\sigma}\right)\right\}, \tag{19}$$

$$\Pr[f_{\text{tr}}(x) = 4|y = 1] = \Phi\left(-\frac{(3/2 - 2p)\beta d}{\sigma}\right)\Phi\left(-\frac{(3/2 - 2p)d}{\sigma}\right). \tag{20}$$

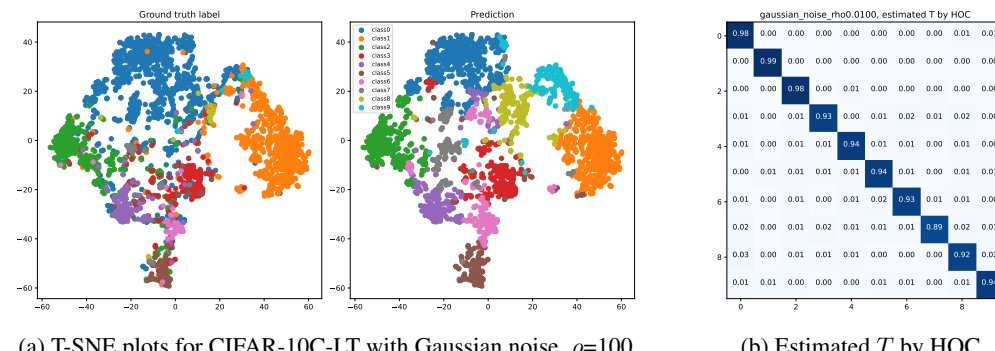

(a) T-SNE plots for CIFAR-10C-LT with Gaussian noise, $\rho$=100          (b) Estimated $T$ by HOC

Figure 6: (a) T-SNE plots of test data with ground truth labels (left) and their predictions (right) for CIFAR-10C-LT with Gaussian noise, $\rho$=100 (b) Estimated $T$ by HOC

Note that $\Phi\left(-\frac{(3/2-2p)\beta d}{\sigma}\right)$ has the following properties: Since $1/4 < p < 1/2$, $\Phi\left(-\frac{(3/2-2p)\beta d}{\sigma}\right) < 1/2$; Since $\beta > 1$, $\Phi\left(-\frac{(3/2-2p)\beta d}{\sigma}\right) < \Phi\left(-\frac{(3/2-2p)d}{\sigma}\right)$; $\frac{\partial}{\partial p}\Phi\left(-\frac{(3/2-2p)\beta d}{\sigma}\right) = C_1\beta\exp\left(-\frac{(3/2-2p)^2\beta^2 d^2}{2\sigma^2}\right)$, where $C_1$ is a positive constant which are independent of $p$ and $\beta$, decreases as $\beta$ grows for $\beta > \frac{\sigma}{(3/2-2p)d}$.

Thus, we can say that

(1) The probability of samples from the head class (class 1) are being confused to tail classes is greater than the reverse direction, specifically, $\Pr[f_{tr}(x) = i|y = 1] > \Pr[f_{tr}(x) = 1|y = i], \forall i \neq 1$, where $f_{tr}$ is a Bayes classifier obtained using the training dataset.

(2) The probability that a sample from the head class is confused to the closer class is larger than the farther classes, specifically, $\Pr[f_{tr}(x) = 2|y = 1] > \Pr[f_{tr}(x) = 3|y = 1] > \Pr[f_{tr}(x) = 4|y = 1]$.

(3) The increasing confusing probability to close class is larger than the one to farther class as class distribution imbalance $p$ increases, specifically, $\frac{\partial}{\partial p}\Pr[f_{tr}(x) = 2|y = 1] > \frac{\partial}{\partial p}\Pr[f_{tr}(x) = 3|y = 1]$ when $2\sigma < d$.

The effects of test-time label distribution shift can be consistently observed not only in this toy example but also in real datasets, including CIFAR-10C-LT.

## D    TRANSITION MATRIX ESTIMATION BY NOISY LABEL LEARNING METHOD

HOC (Zhu et al., 2021) estimates the noisy label transition matrix for a given noisy label dataset under the intuition that the nearest neighbor in the embedding space of a trained classifier shares the same ground truth label. We found that HOC failed to estimate the transition matrix for CIFAR-10C-LT with the label distribution shift of $\rho = 100$. HOC estimates the transition matrix by using the empirical pseudo label distribution of nearest neighbors of each example. However, as observed in Figure 6 left, the nearest neighbors in the embedding space already have the same pseudo labels/predictions for the BN-adapted classifier, which makes it impossible to estimate a correct $T$ depending on the label distribution shift. Thus, the estimated matrix by HOC is similar to the identity matrix as observed in Figure 6 right.

Table 6: Average accuracy (%) on CIFAR-10C-LT of two Oracles that modify *(Oracle (logit))* the classifier output and *(Oracle (prob))* the softmax output, respectively.

| Method | $\rho = 1$ | $\rho = 10$ | $\rho = 100$ |
|---|---|---|---|
| NoAdapt | 71.68±0.00 | 71.28±0.08 | 71.13±0.17 |
| BNAdapt | 85.24±0.08 | 79.01±0.07 | 66.90±0.16 |
| Oracle (logit) | 85.53±0.05 | 85.97±0.18 | 87.77±0.07 |
| Oracle (prob) | 85.24±0.07 | 81.03±0.12 | 78.57±0.16 |

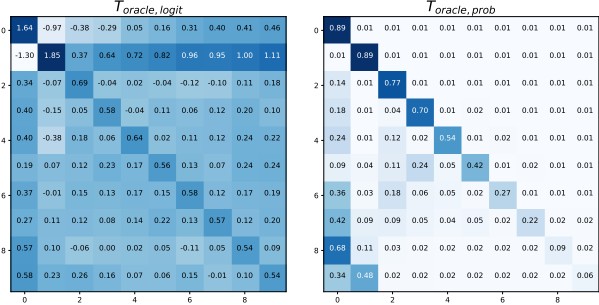

Figure 7: Comparison of trained $T$ of two oracles modifying logit and softmax probability.

# E  ADDITIONAL EXPERIMENTS

## E.1  ORACLE: SOFTMAX OUTPUT MODIFICATION VS LOGIT MODIFICATION

The Oracle method in Section 2.1 modifies the classifier output (logit) by simply multiplying $T_{\text{oracle, logit}}$ that minimizes the following objective

$$T_{\text{oracle, logit}} = \underset{T \in \mathbb{R}^{K \times K}}{\arg\min} \, \mathbb{E}_{(x,y) \in \mathcal{D}_{\text{test}}}[\text{CE}(\text{softmax}(f_\theta(x)T), y)] \tag{21}$$

by gradient descent. However, noisy label learning methods such as HOC usually modify the softmax output, not the logit. Thus, we consider a new Oracle method that modifies the softmax output by simply multiplying $T_{\text{oracle, prob}}$ that minimizes the following objective

$$T_{\text{oracle, prob}} = \underset{T \in \{T \in \mathbb{R}^{K \times K} : \sum_j T_{ij} = 1, 0 \leq T_{ij} \leq 1\}}{\arg\min} \, \mathbb{E}_{(x,y) \in \mathcal{D}_{\text{test}}}[\text{CE}(\text{softmax}(f_\theta(x))T, y)] \tag{22}$$

by gradient descent. In Table 6, we present the test accuracy achievable with $T_{\text{oracle}}$ when applied to the output of the BNAdapt model on CIFAR-10C-LT. We can observe that the Oracle that modifies the logits is more effective in mitigating performance degradation by test-time distribution shift regardless of the class imbalance ratio $\rho$. Thus, the distribution shift-aware module $g_\phi$ of DART focuses on generating a square matrix $T$ that modifies logit, not softmax output.

## E.2  ITERATIVE UPDATES OF T

We consider the variant of DART, which modifies the classifier output and obtains a square matrix by taking the modified classifier outputs iteratively. Specifically, for $i \in \mathbb{N}$

$$T_i = g_\phi(\mathbb{E}_{x \in \mathcal{D}_{\text{test}}}[\text{softmax}(f_\theta(x)\Pi_{j=0}^{i-1}T_j)]), \tag{23}$$

where $T_0$ is set to an identity matrix of size $K$. In Table 7, we observe that the refined pseudo label distribution is similar to the ground truth label distribution when modifying the prediction only once by DART (iteration=1). However, the performance gradually decreases as the number of iterations increases, which shows that the iterative updates does not help in improving the performance. We conjecture that these results originated from the fact that $g_\phi$ is trained to learn how to correct the classifier output of the pre-trained classifier $f_\theta$, but not any classifier including $f_\theta \Pi_{j=0}^{i-1} T_j$.

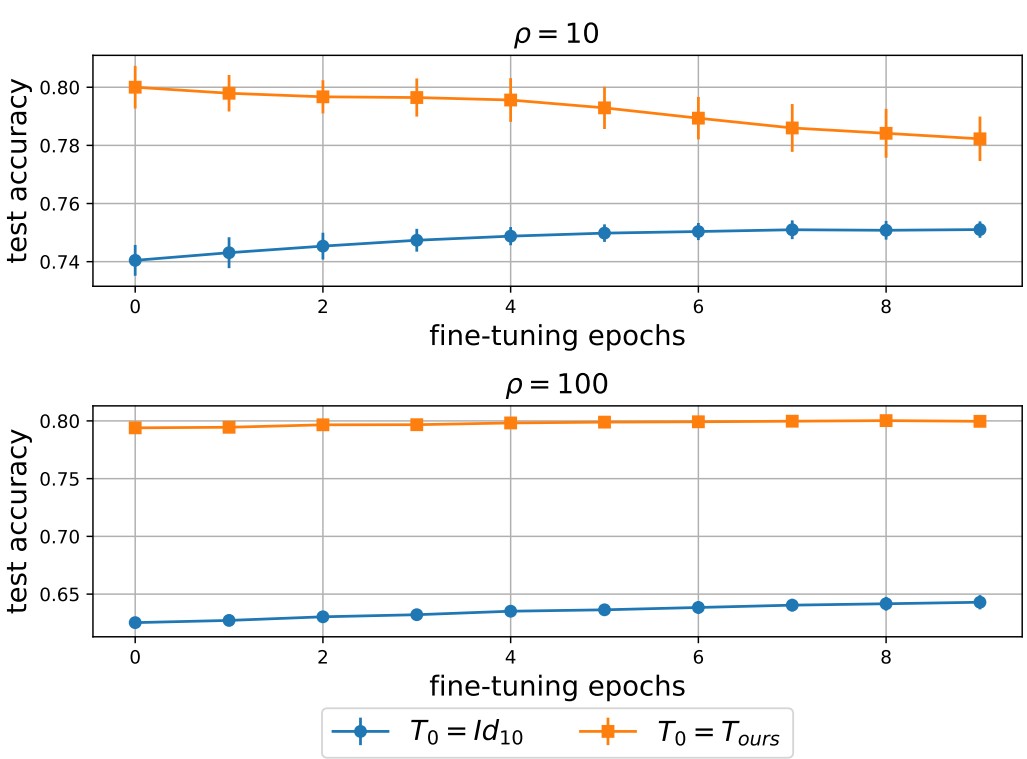

Figure 8: Changes of test accuracy while learning/fine-tuning the square matrix $T$ using the confident pseudo-labeled test data

Table 7: Iterative update of $T$ on CIFAR-10C-LT with Gaussian noise and $\rho$ of 10 and 100. Average accuracy (%) and pseudo-label distribution of the test data are reported.

| | | | 0 | 1 | 2 | 3 | 4 | 5 | 6 | 7 | 8 | 9 | |
|---|---|---|---|---|---|---|---|---|---|---|---|---|---|
| | | | \multicolumn{11}{c}{$\rho = 10$} | |
| | | | \multicolumn{11}{c}{Ground truth label distribution} | |
| iteration | acc | | 0 | 1 | 2 | 3 | 4 | 5 | 6 | 7 | 8 | 9 | |
| | | | 0.2449 | 0.1895 | 0.1467 | 0.1136 | 0.0879 | 0.0681 | 0.0526 | 0.0406 | 0.0316 | 0.0245 | |
| | | | \multicolumn{11}{c}{Pseudo label distribution} | |
| 0 | 0.7392 | | 0.1691 | 0.1570 | 0.1221 | 0.0971 | 0.0964 | 0.0816 | 0.0808 | 0.0639 | 0.0696 | 0.0624 | |
| 1 | 0.8009 | | 0.2269 | 0.2034 | 0.1382 | 0.0913 | 0.0972 | 0.0717 | 0.0703 | 0.0472 | 0.0391 | 0.0146 | |
| 2 | 0.6453 | | 0.4546 | 0.2592 | 0.1035 | 0.0558 | 0.0143 | 0.0503 | 0.0337 | 0.0209 | 0.0065 | 0.0010 | |
| 3 | 0.4817 | | 0.6098 | 0.3026 | 0.0078 | 0.0209 | 0.0021 | 0.0231 | 0.0094 | 0.0094 | 0.0113 | 0.0036 | |
| 4 | 0.4374 | | 0.6362 | 0.3378 | 0.0017 | 0.0051 | 0.0003 | 0.0045 | 0.0012 | 0.0047 | 0.0056 | 0.0028 | |
| | | | \multicolumn{11}{c}{$\rho = 100$} | |
| | | | \multicolumn{11}{c}{Ground truth label distribution} | |
| iteration | acc | | 0 | 1 | 2 | 3 | 4 | 5 | 6 | 7 | 8 | 9 | |
| | | | 0.4036 | 0.2417 | 0.1449 | 0.0868 | 0.0521 | 0.0311 | 0.0186 | 0.0109 | 0.0065 | 0.0040 | |
| | | | \multicolumn{11}{c}{Pseudo label distribution} | |
| 0 | 0.6240 | | 0.2021 | 0.1854 | 0.1245 | 0.0892 | 0.0805 | 0.0630 | 0.0660 | 0.0522 | 0.0701 | 0.0671 | |
| 1 | 0.7922 | | 0.3204 | 0.2408 | 0.1446 | 0.0795 | 0.0688 | 0.0413 | 0.0419 | 0.0203 | 0.0303 | 0.0122 | |
| 2 | 0.7019 | | 0.5846 | 0.2899 | 0.0450 | 0.0426 | 0.0038 | 0.0203 | 0.0083 | 0.0033 | 0.0018 | 0.0004 | |
| 3 | 0.6378 | | 0.6466 | 0.3101 | 0.0054 | 0.0174 | 0.0009 | 0.0052 | 0.0029 | 0.0045 | 0.0035 | 0.0033 | |
| 4 | 0.6246 | | 0.6608 | 0.3261 | 0.0009 | 0.0035 | 0.0001 | 0.0008 | 0.0001 | 0.0028 | 0.0026 | 0.0022 | |

Table 8: Average accuracy (%) on CIFAR-10C/10.1-LT, digit classification, and PACS.

| Method | CIFAR-10C-LT | | CIFAR-10.1-LT | | Digit | PACS |
|---|---|---|---|---|---|---|
| | $\rho = 10$ | $\rho = 100$ | $\rho = 10$ | $\rho = 100$ | | |
| NoAdapt | 71.28±0.08 | 71.13±0.17 | 87.13±0.48 | 86.64±0.97 | 58.45±0.00 | 60.65±0.00 |
| BNAdapt | 79.01±0.07 | 66.90±0.16 | 77.37±0.45 | 64.43±0.97 | 61.10±0.20 | 72.08±0.11 |
| BNAdapt+ours | 84.53±0.20 | 83.34±0.20 | 85.81±0.65 | 80.64±2.12 | 62.60±0.35 | 75.33±0.09 |
| TENT | 83.02±0.19 | 70.49±0.43 | 78.23±0.52 | 64.53±1.53 | 63.59±0.19 | 74.53±0.97 |
| TENT+ours | 85.13±0.31 | 88.56±0.13 | 86.88±0.78 | 82.32±1.60 | 64.85±0.44 | 80.98±1.19 |
| TTT++ | 80.15±0.21 | 68.64±0.37 | 77.74±0.35 | 64.74±0.75 | 60.86±0.06 | 67.11±0.20 |

### E.3 FINE-TUNE T WITH CONFIDENT PSEUDO-LABELED TEST DATA

DART uses the fixed square matrix $T_{\text{test}} = g_\phi(\bar{p})$ where $\bar{p}$ is an averaged pseudo label distribution for test data. We can consider a variant of DART that fine-tunes $T_{\text{test}}$ by the confident pseudo-labeled test data. Specifically, we can obtain by gradient descent

$$T^* = \arg\min_T \mathbb{E}_{x \in \mathcal{D}_{\text{test}}} \left[ \mathbf{1}\{\max \text{softmax} f_\theta(x) \geq \tau\} \text{CE}(\text{softmax}(f_\theta(x)T), \hat{p}_x) \right] + \alpha \text{MSE}(T, T_0), \tag{24}$$

where $\hat{p}_x = \arg\max f_\theta(x)$ is the pseudo label, $\tau$ is the confidence threshold, MSE is the mean square error, $\alpha$ is a hyperparameter for the regularization term, and $T_0$ is $T_{\text{test}}$. Moreover, one might consider obtaining the square matrix using only the confident pseudo-labeled test data (i.e., $T_0$ is set to an identity matrix of size $K$). Here, $\alpha$ and $\tau$ are set to 1 and 0.9, respectively.

In Figure 8, we summarize the test accuracy while fine-tuning $T$ for 10 epochs on CIFAR-10C-LT with Gaussian noise of $\rho = 10$ and 100. We find that (1) fine-tuning $T_{\text{test}}$ improves the test accuracy when $\rho$ is 100, but it worsens the test accuracy when $\rho$ is 10; (2) learning $T$ from scratch enhances the test accuracy, but it is marginal so is worse than one of DART. We conjecture that fine-tuning $T$ utilizing wrong pseudo labels can diminish the efficiency of DART.

### E.4 COMPARISON WITH TEST-TIME TRAINING METHOD TTT++

TTT++ adapts the trained classifiers using instance discrimination loss (contrastive learning) while aligning the feature statistics of training and test time. The original TTT++ performs contrastive learning in the embedding space of the trained contrastive head. However, the trained head can be available only when the instance discrimination loss is used during the training time. Thus, for a fair comparison, we consider a modified TTT++ which performs contrastive learning in the embedding space of the feature extractor. As data augmentation techniques for the contrastive learning, we use RandomHorizontalFlip, RandomResizedCrop, Grayscale, Normalize for digit classification, RandomHorizontalFlip, RandomResizedCrop, ColorJitter, RandomGrayscale, Normalize for CIFAR and PACS benchmarks as described in Liu et al. (2021). DART focuses on improving prediction accuracy that has been reduced due to the test-time class distribution shift. Therefore, DART can not be used as a plug-in method for TTT++ that does not use prediction in test-time training.

In Table 8, we summarize the results for the original and DART-applied TTA and TTT++. the modified TTT++ shows slightly better performances than BNAdapt on CIFAR and digit benchmarks, but it achieves lower performances compared to DART.

### E.5 DART ON BALANCED CIFAR-10C

We summarize the experimental results of DART on the balanced CIFAR-10C in Table 13. We observe that DART-applied TTA methods show worse performance than naive TTA methods. This is attributed to the limited gain even with Oracle method. In Table 1, Oracle achieved only a marginal performance gain of 0.3% on average even when using the labels of test data on balanced CIFAR-10C. Therefore, DART, which uses the same prediction modification scheme, can only achieve limited gains even when generating square matrices similar to the ones of Oracle. We note that experiments on balanced datasets are also challenging for ODS (Zhou et al., 2023), one of the methods alleviating test-time class distribution shift.

### E.6 SENSITIVITY ANALYSIS ON HYPERPARAMETERS

We verify the robustness against the changes of the structure of $g_\phi$ and the test batch size $B$. First, we conducted experiments to check the sensitivity of DART over the hidden dimension $d_h$ and number of layers of $g_\phi$, and the results are summarized on CIFAR-10C-LT of $\rho = 100$ in Table 9. We can observe that DART is robust against the change in the $g_\phi$ structure. And then, we conducted experiments to check the sensitivity of DART over $B$ and the results are summarized in Table 10. We can observe that DART is robust against the change in $B$.

Table 9: Sensitivity analysis about the network design of $g_\phi$.

| | 2-layer MLP | | | | 3-layer MLP | | | |
| | $d_h = 250$ | $d_h = 500$ | $d_h = 1000$ | $d_h = 2000$ | $d_h = 250$ | $d_h = 500$ | $d_h = 1000$ | $d_h = 2000$ |
|---|---|---|---|---|---|---|---|---|
| NoAdapt | 71.13 | | | | | | | |
| BNAdapt | 66.90 | | | | | | | |
| BNAdapt+DART (ours) | 80.6 | 82.17 | 83.34 | 83.83 | 83.83 | 84.27 | 84.78 | 84.97 |
| TENT | 70.49 | | | | | | | |
| TENT+DART (ours) | 87.46 | 88.23 | 88.56 | 88.65 | 88.81 | 88.67 | 88.6 | 88.09 |

Table 10: Sensitivity analysis about the test batch size $B$.

| | $B$=32 | $B$=64 | $B$=128 | $B$=256 |
|---|---|---|---|---|
| NoAdapt | 71.13 | | | |
| BNAdapt | 65.48 | 66.15 | 66.68 | 66.99 |
| BNAdapt+DART (ours) | 81.70 | 82.65 | 83.17 | 83.51 |
| TENT | 71.89 | 71.98 | 71.48 | 69.97 |
| TENT+DART (ours) | 85.63 | 88.20 | 88.86 | 88.30 |

### E.7 COMPARISON WITH SAR

SAR (Niu et al., 2023), which adapts the trained models to lie in a flat region on the entropy loss surface, is widely known as robust to label distribution shifts. Since DART focuses on effectively modifying the inaccurate predictions/pseudo-labels caused by test-time label distribution shifts, DART can be integrated as a plug-in method with any TTA methods, including SAR, that rely on pseudo-labels obtained from the trained classifiers. Thus, DART can also be used with SAR, and we summarize the experimental results on CIFAR-10C-LT in Table 11, and on ImageNet-C-imbalance in Table 12. We can observe that the performances of SAR are worse/better than those of TENT on CIFAR-10C-LT/ImageNet-C-imbalance, respectively. However, DART consistently improves the performance of the SAR in a similar way as it improves the performances of other TTA methods, since DART improves the accuracy of the initial pseudo-labels used for SAR.

Table 11: Average accuracy (%) on CIFAR-10C-LT

|  | $\rho = 10$ | $\rho = 100$ |
| --- | --- | --- |
| NoAdapt | 71.28 | 71.13 |
| BNAdapt | 79.01 | 66.9 |
| BNAdapt+DART (ours) | 84.53 (+5.52) | 83.34 (+16.44) |
| TENT | 83.02 | 70.49 |
| TENT+DART (ours) | 85.13 (+2.11) | 88.56 (+18.07) |
| SAR | 79.76 | 67.3 |
| SAR+DART (ours) | 84.90 (+5.14) | 83.56 (+16.26) |

Table 12: Average accuracy (%) on ImageNet-C-imbalance

|  | $\alpha = 1000$ | $\alpha = 2000$ | $\alpha = 5000$ |
| --- | --- | --- | --- |
| NoAdapt | 18.15 | 18.16 | 18.16 |
| BNAdapt | 19.85 | 14.11 | 8.48 |
| BNAdapt+ours | 25.18 (+5.33) | 20.48 (+6.37) | 14.82 (+6.34) |
| TENT | 22.49 | 13.52 | 6.61 |
| TENT+ours | 26.18 (+3.69) | 18.51 (+4.99) | 11.17 (+4.56) |
| SAR | 26.46 | 17.36 | 9.09 |
| SAR+ours | 32.49 (+6.03) | 23.38 (+6.02) | 12.9 (+3.81) |

# F  FULL RESULTS

Table 13: Test accuracy of CIFAR-10C-LT with $\rho = 1$ when the model is fine-tuned using Adam optimizer by only one epoch.

| | gaussian_noise | shot_noise | impulse_noise | defocus_blur | glass_blur | motion_blur | zoom_blur | snow | frost | fog | brightness | contrast | elastic_transform | pixelate | jpeg_compression | avg |
|---|---|---|---|---|---|---|---|---|---|---|---|---|---|---|---|---|
| NoAdapt | 46.41±0.00 | 51.61±0.00 | 27.11±0.00 | 90.69±0.00 | 67.95±0.00 | 82.06±0.00 | 91.87±0.00 | 83.84±0.00 | 80.20±0.00 | 68.79±0.00 | 91.20±0.00 | 51.59±0.00 | 82.77±0.00 | 81.19±0.00 | 77.94±0.00 | 71.68±0.00 |
| BNAdapt | 80.48±0.12 | 81.85±0.10 | 69.94±0.19 | 91.44±0.15 | 80.18±0.20 | 88.01±0.06 | 92.69±0.07 | 86.27±0.13 | 88.51±0.14 | 83.60±0.12 | 91.26±0.05 | 89.49±0.08 | 83.95±0.10 | 90.46±0.22 | 80.39±0.18 | 85.24±0.08 |
| BNAdapt+ours | 79.92±0.15 | 81.04±0.10 | 69.60±0.16 | 91.02±0.20 | 79.31±0.18 | 87.39±0.21 | 92.31±0.08 | 85.81±0.09 | 88.04±0.17 | 82.94±0.20 | 90.92±0.10 | 89.12±0.07 | 83.47±0.26 | 89.95±0.09 | 79.72±0.28 | 84.70±0.12 |
| BNAdapt+ours (diag) | 79.51±0.31 | 80.48±0.31 | 69.88±0.15 | 90.61±0.19 | 79.31±0.19 | 87.05±0.23 | 91.79±0.13 | 85.61±0.19 | 87.63±0.15 | 82.69±0.11 | 90.65±0.09 | 88.99±0.12 | 83.21±0.16 | 89.77±0.14 | 79.78±0.10 | 84.46±0.14 |
| BNAdapt+ours (online) | 79.71±0.15 | 80.75±0.25 | 68.78±0.60 | 90.46±0.38 | 79.15±0.51 | 86.92±0.25 | 91.36±0.37 | 85.35±0.31 | 87.41±0.32 | 82.36±0.42 | 90.38±0.23 | 88.45±0.21 | 82.94±0.42 | 89.46±0.39 | 78.97±0.97 | 84.16±0.24 |
| TENT | 82.34±0.12 | 83.56±0.42 | 73.62±0.47 | 91.35±0.30 | 80.98±0.74 | 88.54±0.31 | 92.23±0.15 | 87.54±0.34 | 88.54±0.47 | 87.61±0.42 | 91.50±0.25 | 90.47±0.73 | 83.29±0.35 | 90.20±0.39 | 83.44±0.47 | 86.35±0.22 |
| TENT+ours | 77.85±1.04 | 77.46±1.63 | 70.88±0.74 | 88.98±0.64 | 77.04±1.16 | 85.94±0.74 | 89.57±0.61 | 84.91±0.98 | 85.47±0.84 | 85.52±0.98 | 89.00±0.49 | 88.32±0.87 | 80.49±0.73 | 86.96±0.80 | 80.20±0.66 | 83.24±0.75 |
| TENT+ours (diag) | 72.12±1.75 | 70.64±2.56 | 66.35±2.22 | 81.97±2.62 | 71.29±1.87 | 78.44±1.83 | 82.56±2.58 | 77.18±2.20 | 78.35±2.91 | 78.71±3.30 | 82.97±2.23 | 80.68±2.40 | 72.96±1.80 | 80.20±2.41 | 75.02±1.43 | 76.63±2.22 |
| TENT+ours (online) | 72.79±3.52 | 74.10±2.43 | 61.70±2.93 | 80.98±4.05 | 67.03±6.82 | 77.36±2.81 | 79.09±3.82 | 75.77±3.14 | 77.43±5.03 | 78.01±4.08 | 80.60±4.11 | 78.21±6.04 | 70.41±6.14 | 80.38±2.44 | 70.31±6.14 | 74.95±3.25 |
| PL | 82.01±0.30 | 82.44±0.75 | 73.60±0.96 | 91.37±0.42 | 80.67±0.53 | 88.55±0.32 | 92.10±0.19 | 87.40±0.39 | 86.26±0.36 | 87.46±0.34 | 91.19±0.19 | 90.46±0.62 | 83.57±0.92 | 90.32±0.31 | 83.50±0.57 | 86.19±0.13 |
| PL+ours | 78.22±1.60 | 78.20±0.95 | 69.60±0.16 | 90.15±0.98 | 77.35±2.73 | 85.99±1.07 | 91.32±1.09 | 85.56±0.38 | 86.22±1.61 | 80.82±1.30 | 89.73±0.96 | 87.96±0.87 | 82.66±1.37 | 89.10±1.08 | 79.67±0.36 | 83.50±0.61 |
| PL+ours (diag) | 76.29±1.42 | 77.04±0.20 | 69.51±0.65 | 89.52±0.88 | 78.08±1.74 | 84.27±1.79 | 90.01±1.69 | 84.16±1.04 | 84.42±0.53 | 80.14±1.00 | 88.33±0.68 | 87.34±2.19 | 81.19±2.09 | 87.67±1.09 | 78.61±1.41 | 82.44±0.70 |
| PL+ours (online) | 73.95±4.31 | 75.58±3.88 | 62.75±5.30 | 78.61±7.21 | 71.58±4.08 | 74.22±2.38 | 76.50±6.44 | 76.65±2.73 | 77.72±5.66 | 75.02±4.09 | 79.34±6.15 | 79.08±5.19 | 71.28±7.29 | 77.99±6.00 | 70.82±5.26 | 74.74±3.71 |
| DELTA | 80.99±0.88 | 82.40±0.57 | 72.38±0.52 | 91.52±0.32 | 80.66±0.41 | 88.38±0.43 | 92.03±0.20 | 87.28±0.45 | 88.36±0.35 | 87.15±0.47 | 91.03±0.37 | 90.44±0.41 | 83.39±0.44 | 90.17±0.18 | 82.88±0.37 | 85.94±0.16 |
| DELTA+ours | 77.26±0.67 | 78.36±1.15 | 69.44±1.43 | 89.63±0.35 | 77.18±1.28 | 86.19±0.88 | 90.48±0.66 | 85.41±0.83 | 86.09±1.08 | 85.85±0.86 | 89.78±0.52 | 89.33±0.89 | 79.92±1.28 | 87.89±0.84 | 80.76±0.57 | 83.57±0.77 |
| DELTA+ours (diag) | 70.08±3.37 | 69.26±3.82 | 63.20±2.31 | 81.07±3.26 | 68.89±2.16 | 77.00±3.50 | 82.13±3.88 | 74.90±4.25 | 76.12±5.23 | 76.55±5.62 | 81.68±3.42 | 78.99±3.94 | 71.32±3.10 | 78.61±3.45 | 73.38±2.11 | 74.88±3.38 |
| DELTA+ours (online) | 75.67±0.99 | 74.56±1.78 | 62.20±2.39 | 81.80±5.02 | 69.02±6.49 | 81.02±3.08 | 82.13±4.82 | 78.00±3.96 | 80.12±3.38 | 81.05±3.03 | 82.64±4.47 | 79.97±6.72 | 74.58±3.92 | 83.91±2.49 | 73.61±6.74 | 77.35±3.22 |
| NOTE | 74.11±1.14 | 75.57±0.96 | 66.05±0.84 | 86.87±0.42 | 72.53±0.77 | 84.40±0.59 | 88.03±0.15 | 82.43±0.82 | 83.92±0.80 | 82.88±0.65 | 87.53±0.97 | 88.46±0.81 | 76.50±0.61 | 84.64±0.13 | 76.00±0.74 | 80.66±0.21 |
| NOTE+ours | 73.00±1.41 | 75.08±0.78 | 64.76±1.43 | 86.09±0.26 | 71.51±1.54 | 83.74±0.91 | 87.51±0.78 | 82.19±0.65 | 83.32±0.29 | 82.78±0.42 | 86.93±0.80 | 87.99±1.08 | 76.15±1.40 | 83.50±0.12 | 75.23±0.59 | 79.99±0.41 |
| NOTE+ours (diag) | 72.66±2.17 | 73.99±0.44 | 65.45±1.19 | 85.81±1.21 | 72.08±1.23 | 83.63±0.07 | 86.97±0.22 | 81.47±1.23 | 82.58±0.75 | 82.43±0.56 | 86.54±0.63 | 88.17±0.59 | 75.93±0.71 | 84.16±0.11 | 75.17±0.55 | 79.80±0.41 |
| NOTE+ours (online) | 67.40±0.58 | 68.75±1.64 | 59.36±1.17 | 83.67±1.65 | 67.72±3.11 | 81.64±1.04 | 84.85±0.93 | 80.74±1.15 | 80.88±1.44 | 78.82±2.68 | 85.20±1.48 | 85.96±1.34 | 73.53±1.33 | 81.72±1.56 | 72.75±2.60 | 76.87±0.49 |
| LAME | 80.54±0.18 | 81.95±0.09 | 70.10±0.20 | 91.46±0.23 | 80.12±0.21 | 87.99±0.23 | 92.73±0.05 | 86.35±0.08 | 88.54±0.17 | 83.63±0.22 | 91.21±0.10 | 89.48±0.06 | 83.99±0.03 | 90.55±0.23 | 80.36±0.07 | 85.27±0.06 |
| LAME+ours | 80.49±0.26 | 81.83±0.07 | 70.06±0.23 | 91.43±0.28 | 80.07±0.27 | 87.92±0.23 | 92.71±0.08 | 86.35±0.07 | 88.51±0.18 | 83.58±0.23 | 91.20±0.15 | 89.47±0.10 | 83.91±0.07 | 90.51±0.19 | 80.24±0.13 | 85.22±0.05 |
| LAME+ours (diag) | 80.46±0.27 | 81.74±0.10 | 70.11±0.13 | 91.37±0.25 | 80.05±0.27 | 87.89±0.19 | 92.68±0.04 | 86.32±0.06 | 88.48±0.17 | 83.56±0.19 | 91.16±0.09 | 89.47±0.11 | 83.92±0.11 | 90.53±0.16 | 80.22±0.19 | 85.20±0.05 |
| LAME+ours (online) | 80.53±0.20 | 81.78±0.08 | 69.93±0.16 | 91.41±0.20 | 80.07±0.27 | 87.86±0.20 | 92.63±0.08 | 86.31±0.10 | 88.53±0.20 | 83.54±0.17 | 91.19±0.07 | 89.46±0.09 | 83.97±0.07 | 90.56±0.13 | 80.18±0.07 | 85.20±0.05 |
| ODS | 77.41±0.51 | 78.81±0.54 | 69.00±0.49 | 88.83±0.37 | 76.12±0.33 | 86.40±0.30 | 89.73±0.22 | 85.22±0.34 | 86.54±0.13 | 84.82±0.18 | 89.39±0.30 | 90.83±0.26 | 79.44±0.23 | 87.33±0.28 | 78.31±0.79 | 83.21±0.09 |
| ODS+ours | 77.50±0.28 | 78.78±0.48 | 70.02±0.83 | 89.01±0.31 | 76.36±0.35 | 86.57±0.62 | 89.89±0.41 | 85.17±0.25 | 86.79±0.30 | 85.29±0.54 | 89.53±0.34 | 90.92±0.30 | 79.74±0.57 | 87.75±0.07 | 78.03±0.25 | 83.42±0.11 |
| ODS+ours (diag) | 78.22±0.50 | 79.39±0.73 | 69.41±1.12 | 88.94±0.50 | 76.44±0.59 | 86.48±0.50 | 89.86±0.39 | 84.94±0.22 | 86.65±0.24 | 85.00±0.44 | 89.60±0.20 | 91.30±0.28 | 79.78±0.33 | 87.87±0.29 | 78.51±0.62 | 83.49±0.07 |
| ODS+ours (online) | 77.69±0.26 | 78.61±0.53 | 69.47±0.28 | 88.79±0.53 | 76.64±0.94 | 86.49±0.08 | 89.62±0.16 | 85.10±0.51 | 86.43±0.14 | 85.11±0.42 | 89.65±0.38 | 91.09±0.09 | 79.61±0.33 | 87.29±0.48 | 78.05±0.28 | 83.31±0.04 |

Table 14: Test accuracy of CIFAR-10C-LT with $\rho = 10$ when the model is fine-tuned using Adam optimizer by only one epoch.

| | gaussian_noise | shot_noise | impulse_noise | defocus_blur | glass_blur | motion_blur | zoom_blur | snow | frost | fog | brightness | contrast | elastic_transform | pixelate | jpeg_compression | avg |
|---|---|---|---|---|---|---|---|---|---|---|---|---|---|---|---|---|
| NoAdapt | 45.83±0.45 | 50.27±0.38 | 29.24±0.23 | 90.06±0.08 | 65.93±0.42 | 84.61±0.38 | 92.46±0.14 | 82.94±0.53 | 76.81±0.35 | 67.37±0.48 | 91.69±0.31 | 54.73±0.48 | 82.44±0.24 | 79.44±0.37 | 75.42±0.20 | 71.28±0.08 |
| BNAdapt | 73.92±0.48 | 75.24±0.25 | 64.50±0.18 | 85.60±0.16 | 73.45±0.46 | 82.46±0.31 | 86.95±0.21 | 80.12±0.70 | 81.70±0.49 | 76.60±0.23 | 86.10±0.38 | 83.57±0.12 | 77.51±0.37 | 83.75±0.35 | 73.67±0.08 | 79.01±0.07 |
| BNAdapt+ours | 80.09±0.72 | 80.01±0.42 | 70.55±0.28 | 90.20±0.43 | 79.79±0.41 | 87.80±0.26 | 91.50±0.30 | 86.07±0.60 | 86.32±0.38 | 82.62±0.49 | 90.62±0.38 | 87.87±0.33 | 84.32±0.30 | 90.08±0.37 | 80.17±0.40 | 84.53±0.20 |
| BNAdapt+ours (diag) | 79.19±0.49 | 79.81±0.23 | 69.40±0.24 | 90.13±0.10 | 78.67±0.41 | 87.12±0.27 | 91.53±0.36 | 85.32±0.64 | 86.47±0.66 | 81.52±0.51 | 90.62±0.36 | 88.52±0.23 | 82.87±0.28 | 88.97±0.33 | 78.89±0.43 | 83.94±0.15 |
| BNAdapt+ours (online) | 79.49±2.08 | 80.31±0.36 | 69.64±0.30 | 88.68±1.40 | 78.28±1.03 | 86.28±0.82 | 90.59±1.77 | 84.88±1.49 | 86.33±1.25 | 82.02±1.44 | 89.26±1.15 | 87.64±0.97 | 82.32±1.18 | 88.34±2.00 | 79.03±0.51 | 83.54±0.76 |
| TENT | 78.28±1.50 | 79.43±0.81 | 70.13±0.76 | 88.30±0.65 | 77.34±0.89 | 85.69±1.12 | 89.64±0.62 | 84.79±0.78 | 85.14±1.31 | 83.13±1.29 | 89.64±0.59 | 88.15±0.27 | 81.24±0.55 | 86.42±1.28 | 77.93±0.84 | 83.02±0.19 |
| TENT+ours | 81.63±0.80 | 82.22±1.00 | 75.34±0.51 | 89.56±0.60 | 80.37±0.50 | 86.76±0.34 | 89.89±0.69 | 86.58±0.58 | 86.62±0.47 | 86.31±0.78 | 89.21±0.44 | 88.81±0.22 | 83.39±0.61 | 88.93±0.40 | 82.22±1.09 | 85.13±0.31 |
| TENT+ours (diag) | 81.08±0.54 | 81.47±0.76 | 73.93±0.82 | 90.40±0.57 | 80.41±0.59 | 87.98±0.42 | 90.95±0.44 | 86.64±0.61 | 87.32±0.87 | 86.57±0.44 | 90.61±0.37 | 89.89±0.68 | 82.68±0.73 | 89.15±0.48 | 81.07±1.21 | 85.14±0.33 |
| TENT+ours (online) | 80.88±1.37 | 81.38±1.54 | 75.18±0.87 | 88.28±1.10 | 79.60±1.15 | 86.39±0.36 | 89.05±0.85 | 85.66±1.36 | 86.13±1.01 | 85.60±1.18 | 88.40±0.56 | 87.86±0.68 | 81.46±1.46 | 87.26±1.36 | 81.21±1.59 | 84.29±0.46 |
| PL | 78.52±1.44 | 78.89±0.89 | 69.17±1.23 | 88.77±0.64 | 76.63±1.80 | 85.90±0.62 | 90.19±0.62 | 84.84±0.28 | 86.20±0.38 | 83.20±1.40 | 89.60±1.11 | 87.95±1.01 | 81.12±0.71 | 87.29±0.52 | 78.04±0.80 | 83.09±0.28 |
| PL+ours | 80.86±0.97 | 81.10±0.52 | 73.25±0.55 | 88.79±0.63 | 79.62±0.40 | 87.26±0.19 | 89.33±0.40 | 86.17±0.92 | 86.15±0.66 | 85.50±0.70 | 89.38±0.25 | 88.14±0.32 | 82.76±0.63 | 88.22±0.74 | 81.02±0.81 | 84.50±0.39 |
| PL+ours (diag) | 80.53±0.92 | 81.38±0.59 | 69.89±1.45 | 90.45±0.40 | 80.28±0.69 | 87.53±0.71 | 90.81±0.16 | 86.57±1.21 | 87.35±0.94 | 84.39±0.37 | 90.57±0.17 | 89.21±0.74 | 82.60±0.61 | 89.29±0.50 | 80.24±1.18 | 84.74±0.47 |
| PL+ours (online) | 80.61±1.64 | 80.83±1.41 | 71.32±0.78 | 88.31±0.86 | 78.23±1.01 | 86.51±0.57 | 89.25±0.68 | 85.09±1.27 | 86.00±1.01 | 83.81±2.74 | 88.62±0.36 | 87.38±0.45 | 81.26±0.58 | 87.29±2.00 | 80.34±1.27 | 83.66±0.60 |
| DELTA | 76.85±4.70 | 77.64±1.37 | 68.20±1.79 | 87.84±1.12 | 78.51±1.36 | 85.21±1.43 | 89.81±1.42 | 84.54±1.38 | 84.05±4.27 | 83.10±2.26 | 88.15±1.83 | 86.37±1.22 | 80.59±1.53 | 86.94±2.87 | 78.40±1.83 | 82.46±0.39 |
| DELTA+ours | 80.54±1.35 | 80.69±1.72 | 72.29±0.96 | 89.57±0.71 | 79.25±0.74 | 86.77±0.20 | 89.09±0.50 | 85.73±0.16 | 86.85±0.46 | 86.06±0.58 | 89.14±0.41 | 88.60±0.66 | 82.62±0.58 | 88.37±0.71 | 81.27±0.75 | 84.61±0.30 |
| DELTA+ours (diag) | 76.57±1.33 | 77.50±2.31 | 66.04±2.82 | 89.04±1.22 | 75.55±2.34 | 85.77±0.60 | 89.46±0.32 | 84.20±0.91 | 85.09±1.35 | 84.06±0.88 | 89.08±0.68 | 88.72±1.96 | 78.81±1.98 | 86.33±1.48 | 76.97±1.77 | 82.21±0.82 |
| DELTA+ours (online) | 80.15±0.82 | 78.66±3.72 | 72.22±2.40 | 88.30±1.22 | 78.83±0.93 | 86.23±0.86 | 88.19±0.78 | 84.70±2.16 | 85.76±1.30 | 85.35±1.72 | 88.86±0.90 | 88.17±0.87 | 81.12±1.71 | 87.72±1.30 | 80.37±1.51 | 83.61±0.77 |
| NOTE | 72.26±2.57 | 72.74±1.64 | 64.80±1.26 | 87.88±0.95 | 72.71±1.54 | 84.59±1.34 | 88.58±0.58 | 83.18±0.62 | 84.39±1.47 | 81.90±1.25 | 88.70±0.53 | 89.71±0.62 | 76.79±3.26 | 86.62±0.73 | 76.00±1.62 | 80.72±0.23 |
| NOTE+ours | 74.96±1.57 | 74.84±0.51 | 68.92±1.14 | 85.74±0.88 | 73.87±1.07 | 84.07±0.33 | 85.76±0.77 | 84.00±0.45 | 84.46±0.57 | 81.09±0.95 | 87.00±0.54 | 87.84±0.43 | 77.92±0.08 | 85.35±0.42 | 75.85±1.17 | 80.79±0.14 |
| NOTE+ours (diag) | 74.57±1.37 | 74.02±0.56 | 69.04±0.88 | 86.86±1.25 | 74.57±0.47 | 85.43±0.82 | 86.69±0.73 | 85.13±0.39 | 84.63±0.77 | 81.76±0.95 | 88.11±0.83 | 89.54±1.88 | 78.70±1.42 | 85.60±0.67 | 77.68±1.53 | 81.62±0.35 |
| NOTE+ours (online) | 64.31±2.15 | 65.87±1.50 | 58.08±3.69 | 85.14±1.12 | 71.84±0.59 | 82.30±0.86 | 85.24±1.09 | 82.60±1.65 | 82.84±1.53 | 77.60±1.11 | 85.03±0.96 | 85.97±0.90 | 76.54±0.47 | 84.27±1.03 | 75.59±0.94 | 77.55±0.39 |
| LAME | 75.13±0.34 | 76.85±0.16 | 66.06±0.46 | 87.10±0.17 | 74.81±0.32 | 83.91±0.37 | 88.40±0.11 | 81.79±0.86 | 83.28±0.53 | 78.22±0.55 | 87.54±0.27 | 85.10±0.32 | 79.15±0.29 | 85.30±0.25 | 74.89±0.40 | 80.50±0.06 |
| LAME+ours | 77.45±0.64 | 77.92±0.36 | 68.01±0.61 | 88.09±0.27 | 77.13±0.39 | 85.22±0.17 | 89.75±0.62 | 83.80±0.64 | 84.10±0.57 | 80.32±0.30 | 89.06±0.56 | 86.07±0.23 | 81.89±0.33 | 87.44±0.30 | 77.72±0.52 | 82.27±0.20 |
| LAME+ours (diag) | 77.39±0.51 | 78.31±0.29 | 67.51±0.47 | 88.61±0.22 | 76.70±0.60 | 85.36±0.40 | 90.38±0.15 | 83.79±0.58 | 84.87±0.66 | 79.84±0.38 | 89.10±0.44 | 86.86±0.31 | 81.24±0.35 | 87.14±0.09 | 77.08±0.38 | 82.28±0.11 |
| LAME+ours (online) | 77.52±1.32 | 78.44±0.61 | 67.67±0.52 | 86.92±1.41 | 76.69±0.38 | 84.68±0.98 | 89.43±1.81 | 83.25±1.14 | 84.56±0.92 | 80.30±0.61 | 87.90±1.33 | 86.07±0.67 | 80.76±1.20 | 86.45±1.87 | 77.30±0.61 | 81.86±0.68 |
| ODS | 76.49±1.17 | 76.42±0.70 | 67.59±1.02 | 88.36±0.37 | 74.44±1.24 | 85.10±1.28 | 88.91±0.61 | 84.32±1.28 | 84.92±0.75 | 82.62±0.48 | 88.78±0.96 | 90.20±0.51 | 79.40±1.53 | 86.53±1.18 | 76.20±0.72 | 82.01±0.28 |
| ODS+ours | 76.25±0.64 | 77.22±0.55 | 68.13±1.11 | 89.74±0.30 | 75.26±1.02 | 87.16±0.34 | 90.27±0.14 | 85.91±0.48 | 85.93±0.94 | 83.39±0.44 | 90.30±0.52 | 91.28±0.33 | 80.98±1.05 | 87.95±0.79 | 78.31±1.60 | 83.23±0.15 |
| ODS+ours (diag) | 76.65±1.03 | 77.04±0.65 | 67.80±0.92 | 89.26±0.44 | 75.14±1.18 | 86.56±1.00 | 90.15±0.64 | 85.46±1.12 | 85.50±0.59 | 83.20±0.57 | 89.61±0.96 | 90.45±0.51 | 80.04±0.93 | 87.52±0.17 | 77.83±1.30 | 82.82±0.19 |
| ODS+ours (online) | 76.26±1.87 | 76.03±0.49 | 68.17±0.75 | 89.26±0.64 | 76.06±0.75 | 86.64±0.60 | 90.27±0.25 | 85.48±0.69 | 86.23±0.22 | 82.77±0.55 | 90.22±0.51 | 91.04±0.73 | 80.03±1.03 | 87.85±0.61 | 78.16±0.61 | 82.96±0.24 |
| TTT++ | 74.61±0.80 | 75.90±0.31 | 66.06±1.36 | 86.39±0.26 | 74.54±0.72 | 83.34±0.37 | 88.25±0.62 | 80.82±0.40 | 83.01±0.41 | 78.31±0.33 | 86.55±0.74 | 84.90±1.00 | 78.59±0.55 | 84.82±0.59 | 76.12±0.62 | 80.15±0.21 |

Table 15: Test accuracy of CIFAR-10C-LT with $\rho = 100$ when the model is fine-tuned using Adam optimizer by only one epoch.

| | gaussian_noise | shot_noise | impulse_noise | defocus_blur | glass_blur | motion_blur | zoom_blur | snow | frost | fog | brightness | contrast | elastic_transform | pixelate | jpeg_compression | avg |
|---|---|---|---|---|---|---|---|---|---|---|---|---|---|---|---|---|
| NoAdapt | 43.07±0.40 | 46.75±0.46 | 27.29±0.43 | 90.54±0.51 | 65.85±0.67 | 86.74±0.21 | 93.45±0.32 | 82.38±0.81 | 76.44±0.02 | 62.92±0.62 | 92.75±0.27 | 58.73±0.69 | 82.95±0.64 | 81.19±0.37 | 76.73±0.31 | 71.13±0.17 |
| BNAdapt | 62.40±0.26 | 63.47±0.53 | 53.94±0.39 | 73.06±0.67 | 60.85±0.31 | 70.07±0.22 | 74.44±0.48 | 67.31±0.73 | 69.42±1.12 | 64.22±0.34 | 73.63±0.20 | 71.48±0.37 | 65.19±0.58 | 71.40±0.14 | 62.64±0.49 | 66.90±0.16 |
| BNAdapt+ours | 79.22±0.30 | 79.59±0.46 | 69.52±0.88 | 89.24±0.39 | 76.64±0.15 | 86.79±0.45 | 90.84±0.30 | 84.92±0.61 | 85.69±0.30 | 80.73±0.53 | 90.15±0.41 | 87.21±0.54 | 82.47±0.22 | 87.82±0.39 | 79.05±0.64 | 83.34±0.20 |
| BNAdapt+ours (diag) | 72.00±0.32 | 73.00±0.40 | 64.11±0.73 | 82.17±0.20 | 70.89±0.20 | 79.91±0.17 | 83.35±0.45 | 77.32±0.51 | 78.37±0.60 | 73.89±0.60 | 82.96±0.54 | 80.18±0.46 | 75.10±0.53 | 80.59±0.14 | 72.29±0.73 | 76.41±0.21 |
| BNAdapt+ours (online) | 77.61±0.55 | 78.84±1.50 | 67.85±0.86 | 88.69±0.49 | 75.81±1.15 | 86.13±0.91 | 90.27±0.64 | 83.82±1.74 | 85.49±0.75 | 80.19±0.88 | 89.88±0.34 | 86.29±1.44 | 81.48±1.05 | 87.56±0.70 | 78.70±0.62 | 82.57±0.49 |
| TENT | 66.21±1.46 | 67.12±1.85 | 57.65±1.32 | 76.79±1.20 | 65.14±1.65 | 73.05±1.03 | 78.03±1.00 | 70.04±1.28 | 71.52±2.28 | 68.51±0.55 | 77.48±0.56 | 75.10±0.46 | 69.05±0.49 | 75.19±1.36 | 66.51±1.22 | 70.49±0.43 |
| TENT+ours | 85.03±0.47 | 85.41±0.24 | 77.91±0.38 | 93.19±0.32 | 83.21±0.33 | 91.10±0.31 | 93.65±0.30 | 90.15±0.54 | 90.41±1.04 | 88.10±0.95 | 93.31±0.19 | 91.63±0.42 | 87.39±0.42 | 92.24±0.62 | 85.72±0.54 | 88.56±0.13 |
| TENT+ours (diag) | 81.07±0.53 | 81.22±0.46 | 74.15±0.46 | 89.49±0.32 | 80.29±0.28 | 87.52±0.43 | 89.46±0.82 | 86.17±0.23 | 85.94±1.21 | 83.23±1.03 | 90.12±0.70 | 87.49±0.97 | 83.82±0.80 | 88.37±0.56 | 80.90±1.09 | 84.62±0.43 |
| TENT+ours (online) | 83.47±0.81 | 84.93±0.97 | 76.03±1.30 | 92.67±0.60 | 82.47±0.95 | 90.39±0.53 | 93.12±0.22 | 89.51±1.14 | 89.78±0.81 | 86.91±0.86 | 93.15±0.37 | 90.64±1.06 | 86.27±0.87 | 91.89±0.47 | 84.78±0.36 | 87.73±0.28 |
| PL | 65.02±1.66 | 66.57±1.22 | 56.02±1.05 | 75.81±0.96 | 64.47±0.82 | 72.82±1.37 | 76.79±1.08 | 70.82±0.36 | 71.08±0.87 | 68.54±0.71 | 76.27±0.53 | 73.37±1.40 | 68.02±1.15 | 72.84±1.07 | 65.58±0.95 | 69.63±0.46 |
| PL+ours | 84.53±0.49 | 84.42±0.52 | 75.25±1.63 | 92.95±0.18 | 82.79±0.64 | 90.76±0.54 | 93.31±0.30 | 89.45±0.21 | 89.63±0.80 | 87.14±0.73 | 93.30±0.09 | 91.24±0.64 | 86.88±0.54 | 91.84±0.21 | 84.71±0.18 | 87.88±0.07 |
| PL+ours (diag) | 78.68±0.30 | 78.86±1.53 | 66.45±1.29 | 88.33±0.61 | 78.26±0.66 | 85.83±0.54 | 89.64±0.70 | 84.34±0.78 | 84.67±1.51 | 79.83±1.05 | 89.37±0.38 | 86.73±0.51 | 82.34±1.09 | 87.27±0.67 | 78.97±0.85 | 82.64±0.53 |
| PL+ours (online) | 83.00±0.36 | 83.60±1.11 | 72.74±1.45 | 92.39±0.28 | 81.98±1.09 | 89.95±0.54 | 92.90±0.30 | 88.19±1.71 | 89.24±0.56 | 85.88±0.70 | 92.84±0.64 | 90.39±1.23 | 85.91±1.24 | 91.42±0.49 | 83.96±0.77 | 86.96±0.30 |
| DELTA | 64.36±1.33 | 66.24±4.06 | 56.81±7.08 | 77.13±1.59 | 68.95±0.63 | 69.22±4.49 | 77.70±1.59 | 68.62±1.41 | 69.04±3.08 | 70.14±1.55 | 75.56±0.97 | 73.94±3.02 | 69.67±2.75 | 74.14±1.44 | 66.72±3.57 | 69.88±1.47 |
| DELTA+ours | 85.05±0.50 | 86.06±1.20 | 78.57±0.98 | 93.62±0.34 | 84.21±0.75 | 91.67±0.59 | 94.25±0.48 | 91.02±0.21 | 91.29±0.47 | 89.69±0.75 | 93.95±0.24 | 91.74±0.69 | 87.95±0.74 | 93.09±0.35 | 86.53±0.74 | 89.25±0.33 |
| DELTA+ours (diag) | 82.83±0.62 | 83.79±0.57 | 75.24±0.34 | 92.46±0.58 | 82.86±1.18 | 90.31±0.45 | 91.55±1.66 | 88.96±0.78 | 89.18±1.43 | 87.71±1.19 | 92.31±0.54 | 90.70±1.32 | 85.99±1.24 | 91.05±0.35 | 83.79±1.05 | 87.25±0.55 |
| DELTA+ours (online) | 83.68±0.89 | 85.76±0.46 | 76.05±1.13 | 93.52±0.62 | 83.68±0.70 | 91.34±1.01 | 94.05±0.90 | 90.79±0.23 | 90.87±0.61 | 88.97±0.28 | 93.99±0.19 | 90.93±1.08 | 87.06±1.13 | 92.72±0.10 | 85.86±0.75 | 88.61±0.24 |
| NOTE | 66.76±2.12 | 69.18±2.31 | 62.81±1.99 | 88.80±0.88 | 71.05±3.59 | 83.92±0.97 | 89.25±1.74 | 83.83±1.45 | 82.24±2.04 | 77.24±2.30 | 89.02±0.96 | 89.05±1.10 | 78.88±1.37 | 86.20±0.71 | 74.17±0.70 | 79.49±0.41 |
| NOTE+ours | 79.33±1.24 | 78.89±1.14 | 72.49±1.10 | 91.58±0.37 | 76.38±3.49 | 89.26±0.85 | 91.77±0.38 | 89.14±0.66 | 89.07±0.79 | 85.18±1.43 | 91.74±0.38 | 91.89±0.20 | 83.42±1.04 | 89.25±0.45 | 81.33±1.03 | 85.38±0.29 |
| NOTE+ours (diag) | 75.06±1.76 | 75.98±2.85 | 70.20±1.00 | 91.50±0.34 | 78.38±2.03 | 88.55±0.87 | 91.56±1.16 | 89.16±1.10 | 88.21±1.33 | 82.65±1.69 | 91.78±0.96 | 92.15±0.45 | 84.07±1.24 | 89.59±0.92 | 80.04±1.94 | 85.27±0.13 |
| NOTE+ours (online) | 71.32±1.10 | 73.14±2.28 | 61.29±4.96 | 91.51±0.64 | 75.33±1.83 | 88.12±0.51 | 91.66±0.51 | 88.40±0.60 | 89.10±0.82 | 81.90±1.33 | 91.65±0.35 | 91.86±0.38 | 83.59±0.92 | 89.43±0.38 | 80.77±1.94 | 83.27±0.24 |
| LAME | 65.40±0.42 | 67.17±0.60 | 57.22±0.59 | 76.64±1.05 | 64.55±0.34 | 74.09±0.49 | 78.19±0.29 | 71.06±0.46 | 72.70±1.10 | 66.78±0.58 | 77.17±0.31 | 74.75±0.26 | 68.34±0.58 | 75.03±0.34 | 66.04±0.41 | 70.40±0.25 |
| LAME+ours | 75.29±0.46 | 75.67±0.49 | 65.56±0.58 | 86.11±0.49 | 72.20±0.63 | 84.29±0.31 | 87.95±0.56 | 81.46±0.38 | 81.80±0.73 | 77.46±0.66 | 86.67±0.38 | 83.29±0.55 | 78.30±0.37 | 83.89±0.59 | 74.80±0.65 | 79.66±0.12 |
| LAME+ours (diag) | 70.91±0.14 | 71.70±0.76 | 62.70±0.39 | 80.96±0.60 | 68.92±0.40 | 78.97±0.52 | 82.61±0.35 | 75.67±1.03 | 77.33±0.82 | 72.51±0.59 | 81.61±0.53 | 79.32±0.45 | 73.46±0.26 | 79.62±0.14 | 71.06±0.27 | 75.16±0.23 |
| LAME+ours (online) | 74.86±0.80 | 75.37±1.06 | 64.53±1.14 | 85.37±0.59 | 72.42±0.95 | 83.94±0.33 | 87.34±0.83 | 80.88±1.12 | 81.55±1.34 | 77.07±1.24 | 86.41±0.20 | 83.06±1.23 | 77.48±1.57 | 84.16±1.08 | 74.82±0.39 | 79.28±0.26 |
| ODS | 70.33±1.59 | 71.91±1.21 | 63.35±1.92 | 84.96±0.48 | 69.50±2.11 | 82.58±0.52 | 87.03±1.20 | 81.51±0.58 | 80.70±1.27 | 76.56±1.36 | 85.78±0.86 | 87.72±1.57 | 76.79±0.87 | 83.44±1.08 | 72.58±1.41 | 78.32±0.34 |
| ODS+ours | 72.67±1.87 | 75.19±1.00 | 66.26±1.99 | 88.52±1.38 | 72.62±1.98 | 86.00±0.67 | 89.77±0.54 | 84.97±1.15 | 84.87±1.07 | 79.86±1.11 | 89.73±0.35 | 90.10±0.72 | 80.42±0.50 | 87.13±0.40 | 75.24±0.82 | 81.58±0.17 |
| ODS+ours (diag) | 71.44±1.12 | 74.05±1.90 | 64.22±2.63 | 87.88±1.60 | 72.96±1.74 | 84.33±1.18 | 88.39±0.81 | 83.68±1.27 | 83.09±1.36 | 78.30±0.99 | 88.87±0.74 | 89.92±1.04 | 78.80±1.07 | 86.50±0.83 | 73.47±0.79 | 80.33±0.17 |
| ODS+ours (online) | 72.13±1.95 | 72.91±1.75 | 66.33±2.36 | 89.16±0.42 | 73.40±1.99 | 86.75±0.86 | 90.26±0.96 | 84.98±0.60 | 84.98±0.72 | 79.38±1.17 | 90.04±0.61 | 91.06±0.67 | 81.19±1.68 | 86.68±0.24 | 76.17±1.11 | 81.68±0.12 |
| TTT++ | 64.07±1.24 | 64.55±1.77 | 56.13±0.89 | 74.18±1.13 | 61.82±0.78 | 72.23±0.61 | 76.61±0.53 | 68.92±1.14 | 71.45±1.53 | 66.42±0.55 | 76.38±0.39 | 72.94±0.72 | 66.25±0.80 | 73.41±0.98 | 64.19±0.56 | 68.64±0.37 |

Table 16: Test accuracy of CIFAR-10.1-LT with $\rho = 10$ and $100$ when the model is fine-tuned using Adam optimizer by only one epoch.

|  | $\rho = 10$ | $\rho = 100$ |
|---|---|---|
| NoAdapt | 87.13±0.48 | 86.64±0.97 |
| BNAdapt | 77.37±0.45 | 64.43±0.97 |
| BNAdapt+ours | 85.81±0.65 | 80.64±2.12 |
| BNAdapt+ours (diag) | 83.71±0.34 | 74.29±1.26 |
| BNAdapt+ours (online) | 84.33±0.56 | 80.79±1.97 |
| TENT | 78.23±0.52 | 64.53±1.53 |
| TENT+ours | 86.88±0.78 | 82.32±1.60 |
| TENT+ours (diag) | 85.41±0.48 | 75.97±1.52 |
| TENT+ours (online) | 85.31±0.69 | 82.32±1.88 |
| PL | 78.51±0.38 | 64.38±1.09 |
| PL+ours | 86.02±0.93 | 82.16±1.44 |
| PL+ours (diag) | 83.90±0.86 | 74.95±1.35 |
| PL+ours (online) | 84.79±0.38 | 81.81±1.74 |
| DELTA | 78.57±2.42 | 64.79±1.61 |
| DELTA+ours | 87.19±0.35 | 84.25±1.54 |
| DELTA+ours (diag) | 86.73±0.32 | 79.47±0.37 |
| DELTA+ours (online) | 85.68±0.50 | 84.50±2.33 |
| NOTE | 82.94±1.95 | 81.55±1.59 |
| NOTE+ours | 84.67±0.83 | 87.25±1.01 |
| NOTE+ours (diag) | 85.84±0.45 | 87.65±0.94 |
| NOTE+ours (online) | 81.90±1.54 | 87.09±0.42 |
| LAME | 78.79±1.00 | 67.68±2.58 |
| LAME+ours | 82.60±0.76 | 77.34±1.31 |
| LAME+ours (diag) | 82.11±0.80 | 72.97±2.86 |
| LAME+ours (online) | 82.24±0.75 | 78.61±1.51 |
| ODS | 81.83±1.77 | 77.74±1.86 |
| ODS+ours | 82.85±1.12 | 79.32±1.48 |
| ODS+ours (diag) | 82.60±1.11 | 79.17±1.42 |
| ODS+ours (online) | 83.22±1.41 | 82.01±1.45 |
| TTT++ | 77.74±0.35 | 64.74±0.75 |

Table 17: Test accuracy of CIFAR-100C-LT with $\rho = 10$ when the model is fine-tuned using Adam optimizer by only one epoch.

| | gaussian_noise | shot_noise | impulse_noise | defocus_blur | glass_blur | motion_blur | zoom_blur | snow | frost | fog | brightness | contrast | elastic_transform | pixelate | jpeg_compression | avg |
|---|---|---|---|---|---|---|---|---|---|---|---|---|---|---|---|---|
| NoAdapt | 16.60±0.38 | 17.94±0.06 | 7.09±0.40 | 67.11±0.43 | 26.95±0.54 | 54.98±0.53 | 69.72±0.36 | 49.55±0.67 | 40.88±0.55 | 33.77±0.49 | 62.47±0.59 | 17.50±0.64 | 52.18±0.51 | 51.30±0.47 | 47.56±0.42 | 41.04±0.17 |
| BNAdapt | 52.48±1.05 | 51.77±0.16 | 42.71±0.75 | 67.45±0.85 | 52.75±0.38 | 62.73±0.66 | 70.60±0.15 | 56.88±0.30 | 60.72±0.43 | 52.13±0.75 | 65.91±0.36 | 51.56±0.52 | 57.55±0.31 | 66.34±0.39 | 53.33±0.41 | 58.33±0.15 |
| BNAdapt+ours | 53.60±0.98 | 53.19±0.19 | 43.54±0.76 | 69.19±0.58 | 54.12±0.58 | 64.66±0.86 | 71.99±0.20 | 58.31±0.18 | 62.48±0.40 | 53.41±0.39 | 67.76±0.30 | 63.37±0.75 | 58.89±0.32 | 68.17±0.43 | 54.12±0.41 | 59.79±0.18 |
| BNAdapt+ours (diag) | 53.26±0.81 | 52.47±0.40 | 43.74±0.76 | 68.26±0.54 | 53.68±0.49 | 63.40±0.70 | 71.63±0.35 | 57.76±0.12 | 61.62±0.58 | 52.84±0.68 | 67.23±0.40 | 62.44±0.63 | 58.71±0.22 | 67.47±0.20 | 53.91±0.36 | 59.23±0.12 |
| BNAdapt+ours (online) | 53.31±0.85 | 53.03±0.33 | 43.35±0.72 | 69.12±0.61 | 54.01±0.53 | 64.54±0.87 | 71.94±0.39 | 58.16±0.06 | 62.33±0.62 | 53.42±0.40 | 67.78±0.24 | 63.27±0.61 | 59.11±0.09 | 68.16±0.44 | 54.06±0.38 | 59.71±0.19 |
| TENT | 55.48±0.92 | 54.81±0.78 | 46.63±0.89 | 69.37±0.77 | 55.45±0.89 | 65.62±0.75 | 71.25±0.47 | 60.04±0.40 | 62.65±0.59 | 60.44±0.23 | 68.09±0.65 | 67.14±0.35 | 59.09±0.78 | 67.47±0.85 | 56.30±0.80 | 61.32±0.20 |
| TENT+ours | 56.42±0.75 | 56.77±0.33 | 48.07±0.90 | 70.12±0.41 | 56.78±0.39 | 66.72±0.63 | 71.47±0.39 | 61.67±0.27 | 63.80±0.73 | 61.13±0.21 | 69.28±0.54 | 68.30±0.19 | 60.53±0.84 | 69.06±0.42 | 58.02±0.55 | 62.54±0.28 |
| TENT+ours (diag) | 56.08±1.05 | 55.64±0.63 | 47.05±1.06 | 69.73±0.68 | 55.95±0.61 | 65.83±0.70 | 71.65±0.38 | 60.49±0.45 | 63.66±0.78 | 60.80±0.19 | 68.58±0.76 | 67.74±0.27 | 59.77±0.60 | 68.45±0.34 | 56.95±0.67 | 61.89±0.12 |
| TENT+ours (online) | 56.19±0.60 | 56.59±0.29 | 48.11±0.72 | 70.02±0.41 | 56.66±0.29 | 66.76±0.53 | 71.38±0.44 | 61.69±0.42 | 63.69±0.73 | 60.94±0.11 | 69.18±0.28 | 68.12±0.17 | 60.50±0.66 | 68.91±0.46 | 57.81±0.55 | 62.44±0.26 |
| PL | 53.42±0.90 | 53.14±0.84 | 45.37±0.57 | 68.67±0.73 | 54.23±0.43 | 64.32±1.41 | 70.73±0.09 | 59.46±0.36 | 61.80±0.58 | 57.66±0.87 | 67.40±0.72 | 65.90±1.37 | 58.55±0.55 | 67.12±0.71 | 55.41±0.50 | 60.21±0.24 |
| PL+ours | 51.74±1.22 | 52.17±0.44 | 42.71±0.65 | 67.57±0.54 | 52.33±0.41 | 62.98±0.76 | 70.22±0.30 | 56.79±0.49 | 60.96±0.77 | 53.03±0.88 | 66.00±0.87 | 61.64±1.42 | 57.62±0.68 | 66.65±0.73 | 53.51±0.54 | 58.39±0.22 |
| PL+ours (diag) | 54.32±0.35 | 54.04±0.39 | 46.13±0.24 | 68.63±0.80 | 54.98±0.27 | 65.18±0.79 | 71.03±0.76 | 59.58±0.92 | 62.46±0.68 | 58.40±1.25 | 67.57±0.91 | 66.37±0.46 | 59.02±0.60 | 67.72±0.71 | 56.09±0.62 | 60.77±0.13 |
| PL+ours (online) | 51.93±1.17 | 52.03±0.53 | 42.47±0.56 | 67.41±0.80 | 52.19±0.81 | 62.49±0.90 | 70.04±0.73 | 57.11±1.07 | 60.60±0.92 | 53.16±1.07 | 66.43±1.07 | 61.37±0.89 | 57.17±0.37 | 66.25±0.81 | 53.15±0.89 | 58.25±0.16 |
| DELTA | 55.34±0.88 | 53.65±0.39 | 43.69±0.66 | 69.28±0.87 | 54.03±0.66 | 64.58±0.67 | 70.72±0.82 | 59.15±0.43 | 62.03±1.11 | 57.20±0.43 | 66.96±0.77 | 66.81±0.44 | 57.79±0.18 | 66.25±0.50 | 55.40±0.59 | 60.06±0.16 |
| DELTA+ours | 54.33±0.75 | 55.15±0.60 | 45.23±1.29 | 69.74±0.49 | 55.20±0.68 | 66.02±0.86 | 70.87±0.41 | 60.59±0.75 | 63.01±0.56 | 59.91±0.66 | 68.43±1.08 | 67.34±0.38 | 59.17±0.94 | 68.25±0.61 | 56.40±1.37 | 61.31±0.32 |
| DELTA+ours (diag) | 54.37±1.29 | 54.10±0.47 | 43.61±1.09 | 69.21±1.03 | 54.64±0.21 | 64.82±0.67 | 70.63±0.27 | 59.39±0.98 | 62.82±0.73 | 58.31±0.85 | 67.91±0.68 | 66.74±0.60 | 57.46±0.90 | 66.77±0.35 | 55.90±0.79 | 60.45±0.10 |
| DELTA+ours (online) | 54.14±1.12 | 55.19±0.66 | 45.12±1.33 | 69.78±0.28 | 55.21±0.77 | 65.94±0.76 | 70.88±0.31 | 60.66±0.71 | 63.18±0.71 | 59.84±0.68 | 68.61±1.02 | 67.45±0.27 | 59.05±0.80 | 68.32±0.73 | 56.32±1.40 | 61.31±0.29 |
| NOTE | 29.10±1.50 | 31.13±1.97 | 23.81±0.84 | 47.15±0.70 | 33.62±1.01 | 45.02±1.66 | 49.37±0.73 | 41.29±1.09 | 42.34±1.65 | 38.78±0.84 | 46.48±1.09 | 49.22±1.41 | 37.27±1.02 | 44.85±0.77 | 32.85±1.24 | 39.49±0.29 |
| NOTE+ours | 26.65±1.77 | 28.76±1.61 | 21.43±0.31 | 41.60±1.28 | 30.09±1.34 | 40.67±0.99 | 43.50±0.96 | 36.16±0.59 | 37.82±1.51 | 34.40±0.82 | 39.91±0.50 | 44.94±1.60 | 32.91±0.68 | 39.98±0.84 | 28.53±1.02 | 35.16±0.51 |
| NOTE+ours (diag) | 29.60±1.52 | 30.61±1.32 | 23.70±0.39 | 45.24±0.19 | 32.54±0.48 | 43.49±1.12 | 47.48±0.46 | 39.72±1.02 | 40.23±1.22 | 38.25±0.60 | 45.36±0.98 | 47.95±1.27 | 36.07±2.09 | 42.89±1.72 | 31.75±1.13 | 38.33±0.35 |
| NOTE+ours (online) | 24.91±1.71 | 27.38±0.66 | 20.11±1.07 | 40.29±0.88 | 28.41±1.11 | 38.62±2.72 | 41.02±0.82 | 34.58±0.87 | 35.35±1.00 | 33.19±0.86 | 39.58±0.31 | 42.43±0.68 | 31.71±1.09 | 39.38±0.79 | 27.58±0.90 | 33.64±0.59 |
| LAME | 53.47±0.97 | 52.89±0.17 | 43.47±0.68 | 68.78±0.37 | 53.82±0.48 | 64.08±0.75 | 71.88±0.20 | 58.41±0.26 | 62.13±0.66 | 53.16±0.65 | 67.72±0.50 | 63.01±0.22 | 58.65±0.35 | 67.91±0.27 | 54.06±0.38 | 59.56±0.22 |
| LAME+ours | 54.19±0.81 | 54.08±0.43 | 44.49±0.81 | 69.81±0.23 | 55.07±0.43 | 65.04±0.44 | 71.92±0.34 | 58.98±0.18 | 63.09±0.55 | 54.13±0.80 | 68.46±0.73 | 63.62±0.12 | 59.63±0.38 | 68.27±0.53 | 54.32±0.35 | 60.34±0.19 |
| LAME+ours (diag) | 53.77±0.91 | 53.77±0.32 | 43.96±0.93 | 69.17±0.54 | 54.15±0.21 | 64.28±0.82 | 72.12±0.14 | 58.37±0.15 | 62.63±0.83 | 53.70±0.81 | 67.98±0.34 | 63.22±0.38 | 59.28±0.42 | 68.06±0.60 | 54.08±0.34 | 59.90±0.24 |
| LAME+ours (online) | 54.06±0.84 | 53.90±0.33 | 44.39±0.77 | 69.79±0.22 | 54.99±0.41 | 65.07±0.49 | 71.85±0.29 | 59.02±0.15 | 63.00±0.62 | 54.17±0.80 | 68.35±0.72 | 63.60±0.20 | 59.73±0.41 | 68.12±0.53 | 54.35±0.51 | 60.29±0.19 |
| ODS | 42.51±0.54 | 42.89±0.66 | 34.79±0.99 | 56.74±0.38 | 43.14±0.56 | 55.05±0.82 | 58.49±0.65 | 50.58±0.70 | 52.54±0.38 | 48.05±0.87 | 57.15±0.42 | 60.73±1.02 | 47.44±0.92 | 55.86±0.46 | 42.96±0.30 | 49.93±0.13 |
| ODS+ours | 43.42±0.49 | 43.93±1.02 | 35.62±0.60 | 57.68±0.64 | 44.32±0.15 | 55.80±0.45 | 58.52±0.22 | 50.92±1.21 | 53.89±0.71 | 48.22±1.11 | 57.80±0.81 | 61.80±1.10 | 48.13±0.48 | 56.70±0.41 | 43.61±0.55 | 50.69±0.46 |
| ODS+ours (diag) | 42.95±0.80 | 43.62±0.38 | 35.02±0.59 | 57.03±0.54 | 43.74±0.19 | 54.78±0.62 | 58.22±0.55 | 50.86±1.07 | 53.26±0.56 | 48.26±0.72 | 57.39±0.57 | 61.11±0.87 | 47.83±1.11 | 55.60±0.56 | 42.92±0.14 | 50.17±0.21 |
| ODS+ours (online) | 42.38±0.64 | 43.27±0.78 | 35.01±0.54 | 57.14±0.34 | 44.02±0.31 | 55.26±0.57 | 58.04±0.47 | 50.34±0.82 | 52.92±1.04 | 47.67±1.08 | 57.26±1.33 | 61.20±1.21 | 47.63±0.62 | 56.15±0.98 | 43.05±0.77 | 50.09±0.54 |

Table 18: Test accuracy of CIFAR-100C-LT with $\rho = 100$ when the model is fine-tuned using Adam optimizer by only one epoch.

| | gaussian_noise | shot_noise | impulse_noise | defocus_blur | glass_blur | motion_blur | zoom_blur | snow | frost | fog | brightness | contrast | elastic_transform | pixelate | jpeg_compression | avg |
|---|---|---|---|---|---|---|---|---|---|---|---|---|---|---|---|---|
| NoAdapt | 16.69±0.30 | 18.38±0.63 | 6.84±0.36 | 66.73±0.51 | 26.95±0.64 | 54.45±0.76 | 69.54±0.62 | 48.86±0.58 | 39.85±0.43 | 34.29±0.78 | 61.80±0.62 | 16.41±0.60 | 51.14±1.23 | 51.51±0.20 | 47.25±0.39 | 40.71±0.24 |
| BNAdapt | 49.64±0.68 | 49.50±0.31 | 40.04±0.41 | 63.75±0.79 | 49.98±1.34 | 59.62±1.45 | 66.62±0.64 | 54.36±0.34 | 57.98±0.67 | 50.31±0.96 | 62.48±0.72 | 57.52±0.93 | 54.11±0.48 | 62.76±0.14 | 50.12±0.74 | 55.25±0.13 |
| BNAdapt+ours | 53.19±0.79 | 53.82±0.82 | 43.00±0.62 | 69.09±0.60 | 54.43±1.34 | 64.62±1.39 | 71.87±0.38 | 58.51±0.29 | 62.83±0.30 | 54.27±1.03 | 67.66±1.02 | 62.55±0.76 | 58.08±0.32 | 68.08±1.02 | 54.14±0.73 | 59.74±0.16 |
| BNAdapt+ours (diag) | 52.32±0.61 | 52.23±0.44 | 41.95±0.87 | 66.68±0.27 | 52.76±1.61 | 61.84±0.99 | 69.67±0.67 | 56.55±0.44 | 60.73±0.49 | 52.67±0.98 | 65.23±0.40 | 60.23±1.01 | 56.31±0.30 | 65.66±0.64 | 52.40±0.61 | 57.82±0.31 |
| BNAdapt+ours (online) | 53.23±0.94 | 53.39±1.04 | 43.11±0.69 | 69.03±0.41 | 54.49±1.34 | 64.45±1.30 | 71.81±0.42 | 58.42±0.74 | 62.75±0.49 | 54.15±0.95 | 67.66±1.07 | 62.33±0.98 | 58.00±0.21 | 67.89±0.89 | 54.00±1.06 | 59.65±0.25 |
| TENT | 52.38±1.59 | 52.16±1.16 | 43.43±0.91 | 66.19±0.73 | 53.30±1.20 | 62.65±1.52 | 68.29±0.57 | 56.93±0.67 | 60.48±0.69 | 56.09±0.81 | 64.48±1.55 | 62.66±0.99 | 55.89±0.46 | 64.64±0.92 | 53.52±1.01 | 58.21±0.40 |
| TENT+ours | 56.98±0.80 | 58.28±0.80 | 47.81±0.50 | 72.06±0.63 | 58.49±0.94 | 67.98±1.47 | 73.76±0.62 | 61.90±0.38 | 66.27±1.08 | 61.65±0.36 | 70.89±0.72 | 69.43±0.51 | 61.63±0.81 | 71.00±1.12 | 58.76±0.99 | 63.79±0.22 |
| TENT+ours (diag) | 55.38±0.63 | 54.76±0.90 | 46.05±1.08 | 69.04±0.25 | 55.94±2.19 | 64.91±1.38 | 71.43±0.55 | 59.02±1.23 | 63.42±1.23 | 58.62±0.70 | 67.77±0.79 | 65.49±0.67 | 58.54±0.66 | 67.88±1.46 | 55.77±0.49 | 60.94±0.16 |
| TENT+ours (online) | 57.14±0.75 | 57.74±0.72 | 47.97±1.05 | 72.01±0.83 | 58.61±1.02 | 67.81±1.25 | 73.69±0.37 | 61.73±1.06 | 66.26±1.43 | 61.72±0.34 | 70.61±0.94 | 69.03±0.94 | 61.72±0.78 | 70.72±0.99 | 58.52±1.22 | 63.69±0.24 |
| PL | 50.40±0.91 | 51.43±0.72 | 41.67±0.91 | 65.03±0.50 | 51.65±1.43 | 61.05±0.99 | 67.74±0.59 | 55.51±0.37 | 59.92±0.90 | 52.70±0.91 | 63.54±1.02 | 60.43±0.44 | 54.97±0.57 | 63.47±0.76 | 51.92±0.98 | 56.76±0.16 |
| PL+ours | 52.37±0.48 | 53.20±0.64 | 42.44±0.77 | 67.79±1.15 | 54.29±1.18 | 63.80±1.59 | 70.82±0.45 | 57.93±0.31 | 61.94±0.83 | 53.63±0.74 | 67.74±1.13 | 61.90±0.70 | 57.79±0.46 | 67.90±1.30 | 52.65±1.30 | 59.08±0.25 |
| PL+ours (diag) | 53.19±0.94 | 53.91±0.77 | 44.10±0.94 | 67.95±0.56 | 54.59±1.58 | 63.97±0.81 | 70.76±0.60 | 57.90±0.94 | 62.04±1.06 | 56.72±1.02 | 67.10±0.72 | 64.28±1.14 | 57.97±0.88 | 67.19±0.97 | 54.76±1.10 | 59.76±0.18 |
| PL+ours (online) | 52.75±0.59 | 53.07±0.89 | 42.97±1.20 | 67.96±1.31 | 54.19±1.63 | 63.77±1.57 | 70.95±0.38 | 57.98±0.83 | 61.84±0.99 | 53.38±0.68 | 67.33±1.04 | 61.33±0.65 | 57.68±0.88 | 67.67±1.52 | 52.59±1.21 | 59.03±0.31 |
| DELTA | 50.94±1.02 | 52.66±1.60 | 41.54±0.66 | 66.66±1.15 | 52.83±0.86 | 62.07±2.12 | 69.12±1.04 | 56.75±0.81 | 61.32±0.60 | 56.43±1.36 | 65.20±1.68 | 63.71±1.43 | 56.16±1.21 | 64.44±0.95 | 53.71±1.13 | 58.24±0.28 |
| DELTA+ours | 55.31±0.91 | 58.15±0.53 | 45.40±1.27 | 71.86±0.63 | 57.65±1.60 | 67.35±1.58 | 73.72±1.06 | 61.70±0.91 | 65.90±1.39 | 61.07±1.19 | 71.15±0.93 | 69.13±1.76 | 61.61±1.58 | 70.67±1.44 | 58.51±1.39 | 63.28±0.44 |
| DELTA+ours (diag) | 53.95±0.99 | 54.80±1.10 | 43.83±1.44 | 69.20±0.83 | 54.89±1.49 | 64.18±1.89 | 71.22±0.81 | 58.80±1.60 | 63.49±1.08 | 57.94±1.33 | 68.49±1.44 | 65.19±1.76 | 58.28±1.36 | 67.25±1.17 | 55.63±0.94 | 60.48±0.56 |
| DELTA+ours (online) | 55.48±1.09 | 57.77±0.54 | 45.19±1.52 | 71.81±0.73 | 57.53±1.95 | 67.14±1.51 | 73.84±1.35 | 61.76±0.87 | 65.97±1.43 | 61.16±1.21 | 71.26±1.08 | 68.96±1.76 | 61.57±1.24 | 70.42±1.17 | 58.49±1.56 | 63.22±0.38 |
| NOTE | 31.05±1.62 | 34.60±1.07 | 25.62±1.10 | 51.56±2.03 | 38.37±1.09 | 51.38±0.76 | 55.85±1.29 | 46.41±0.89 | 47.28±0.29 | 43.02±0.65 | 52.85±2.27 | 54.73±1.46 | 42.71±0.96 | 50.73±1.53 | 34.70±3.14 | 44.06±0.80 |
| NOTE+ours | 32.86±1.17 | 35.37±1.45 | 25.34±0.44 | 51.81±1.18 | 39.71±1.91 | 51.21±1.33 | 53.47±0.92 | 46.08±0.39 | 48.06±1.43 | 43.37±0.94 | 52.58±1.54 | 54.36±2.31 | 41.60±1.28 | 49.93±1.18 | 35.79±1.63 | 44.10±0.76 |
| NOTE+ours (diag) | 34.50±2.20 | 36.93±1.45 | 26.56±1.18 | 53.45±1.77 | 41.31±1.32 | 52.99±1.25 | 57.30±1.27 | 49.27±0.79 | 49.25±1.54 | 45.72±1.48 | 56.11±0.75 | 57.40±1.39 | 44.35±1.40 | 51.57±1.10 | 37.98±0.43 | 46.31±0.73 |
| NOTE+ours (online) | 29.45±2.27 | 31.80±1.93 | 23.52±0.53 | 48.01±1.82 | 36.13±2.33 | 47.13±2.15 | 50.34±1.21 | 41.33±2.01 | 43.59±1.10 | 39.56±0.82 | 48.53±1.97 | 50.21±2.52 | 39.36±1.04 | 46.06±0.68 | 32.30±1.34 | 40.49±0.58 |
| LAME | 53.54±0.54 | 53.73±0.56 | 42.97±0.82 | 68.00±0.56 | 53.86±0.92 | 63.33±1.24 | 71.14±0.39 | 58.40±0.41 | 62.35±0.77 | 54.65±0.98 | 67.41±0.62 | 62.59±0.30 | 57.71±0.13 | 67.69±0.59 | 53.46±0.56 | 59.39±0.05 |
| LAME+ours | 56.37±0.83 | 57.30±0.83 | 46.11±0.46 | 71.89±0.67 | 58.17±1.35 | 67.37±1.45 | 74.31±0.90 | 61.46±0.45 | 65.95±0.32 | 57.65±0.64 | 70.81±0.44 | 65.74±0.65 | 61.73±0.73 | 71.09±0.75 | 56.55±1.24 | 62.83±0.75 |
| LAME+ours (diag) | 54.99±0.71 | 54.99±0.42 | 44.49±0.56 | 70.08±0.50 | 55.44±1.49 | 65.00±1.07 | 72.89±0.46 | 59.50±0.56 | 63.49±0.80 | 55.62±1.42 | 68.69±0.44 | 64.29±0.81 | 59.47±0.60 | 69.16±0.59 | 54.76±0.79 | 60.86±0.16 |
| LAME+ours (online) | 56.42±0.84 | 57.04±1.02 | 46.16±0.56 | 71.91±0.77 | 58.11±1.37 | 67.18±1.20 | 74.39±0.64 | 61.41±0.55 | 65.87±0.42 | 57.61±0.67 | 70.72±0.34 | 65.67±0.68 | 61.79±0.54 | 71.08±0.72 | 56.58±1.40 | 62.80±0.15 |
| ODS | 42.89±1.27 | 44.38±0.18 | 35.11±0.43 | 58.05±0.36 | 45.12±0.96 | 56.51±0.84 | 60.27±0.93 | 51.49±0.34 | 54.69±0.78 | 48.16±0.37 | 58.49±0.35 | 62.01±1.03 | 48.91±0.65 | 57.14±1.50 | 43.15±1.14 | 51.09±0.19 |
| ODS+ours | 45.57±1.23 | 47.78±1.43 | 37.02±0.55 | 61.38±0.77 | 48.47±1.77 | 60.44±0.94 | 62.62±0.73 | 54.80±0.62 | 58.24±0.78 | 51.79±0.96 | 62.04±0.53 | 65.27±1.22 | 52.58±0.70 | 60.99±0.93 | 46.07±1.29 | 54.34±0.40 |
| ODS+ours (diag) | 45.19±1.39 | 46.27±0.84 | 36.54±0.72 | 59.87±0.21 | 46.64±1.63 | 58.45±0.78 | 61.51±0.77 | 53.46±1.07 | 56.32±0.86 | 50.27±0.96 | 60.79±0.62 | 63.87±0.99 | 51.06±0.60 | 58.42±0.92 | 44.42±0.57 | 52.87±0.19 |
| ODS+ours (online) | 44.01±1.14 | 46.16±0.68 | 36.19±0.49 | 59.44±0.63 | 46.52±1.51 | 58.21±0.90 | 60.69±0.95 | 53.39±0.62 | 56.45±0.38 | 49.23±1.15 | 60.02±0.77 | 63.97±1.31 | 50.47±0.29 | 58.67±0.21 | 43.97±0.93 | 52.49±0.39 |

Table 19: Test accuracy of PACS when the model is fine-tuned using Adam optimizer by only one epoch.

| | a2c | a2p | a2s | c2a | c2p | c2s | p2a | p2c | p2s | s2a | s2c | s2p | avg |
|---|---|---|---|---|---|---|---|---|---|---|---|---|---|
| NoAdapt | 66.00±0.00 | 97.84±0.00 | 57.27±0.00 | 75.59±0.00 | 90.24±0.00 | 72.21±0.00 | 73.19±0.00 | 39.72±0.00 | 43.93±0.00 | 23.54±0.00 | 50.30±0.00 | 37.96±0.00 | 60.65±0.00 |
| BNAdapt | 75.19±0.19 | 96.99±0.23 | 69.66±0.13 | 81.92±0.45 | 94.69±0.47 | 73.48±0.10 | 77.22±0.08 | 64.61±0.51 | 46.13±0.59 | 59.01±0.54 | 68.68±0.32 | 57.34±0.60 | 72.08±0.11 |
| BNAdapt+ours | 74.09±0.09 | 96.63±0.10 | 72.09±0.50 | 83.79±0.44 | 92.57±0.72 | 74.26±0.34 | 76.04±0.42 | 61.36±0.76 | 53.11±0.28 | 72.01±0.88 | 75.81±0.33 | 72.14±1.05 | 75.33±0.09 |
| BNAdapt+ours (diag) | 72.73±1.04 | 96.77±0.47 | 71.80±0.90 | 79.69±1.91 | 86.54±4.93 | 69.99±1.91 | 66.41±1.04 | 54.74±0.65 | 58.34±0.92 | 59.25±5.38 | 59.47±8.66 | 60.70±6.55 | 69.46±1.84 |
| BNAdapt+ours (online) | 73.02±1.16 | 95.33±1.81 | 71.48±0.28 | 82.32±0.88 | 90.61±4.47 | 74.18±1.39 | 75.56±0.76 | 60.62±1.35 | 52.91±0.34 | 70.41±2.24 | 75.12±0.43 | 70.94±3.73 | 74.38±0.56 |
| TENT | 75.92±3.16 | 97.53±0.46 | 73.24±3.00 | 87.04±0.71 | 96.65±0.26 | 73.27±4.89 | 81.85±0.74 | 74.15±2.13 | 52.63±6.82 | 56.24±1.19 | 70.38±2.03 | | 74.53±0.97 |
| TENT+ours | 77.29±3.79 | 97.83±0.29 | 75.23±1.80 | 88.24±1.51 | 96.60±0.58 | 74.64±3.27 | 80.44±1.32 | 58.81±3.47 | 67.40±5.65 | 84.56±1.04 | 83.13±0.94 | 87.59±11.74 | 80.98±1.19 |
| TENT+ours (diag) | 70.05±5.75 | 96.92±0.86 | 60.02±5.28 | 79.70±1.37 | 83.80±4.18 | 55.06±1.84 | 67.10±1.31 | 46.34±3.34 | 63.44±4.68 | 56.67±3.42 | 46.48±4.70 | 73.89±4.86 | 66.64±1.69 |
| TENT+ours (online) | 70.87±4.25 | 97.26±0.43 | 72.52±1.00 | 85.96±3.68 | 92.60±5.59 | 69.53±6.99 | 79.10±3.84 | 58.67±6.65 | 66.57±5.15 | 83.58±1.94 | 80.22±3.98 | 81.41±14.48 | 78.19±1.71 |
| PL | 76.82±2.17 | 97.07±0.39 | 51.89±5.32 | 84.23±0.98 | 95.28±0.31 | 66.92±2.26 | 76.07±1.26 | 67.17±1.74 | 46.18±0.63 | 57.82±1.27 | 69.93±0.45 | 57.29±1.83 | 70.56±0.75 |
| PL+ours | 77.69±1.73 | 97.57±0.29 | 72.28±2.22 | 88.45±0.88 | 94.54±3.08 | 73.38±3.09 | 78.60±1.26 | 66.39±6.59 | 61.05±2.60 | 83.12±1.58 | 81.37±1.67 | 87.02±4.15 | 80.12±0.49 |
| PL+ours (diag) | 72.01±2.09 | 93.41±1.33 | 57.06±5.10 | 77.38±3.23 | 84.60±4.28 | 48.38±0.67 | 79.39±2.94 | 46.60±3.34 | 49.39±2.94 | 55.57±5.88 | 82.46±3.04 | 80.73±2.39 | 63.64±1.27 |
| PL+ours (online) | 72.71±2.93 | 96.23±2.30 | 69.27±3.20 | 86.43±1.94 | 89.78±7.03 | 71.79±6.89 | 77.31±2.14 | 64.71±4.92 | 55.57±5.88 | 82.46±3.04 | 80.73±2.39 | 87.01±7.49 | 77.83±2.32 |
| DELTA | 82.03±2.83 | 98.37±0.35 | 72.31±3.17 | 90.09±0.43 | 98.05±0.25 | 79.17±2.12 | 82.98±0.35 | 79.24±2.03 | 60.17±6.03 | 58.20±0.53 | 73.94±0.69 | 56.65±0.61 | 77.60±0.87 |
| DELTA+ours | 81.35±1.96 | 98.28±0.26 | 77.53±4.20 | 90.65±0.15 | 97.38±0.74 | 80.46±2.31 | 82.23±1.51 | 66.25±0.42 | 67.10±4.28 | 87.06±0.79 | 84.33±0.76 | 88.94±12.07 | 83.46±1.33 |
| DELTA+ours (diag) | 67.78±5.84 | 97.49±0.54 | 65.63±4.97 | 80.90±2.63 | 83.04±3.70 | 63.45±7.58 | 66.75±0.99 | 51.26±3.01 | 58.34±0.92 | 46.99±9.34 | 36.65±5.38 | 50.91±9.76 | 67.89±1.17 |
| DELTA+ours (online) | 79.49±3.87 | 98.16±0.19 | 80.21±2.40 | 89.37±0.84 | 95.13±3.73 | 78.67±1.77 | 80.35±1.63 | 65.60±0.97 | 67.67±4.35 | 86.21±0.89 | 83.30±3.17 | 83.43±13.95 | 82.30±1.06 |
| NOTE | 75.65±1.22 | 97.19±0.21 | 61.70±6.38 | 78.96±1.65 | 94.27±1.28 | 58.97±5.93 | 76.14±0.97 | 65.65±7.78 | 47.37±4.70 | 46.40±4.23 | 60.32±2.57 | 51.45±1.43 | 67.84±0.56 |
| NOTE+ours | 74.24±3.52 | 96.33±0.62 | 63.08±5.63 | 79.22±1.05 | 94.61±0.21 | 61.10±6.94 | 73.55±1.44 | 61.98±5.53 | 43.13±4.50 | 53.34±2.27 | 65.86±1.46 | 58.62±3.58 | 68.76±0.90 |
| NOTE+ours (diag) | 75.66±3.12 | 95.49±0.71 | 61.70±3.85 | 76.12±4.57 | 86.45±5.71 | 62.95±7.34 | 66.42±2.00 | 59.91±2.38 | 46.99±9.34 | 36.65±5.38 | 50.91±9.76 | 43.11±8.91 | 63.53±3.33 |
| NOTE+ours (online) | 73.57±1.21 | 95.94±0.65 | 62.15±6.19 | 72.69±2.69 | 90.72±1.78 | 57.39±6.99 | 73.23±4.21 | 64.54±4.45 | 41.36±5.74 | 30.21±2.13 | 57.94±3.59 | 48.56±2.59 | 64.03±1.91 |
| LAME | 74.43±0.68 | 97.23±0.16 | 67.42±0.72 | 81.35±0.57 | 95.37±0.45 | 67.40±0.86 | 70.10±0.66 | 47.34±2.67 | 13.81±0.88 | 59.19±0.63 | 65.26±0.37 | 56.29±1.83 | 65.43±0.27 |
| LAME+ours | 76.15±0.23 | 97.38±0.13 | 72.80±0.39 | 85.31±0.36 | 94.21±0.61 | 71.86±0.77 | 76.73±1.13 | 61.03±1.32 | 47.41±1.82 | 75.07±0.97 | 78.37±0.23 | 81.72±1.49 | 76.50±0.11 |
| LAME+ours (diag) | 72.92±1.84 | 95.84±0.65 | 67.83±0.85 | 72.50±2.91 | 78.94±2.73 | 59.21±2.77 | 64.44±1.02 | 50.21±1.22 | 33.84±1.82 | 41.49±2.93 | 50.33±1.31 | | 60.57±0.77 |
| LAME+ours (online) | 74.70±1.92 | 96.08±1.81 | 71.79±1.24 | 84.14±0.44 | 90.58±6.61 | 71.56±1.29 | 75.96±1.77 | 59.92±1.88 | 46.93±1.36 | 73.39±2.07 | 77.74±0.80 | 80.40±8.24 | 75.27±0.62 |
| ODS | 74.47±3.03 | 95.60±0.40 | 63.45±3.42 | 76.23±1.02 | 89.66±1.16 | 62.38±5.18 | 63.72±2.81 | 54.12±1.93 | 31.01±3.02 | 46.17±1.11 | 60.48±2.07 | 53.65±0.45 | 64.25±0.13 |
| ODS+ours | 77.60±2.80 | 96.30±0.10 | 58.07±8.48 | 78.48±0.85 | 92.68±1.72 | 60.30±2.82 | 66.00±3.95 | 58.37±3.63 | 30.45±5.00 | 49.38±0.78 | 63.93±1.81 | 56.41±1.32 | 65.66±0.75 |
| ODS+ours (diag) | 70.97±1.79 | 95.76±0.24 | 58.11±3.67 | 73.96±2.77 | 89.82±0.82 | 59.49±7.23 | 65.80±4.12 | 52.59±1.56 | 30.64±5.68 | 41.83±1.07 | 60.61±0.64 | 49.39±2.05 | 62.41±1.14 |
| ODS+ours (online) | 75.53±1.92 | 96.12±0.67 | 66.05±5.90 | 79.19±0.83 | 92.34±1.08 | 60.11±6.03 | 71.46±2.99 | 58.78±5.43 | 36.46±8.30 | 43.18±1.33 | 62.04±2.32 | 52.01±1.56 | 66.11±0.68 |
| TTT++ | 69.18±1.37 | 96.17±0.46 | 56.03±0.48 | 76.59±1.18 | 93.77±0.08 | 66.63±1.11 | 70.35±0.47 | 57.19±0.70 | 41.60±1.34 | 54.41±0.67 | 65.91±0.82 | 57.47±1.22 | 67.11±0.20 |

Table 20: Test accuracy of digit classification when the model is fine-tuned using Adam optimizer by only one epoch.

|  | mnist | usps | mnistm | avg |
|---|---|---|---|---|
| NoAdapt | 57.68±0.00 | 74.49±0.00 | 43.17±0.00 | 58.45±0.00 |
| BNAdapt | 63.58±0.18 | 74.31±0.38 | 45.40±0.11 | 61.10±0.20 |
| BNAdapt+ours | 64.52±0.86 | 76.15±0.26 | 47.15±0.27 | 62.60±0.35 |
| BNAdapt+ours (diag) | 63.16±0.75 | 74.75±0.44 | 45.85±0.24 | 61.25±0.42 |
| BNAdapt+ours (online) | 64.62±1.04 | 75.62±0.53 | 47.16±0.27 | 62.47±0.58 |
| TENT | 68.43±0.37 | 75.71±0.34 | 46.64±0.09 | 63.59±0.19 |
| TENT+ours | 68.26±0.85 | 77.59±0.40 | 48.70±0.38 | 64.85±0.44 |
| TENT+ours (diag) | 66.79±1.04 | 76.38±0.71 | 47.39±0.52 | 63.52±0.66 |
| TENT+ours (online) | 68.74±1.68 | 77.01±0.35 | 48.75±0.27 | 64.83±0.72 |
| PL | 67.89±0.29 | 75.40±0.39 | 46.45±0.31 | 63.25±0.23 |
| PL+ours | 68.04±0.74 | 77.43±0.49 | 48.34±0.18 | 64.60±0.33 |
| PL+ours (diag) | 66.09±2.00 | 76.06±0.75 | 46.98±0.40 | 63.04±0.97 |
| PL+ours (online) | 68.22±2.35 | 76.88±0.51 | 48.36±0.35 | 64.49±0.97 |
| DELTA | 70.42±0.25 | 76.17±0.54 | 47.33±0.17 | 64.64±0.23 |
| DELTA+ours | 70.24±0.75 | 78.05±0.23 | 49.21±0.39 | 65.83±0.34 |
| DELTA+ours (diag) | 68.28±1.08 | 76.46±0.71 | 47.75±0.58 | 64.16±0.74 |
| DELTA+ours (online) | 70.63±0.67 | 77.54±0.20 | 49.22±0.32 | 65.80±0.29 |
| NOTE | 68.50±0.77 | 76.77±0.30 | 46.29±0.39 | 63.85±0.41 |
| NOTE+ours | 68.12±0.89 | 76.38±0.43 | 47.95±0.25 | 64.15±0.29 |
| NOTE+ours (diag) | 66.40±1.84 | 73.58±0.83 | 47.12±0.35 | 62.37±0.96 |
| NOTE+ours (online) | 65.01±0.93 | 75.30±0.95 | 46.36±0.38 | 62.22±0.65 |
| LAME | 66.82±0.21 | 78.94±0.26 | 46.15±0.06 | 63.97±0.14 |
| LAME+ours | 66.02±0.81 | 80.17±0.76 | 47.42±0.28 | 64.54±0.56 |
| LAME+ours (diag) | 64.51±0.42 | 78.67±0.44 | 46.62±0.33 | 63.27±0.22 |
| LAME+ours (online) | 66.17±1.23 | 79.87±0.96 | 47.40±0.26 | 64.48±0.70 |
| ODS | 69.94±0.58 | 78.75±0.19 | 47.80±0.45 | 65.50±0.26 |
| ODS+ours | 70.05±0.66 | 79.36±0.10 | 48.02±0.10 | 65.81±0.23 |
| ODS+ours (diag) | 70.06±0.90 | 79.36±0.16 | 47.66±0.31 | 65.69±0.33 |
| ODS+ours (online) | 69.89±0.82 | 79.20±0.17 | 47.91±0.20 | 65.66±0.29 |
| TTT++ | 62.85±0.30 | 74.25±0.31 | 45.47±0.10 | 60.86±0.06 |

Table 21: Test accuracy of ImageNet-C online labels distribution shift with $\alpha = 1000$ when the model is fine-tuned using SGD optimizer by only one epoch.

|  | gaussian_noise | shot_noise | impulse_noise | defocus_blur | glass_blur | motion_blur | zoom_blur | snow | frost | fog | brightness | contrast | elastic_transform | pixelate | jpeg_compression | avg |
|---|---|---|---|---|---|---|---|---|---|---|---|---|---|---|---|---|
| NoAdapt | 2.96±0.08 | 3.68±0.06 | 2.62±0.05 | 17.82±0.07 | 9.71±0.12 | 14.71±0.07 | 22.49±0.28 | 16.57±0.11 | 23.08±0.11 | 24.05±0.04 | 59.13±0.17 | 5.33±0.03 | 16.66±0.04 | 20.80±0.19 | 32.63±0.15 | 18.15±0.06 |
| BNAdapt | 10.34±0.12 | 10.68±0.07 | 10.64±0.08 | 9.22±0.10 | 9.36±0.18 | 16.12±0.06 | 23.66±0.25 | 21.71±0.03 | 21.26±0.12 | 30.06±0.08 | 41.90±0.16 | 10.47±0.10 | 27.19±0.35 | 30.19±0.19 | 24.96±0.09 | 19.85±0.10 |
| BNAdapt+ours | 12.52±0.38 | 12.88±0.27 | 12.99±0.41 | 11.73±0.42 | 11.13±0.50 | 20.92±0.84 | 30.41±1.06 | 27.88±0.89 | 27.01±0.68 | 39.15±1.04 | 53.17±1.30 | 13.15±0.32 | 33.67±1.05 | 38.77±1.20 | 32.40±0.98 | 25.18±0.75 |
| TENT | 14.54±0.38 | 15.21±0.09 | 16.06±0.14 | 12.39±0.22 | 11.26±0.57 | 18.26±0.40 | 27.43±0.32 | 26.47±0.24 | 19.50±0.23 | 34.11±0.18 | 41.31±0.18 | 4.25±0.22 | 31.60±0.23 | 34.39±0.27 | 30.49±0.19 | 22.49±0.15 |
| TENT+ours | 13.05±0.56 | 15.48±1.22 | 15.83±1.54 | 13.00±0.85 | 10.51±0.21 | 19.67±1.75 | 34.33±1.13 | 28.52±1.66 | 21.06±1.39 | 43.42±1.12 | 53.25±1.48 | 4.96±0.52 | 38.35±1.10 | 43.15±1.18 | 38.11±1.16 | 26.18±0.88 |
| SAR | 18.60±1.61 | 18.64±0.44 | 20.33±0.14 | 16.23±0.66 | 16.20±0.59 | 25.65±0.23 | 30.10±0.35 | 29.78±0.27 | 26.60±0.18 | 36.05±0.13 | 42.76±0.18 | 13.01±3.94 | 34.03±0.14 | 36.30±0.26 | 32.63±0.24 | 26.46±0.39 |
| SAR+ours | 24.18±0.87 | 22.60±0.60 | 25.22±0.94 | 18.94±0.60 | 19.16±0.48 | 32.36±1.16 | 37.93±1.24 | 38.12±1.27 | 33.37±0.74 | 45.98±1.21 | 53.60±1.31 | 5.70±1.19 | 42.79±1.28 | 45.84±1.42 | 41.55±1.33 | 32.49±0.83 |

Table 22: Test accuracy of ImageNet-C online labels distribution shift with $\alpha = 2000$ when the model is fine-tuned using SGD optimizer by only one epoch.

|  | gaussian_noise | shot_noise | impulse_noise | defocus_blur | glass_blur | motion_blur | zoom_blur | snow | frost | fog | brightness | contrast | elastic_transform | pixelate | jpeg_compression | avg |
|---|---|---|---|---|---|---|---|---|---|---|---|---|---|---|---|---|
| NoAdapt | 2.98±0.04 | 3.68±0.04 | 2.62±0.05 | 17.90±0.03 | 9.70±0.08 | 14.80±0.06 | 22.52±0.23 | 16.53±0.07 | 23.01±0.09 | 24.00±0.07 | 59.12±0.08 | 5.42±0.02 | 16.54±0.18 | 20.90±0.03 | 32.72±0.06 | 18.16±0.01 |
| BNAdapt | 7.54±0.07 | 7.72±0.15 | 7.84±0.07 | 6.54±0.01 | 6.60±0.14 | 11.47±0.04 | 16.53±0.09 | 15.52±0.06 | 15.39±0.14 | 21.30±0.15 | 29.78±0.21 | 7.45±0.05 | 18.97±0.13 | 21.17±0.16 | 17.78±0.06 | 14.11±0.04 |
| BNAdapt+ours | 10.16±0.46 | 10.50±0.31 | 10.75±0.47 | 9.41±0.51 | 8.61±0.47 | 16.86±0.73 | 24.34±0.97 | 23.00±0.96 | 22.64±0.93 | 32.08±1.31 | 43.90±1.50 | 10.63±0.41 | 26.66±0.82 | 31.15±1.19 | 26.56±1.12 | 20.48±0.80 |
| TENT | 8.25±0.18 | 8.59±0.22 | 9.51±0.20 | 6.84±0.15 | 5.62±0.22 | 7.98±0.39 | 17.39±0.20 | 12.35±0.56 | 10.98±0.20 | 22.56±0.13 | 28.11±0.23 | 2.25±0.11 | 20.15±0.18 | 22.49±0.18 | 19.76±0.11 | 13.52±0.11 |
| TENT+ours | 8.70±0.58 | 9.75±0.63 | 10.30±1.07 | 8.55±0.55 | 6.17±0.62 | 10.75±1.31 | 25.12±1.15 | 15.20±2.20 | 14.16±1.23 | 33.76±1.38 | 42.68±1.75 | 3.18±0.43 | 27.77±0.91 | 32.90±1.49 | 28.70±1.05 | 18.51±0.98 |
| SAR | 12.74±0.37 | 9.27±1.14 | 12.95±0.82 | 9.39±1.18 | 8.90±1.31 | 16.57±0.75 | 20.53±0.17 | 20.47±0.15 | 18.11±0.54 | 25.22±0.24 | 30.16±0.24 | 4.74±1.29 | 23.49±0.20 | 25.23±0.18 | 22.67±0.17 | 17.36±0.13 |
| SAR+ours | 16.05±1.45 | 12.85±0.97 | 17.67±0.45 | 10.60±0.95 | 7.25±0.86 | 22.03±2.11 | 28.81±1.16 | 29.35±1.06 | 21.33±1.41 | 36.73±1.32 | 43.44±1.53 | 2.67±0.49 | 33.30±1.17 | 36.02±1.29 | 32.65±1.39 | 23.38±0.81 |

Table 23: Test accuracy of ImageNet-C online labels distribution shift with $\alpha = 5000$ when the model is fine-tuned using SGD optimizer by only one epoch.

| | gaussian_noise | shot_noise | impulse_noise | defocus_blur | glass_blur | motion_blur | zoom_blur | snow | frost | fog | brightness | contrast | elastic_transform | pixelate | jpeg_compression | avg |
|---|---|---|---|---|---|---|---|---|---|---|---|---|---|---|---|---|
| NoAdapt | 2.96±0.06 | 3.70±0.03 | 2.60±0.06 | 17.95±0.11 | 9.69±0.07 | 14.79±0.13 | 22.56±0.14 | 16.56±0.03 | 23.01±0.08 | 23.98±0.08 | 59.17±0.24 | 5.34±0.06 | 16.59±0.03 | 20.85±0.06 | 32.68±0.04 | 18.16±0.04 |
| BNAdapt | 4.72±0.02 | 4.88±0.05 | 4.81±0.09 | 3.91±0.07 | 3.94±0.06 | 6.81±0.11 | 9.78±0.09 | 9.48±0.11 | 9.64±0.05 | 12.73±0.04 | 17.79±0.12 | 4.58±0.04 | 11.15±0.13 | 12.32±0.07 | 10.64±0.07 | 8.48±0.06 |
| BNAdapt+ours | 7.48±0.42 | 7.77±0.40 | 7.80±0.50 | 6.71±0.49 | 5.90±0.30 | 12.06±0.83 | 17.45±0.99 | 16.72±0.99 | 16.96±0.92 | 23.63±1.05 | 31.98±1.39 | 7.72±0.48 | 18.54±0.72 | 22.16±1.19 | 19.37±1.11 | 14.82±0.78 |
| TENT | 4.04±0.16 | 4.11±0.05 | 4.40±0.14 | 3.04±0.10 | 2.50±0.16 | 3.53±0.18 | 8.67±0.10 | 4.74±0.12 | 5.07±0.15 | 11.72±0.27 | 15.35±0.11 | 1.01±0.06 | 9.75±0.17 | 11.29±0.15 | 9.94±0.13 | 6.61±0.08 |
| TENT+ours | 5.05±0.45 | 5.39±0.58 | 5.59±0.23 | 4.50±0.58 | 3.55±0.28 | 5.75±0.54 | 15.32±1.03 | 7.88±1.02 | 7.45±1.16 | 21.96±1.14 | 29.18±1.64 | 1.62±0.24 | 16.27±0.71 | 20.24±1.19 | 17.82±1.31 | 11.17±0.77 |
| SAR | 6.08±1.39 | 3.98±0.34 | 6.12±0.78 | 3.21±0.83 | 3.36±0.42 | 7.35±1.14 | 11.18±0.10 | 11.50±0.10 | 9.40±0.64 | 14.59±0.08 | 17.64±0.11 | 1.81±0.31 | 13.09±0.08 | 14.25±0.15 | 12.72±0.14 | 9.09±0.18 |
| SAR+ours | 6.95±1.76 | 4.00±0.44 | 6.73±2.08 | 2.16±0.62 | 2.81±0.30 | 5.87±1.50 | 17.64±0.56 | 15.54±2.25 | 8.85±1.62 | 25.22±1.11 | 30.56±1.32 | 0.81±0.12 | 21.00±0.77 | 23.98±1.02 | 21.32±1.03 | 12.90±0.77 |

