# OpenReview forum: "Distribution Shift-Aware Prediction Refinement for Test-Time Adaptation"
_ICLR.cc/2024/Conference — Submitted to ICLR 2024_

### Official Review · Reviewer_D6tt · 2023-10-31

**Soundness:** 3 good
**Presentation:** 3 good
**Contribution:** 2 fair
**Rating:** 5
**Confidence:** 5

**Summary:**

This paper focuses on a challenging test-time adaptation setup where both the label shift and covariate shift exist in the testing phase and proposes a novel TTA method, named Distribution shift-Aware prediction Refinement for Test-time adaptation (DART).
In particular, DART refines the predictions made by the trained classifiers by focusing on class-wise confusion patterns, introducing a learnable module to map the class distribution onto the class-to-class matrix.
When combined with many TTA methods like TENT and BNAdapt, DART helps increase the accuracy under both covariate and label distribution shifts at test time.

**Strengths:**

- this paper is well-written and easy to follow

- the proposed method is simple yet effective and the key idea sounds interesting

- the results on many datasets are impressive

**Weaknesses:**

- the results are limited to sever label shifts, while the effectiveness of the proposed method under only covariate shift is not well studied

- could the proposed method be extended to source-free domain adaptation like SHOT (Liang et al., ICML 2020) (more epochs)? More results are welcome to verify the versatility of the proposed method

- concerning the mapping module g_\phi, is the network design (like two-layer MLP or the hidden dimensional) sensitive?

- how about the sensitivity of the batch size $B$ in the proposed method?

**Questions:**

see the weakness above

---

> ### Author Response · Authors · 2023-11-16
> **Response to Reviewer D6tt (1/2)**
>
> Thank you for the positive feedback and summary of our work. We hope the responses to the weaknesses below resolve your concerns.
>
> >(W1) the results are limited to severe label shifts, while the effectiveness of the proposed method under only covariate shift is not well studied
>
> We focus on the problem setup where **both** covariate and label distribution shifts occur during the test time. In Section 2, we first examine the impact of the label distribution shift on BNAdapt, which is known to effectively address covariate shifts, thus serving as the foundation for entropy minimization-based TTA methods including TENT. We observe that BNAdapt suffers from a significant performance degradation under the label distribution shift, since the updated BN statistics follow the test label distribution, which causes bias in the classifier. In Section 3, we propose a novel refinement scheme for the predictions generated by the BN-adapted classifiers under both the covariate and label distribution shifts. This refinement mainly corrects the misclassification generated from the label distribution shifts. Thus, without the label distribution shift, our module does not have a big impact in adjusting the classifier predictions. We actually observe this from the oracle analysis in Table 1 for CIFAR-10C-LT with $\rho=1$ (balanced case), where a limited gain of 0.3% is achieved even with the oracle method (using true test labels to find the oracle square matrix $T$). Therefore, DART, which uses the same prediction modification scheme, can only achieve limited gains when there is only covariate shift. We report the experimental results of DART-applied TTA methods on balanced CIFAR-10C in Appendix E.5. We’d like to point that similar trends of limited gain for balanced datasets were also reported for ODS (Zhou et al., ICML’23), another attempt to alleviate test-time class distribution shifts.
>
> >(W2) could the proposed method be extended to source-free domain adaptation like SHOT (Liang et al., ICML 2020) (more epochs)? More results are welcome to verify the versatility of the proposed method
>
> The major drawback of the entropy minimization-based methods is the error accumulation in training due to the utilization of inaccurate pseudo-labels. Therefore, improving pseudo-label accuracy before test-time training, as in DART, can significantly improve performance. We expect that our method can also be utilized as a plug-in method for source-free domain adaptation (SFDA) methods that use pseudo-labels for training like SHOT (Liang et al., ICML’20) and NRC (Yang et al., NeurIPS’21). For a fair comparison, we fine-tune BN layers in pre-trained models by Adam optimizer with learning rate 1e-3 for those SFDA methods. To test the scalability of DART to SFDA methods under test-time label distribution shift, we conduct experiments on CIFAR-10C-LT of $\rho=100$ in the offline manner (multiple epochs). SHOT and NRC utilize the prediction diversity loss  ($\mathcal{L}_{\text{div}} = KL(\bar{p}||u) $, where $\bar{p}$ is the averaged pseudo label distribution and $u$ is a uniform label distribution) to prevent the adapted classifier from predicting all data into some classes. Since the test label distributions are imbalanced, the loss may cause prediction degradation. Thus, we report the experimental results in Table R1 for both cases when this loss is used and not used.
> While TENT/SHOT/NRC showed almost similar test accuracy of 85.83/85.67/85.97% on balanced CIFAR-10C (i.e., $\rho=1$), all the methods showed degraded performance on CIFAR-10C-LT of $\rho=100$. Especially, SHOT showed poor performance on CIFAR-10C-LT due to inaccurate self-supervised pseudo labels caused by the severe class imbalance of test data. However, we can observe that DART consistently improves the performance of adapted classifiers for all the SFDA methods.
>
> **R1. Comparison under offline setup, which adapts the pre-trained models by multiple epochs. We report average accuracy (%) on CIFAR-10C-LT of $\rho=100$.**
> |                       | w/ $\mathcal{L}_{\text{div}}$ | w/o $\mathcal{L}_{\text{div}}$  |
> |-----------------------|------------------------------|--------------------------------|
> | BNAdapt               | -                            | 66.9                           |
> | BNAdapt + DART (ours) | -                            | 83.34                          |
> | TENT                  | -                            | 74.58                          |
> | TENT+DART (ours)      | -                            | 90.21                          |
> | SHOT                  | 66.39                        | 70.1                           |
> | SHOT+DART (ours)      | 65.86                        | 72.26                          |
> | NRC                   | 68.6                         | 81.52                          |
> | NRC+DART (ours)       | 80.45                        | 87.19                          |

---

> ### Author Response · Authors · 2023-11-16
> **Response to Reviewer D6tt (2/2)**
>
> > (W3) concerning the mapping module $g_\phi$ is the network design (like two-layer MLP or the hidden dimensional) sensitive?
>
> We conducted experiments to check the sensitivity of DART over the hidden dimension $d_h$ and number of layers of $g_\phi$, and the results are summarized on CIFAR-10C-LT of $\rho=100$ in Table R2. We can observe that DART is robust against the change in the $g_\phi$ structure.
>
> **R2. Sensitivity analysis about the network design of $g_\phi$.**
>
> | |2-layer MLP (used)| | | |3-layer MLP| | | |
> |:----|:----:|:----:|:----:|:----:|:----:|:----:|:----:|:----:|
> | |$d_h = 250$|$d_h = 500$|$d_h = 1000$ (used)|$d_h = 2000$|$d_h = 250$|$d_h = 500$|$d_h = 1000$|$d_h = 2000$|
> |NoAdapt|71.13|71.13|71.13|71.13|71.13|71.13|71.13|71.13|
> |BNAdapt|66.90|66.90|66.90|66.90|66.90|66.90|66.90|66.90 |
> |BNAdapt+DART (ours)|80.6|82.17|83.34|83.83|83.83|84.27|84.78|84.97|
> |TENT|70.49|70.49|70.49|70.49|70.49|70.49|70.49|70.49|
> |TENT+DART (ours)|87.46|88.23|88.56|88.65|88.81|88.67|88.6|88.09|
>
>
> > (W4) how about the sensitivity of the batch size $B$ in the proposed method?
>
> We conducted experiments to check the sensitivity of DART over $B$, the test batch size, and the results are summarized in Table R3.  We can observe that DART is robust against the change in $B$.
>
> **R3. Sensitivity analysis about the test batch size B. We report average accuracy (%) on CIFAR-10C-LT of $\rho=100$.**
>
> |                     | $B=32$  | $B=64$  | $B=128$ | $B=256$  |
> |---------------------|-------|-------|-------|--------|
> | NoAdapt             | 71.13 | 71.13 | 71.13 | 71.13  |
> | BNAdapt             | 65.48 | 66.15 | 66.68 | 66.99  |
> | BNAdapt+DART (ours) | 81.7  | 82.65 | 83.17 | 83.51  |
> | TENT                | 71.89 | 71.98 | 71.48 | 69.97  |
> | TENT+DART (ours)    | 85.63 | 88.2  | 88.86 | 88.3   |

---

> ### Comment · Reviewer_D6tt · 2023-11-22
> **two major concerns still remain**
>
> Thank the authors for the responses. However, two major concerns remain, leading to a final score in the range of 5-6. (The main idea is interesting but the proposed method is not solid.)
>
> - As acknowledged by the authors, the proposed method seems to fail in common covariate shifts with the existence of labels shift, heavily limiting its application in real-world TTA problems.
>
> - As stated in the original review, the reviewer is curious about the performance under small batch sizes. However, only results under larger batch sizes are provided in the rebuttal.
>
> Besides, the results of source-free domain adaptation are also not persuasive. Without the strong long-tail distribution shift, would the proposed method work? Since the label distribution in common domain adaptation (DA) datasets is also not balanced, I strongly suggest the authors validate the effectiveness of the proposed method on standard DA benchmarks like OfficeHome and DomainNet.
>
> If the authors could provide **evidence about the effectiveness of the proposed method beyond long-tail datasets**, I would increase my score.

---

> > ### Author Response · Authors · 2023-11-22
> > **Response to the additional comments**
> >
> > We appreciate the reviewer for the further comments.
> >
> > > As acknowledged by the authors, the proposed method seems to fail in common covariate shifts with the existence of labels shift, heavily limiting its application in real-world TTA problems.
> >
> > We’d like to clarify that we tested our method not only on long-tailed distributions but also for PACS and digit classification (SVHN->MNIST/MNIST-M/USPS) without imposing any label distribution shifts on the original datasets and confirmed the efficacy of our method in these various setups as summarized in Table 2. In particular, PACS and digit classification (SVHN->MNIST/MNIST-M/USPS) are widely used benchmarks in common domain adaptation (DA), and the class distributions are neither balanced nor severely long-tailed as seen in Fig. 2-(a). Our experimental results in Table 2 clearly demonstrate that DART works as a valuable plug-in method that can be combined with other TTA methods to resolve the challenges in real-world TTAs where not only covariate shifts but also any label distribution shift (not necessarily long-tailed) are encountered. Additionally, during the discussion period, we confirmed that DART can be combined with SFDA methods such as SHOT and NRC.
> >
> > > As stated in the original review, the reviewer is curious about the performance under small batch sizes. However, only results under larger batch sizes are provided in the rebuttal.
> >
> > In the paper, we conducted experiments on CIFAR-10C-LT with a batch size of 200. Therefore, to check the sensitivity on the test batch size, we additionally conducted experiments on CIFAR-10C-LT with the smaller (32,64, and 128) and larger batch sizes (256) during the discussion period.
> >
> >
> > > Evidence about the effectiveness of the proposed method beyond long-tail datasets
> >
> > We’d like to again emphasize that we tested our method not only on long-tailed distributions but also for PACS and digit classification (SVHN->MNIST/MNIST-M/USPS) without imposing any label distribution shifts on the original datasets and confirmed the efficacy of our method in these various setups as summarized in Table 2.
> >
> > The data generation/collection process of the digit classification and PACS benchmarks is different across domains, resulting in differently imbalanced class distribution (not long-tailed), as illustrated in Figure 2 and 5. In PACS, we conduct experiments across 12 different scenarios, each using the four domains (photo, art, cartoon, and sketch) as training and test domains, respectively. Similarly, in digit classification, we test the robustness of classifiers trained on SVHN on MNIST, MNIST-M, and USPS.
> >
> > Hope these answers resolve the reviewer's concern and we will do our best to conduct further experiments using OfficeHome or DomainNet if time allows as the reviewer suggested.

---

> > > ### Comment · Reviewer_D6tt · 2023-11-22
> > > **Could the proposed method work really work**
> > >
> > > Thanks for the replies.
> > >
> > > 1. Is it true that the proposed method does not work for balanced datasets like CIFAR10-C and CIFAR100-C in TTA methods?
> > >
> > > 2. For domain adaptation and generalization tasks, both PACS and digits are simple, so why not try naturally-imbalanced datasets like Office and OfficeHome, I believe they are not large-scale datasets.
> > >
> > > 3. Regarding the batch size, 32 is relatively larger, the authors are suggested to try smaller batch sizes like 4, and 8 instead of 32 and 64. I think many previous TTA methods in the literature have already shown similar studies in their experiments. Even some TTA methods work under a single instance (batch size = 1).

---

> > > > ### Author Response · Authors · 2023-11-22
> > > > **Response to the questions**
> > > >
> > > > We thank the reviewer for further questions.
> > > >
> > > >
> > > > > Is it true that the proposed method does not work for balanced datasets like CIFAR10-C and CIFAR100-C in TTA methods?
> > > >
> > > >
> > > > The proposed method does effectively fix the performance degradation of BNAdpat under label-distribution shifts (not necessarily long-tailed), but when there exists no label distribution shift like in CIFAR-10/100-C, the maximum gain our method can bring is limited by that of the Oracle in Table 1 of the manuscript, e.g., 0.3\% gain for CIFAR-10-C.
> > > >
> > > >
> > > > > For domain adaptation and generalization tasks, both PACS and digits are simple, so why not try naturally-imbalanced datasets like Office and OfficeHome.
> > > >
> > > >
> > > > We’d like to mention that PACS is not a simple dataset: it contains 9991 images from 7 classes (Dog, Elephant, Giraffe, Guitar, Horse, House, and Person) and 4 domains (Art, painting, Cartoon, Photo, and Sketch) with image size of 224$\times$ 224. Officehome contains around 15,500 images from 65 different classes and 4 domains (artistic, clip art, product, and real-world). Thus, the complexity of Officehome may come from the increased number of classes. However, we already checked the scalability of our algorithm for CIFAR-100 and ImageNet-1k, as explained in “DART on large-scale datasets” paragraph of Sec 4.2 Experimental results. We will update the reviewer after we conduct further experiments on OfficeHome.
> > > >
> > > >
> > > > > Regarding the batch size, 32 is relatively larger, the authors are suggested to try smaller batch sizes like 4, and 8 instead of 32 and 64. I think many previous TTA methods in the literature have already shown similar studies in their experiments. Even some TTA methods work under a single instance (batch size = 1).
> > > >
> > > >
> > > > We further conducted the small batch experiments as the reviewer requested and summarized the results below. We can observe consistent gains even for smaller batch sizes.
> > > > |              | $B=4$   |$B=8$   | $B=16$  |
> > > > |--------------|-------|-------|-------|
> > > > | NoAdapt      | 71.13 | 71.13 | 71.13 |
> > > > | BNAdapt      | 58.39 | 61.17 | 63.83 |
> > > > | BNAdapt+ours | 70.40 | 76.75 | 80.17 |
> > > > | TENT         | 46.42 | 61.22 | 70.02 |
> > > > | TENT+ours    | 51.94 | 67.63 | 79.56 |

---

> ### Comment · Reviewer_D6tt · 2023-11-22
>
> Thanks for your reply.
>
> 1. The Oracle is the best built-in variant of your method, which does not mean that the negligible gains are acceptable. It is disappointing to find even the Oracle variant of the proposed method cannot address covariate shifts.
>
> 2. Even for the ImageNet testbed, a high imbalance ratio is required in Table 3, $\alpha>1000$. OfficeHome is much not larger, each domain only has several thousand images, which makes it easy for a quick comparison **under naturally imbalanced label shift**. By the way, is a dataset consisting of only 7 classes hard?
>
> 3. For the sensitivity of the batch size, would you like to provide the results except for long-tail datasets, e.g., CIFAR10-C, CIFAR100-C, or Digits?
>
> Generally speaking, the proposed method is limited to highly imbalanced datasets. It would be much more suitable if the authors studied the test time prior shift and compared the proposed method with counterparts [1-2] in that field.
>
> [1]. Beyond invariance: Test-time label-shift adaptation for distributions with" spurious" correlations
>
> [2]. Self-supervised aggregation of diverse experts for test-agnostic long-tailed recognition

---

> > ### Author Response · Authors · 2023-11-23
> > **Response to the comments (1/2)**
> >
> > We thank the reviewer for further questions.
> >
> > > The Oracle is the best built-in variant of your method, which does not mean that the negligible gains are acceptable. It is disappointing to find even the Oracle variant of the proposed method cannot address covariate shifts.
> >
> > The results in Table 1 where we compared the performances of NoAdapt, BNAdapt, and Oracle (the best built-in variant of our method) for CIFAR-10C-LT with $\rho=1,10,100$ show that BNAdapt effectively addresses the covariate shifts when there is no label shift ($\rho=1$). However, it suffers from a significant performance degradation under the label distribution shift. We then show that the oracle method, which simply multiplies the optimized class-dimensional square matrix to the logits, reverses the performance degradation at the BN-adapted classifiers caused by the label distribution shift. This observation has motivated the particular design of our method to refine the predictions from the BNAdapted classifier, similar to the oracle method. Since BN adaptation is an effective method to address covariate shift and most TTA methods apply the BN adaptation, our method can be used as a plug-in method for most TTA methods to address the performance degradations caused by both covariate shift and label-distribution shift.
> >
> > > Even for the ImageNet testbed, a high imbalance ratio is required in Table 3, $\alpha>1000$. OfficeHome is much not larger, each domain only has several thousand images, which makes it easy for a quick comparison under naturally imbalanced label shift.
> >
> > We conducted the suggested experiments on OfficeHome and summarized the results below. We used ResNet-50 which was released by the authors of TTAB (Zhao et al., ICML’23). For the intermediate time training, we followed the implementation details of ImageNet-C in Section 4.2 "DART on large-scale datasets" in the paper. For the test time, we set the test batch size to 32. Due to time constraints, we focused our testing of DART only on “Art->Clipart (a2c), product (a2p), and real-world (a2r)” in the OfficeHome dataset. The full results for other experiments will be included in the final paper once completed. Through these experiments, we could observe that DART effectively refines the output of the BNAdapted classifier under natural test-time distribution shift, leading to enhanced prediction/pseudo label accuracy. These refined pseudo labels generated by DART are expected to contribute to the improved performance of the subsequent TTA methods, similar to the outcomes seen in the other experiments presented in Table 2-3.
> > |              | a2c           | a2p           | a2r           |
> > |--------------|---------------|---------------|---------------|
> > | BNAdapt      | 47.03         | 60.85         | 69.98         |
> > | BNAdapt+ours | 47.26 (+0.23) | 62.87 (+2.02) | 70.85 (+0.87) |
> >
> > > For the sensitivity of the batch size, would you like to provide the results except for long-tail datasets, e.g., CIFAR10-C, CIFAR100-C, or Digits?
> >
> > We further conducted the small batch experiments on PACS, whose test batch size was originally set to 32 in the paper, and summarized the results below. We can observe consistent gains even for smaller batch sizes.
> >
> > |              | 4     | 8     | 16    | 32 (used) |
> > |--------------|-------|-------|-------|-----------|
> > | NoAdapt      | 60.65 | 60.65 | 60.65 | 60.65     |
> > | BNAdapt      | 64.10 | 68.59 | 70.70 | 72.08     |
> > | BNAdapt+ours | 67.79 | 72.04 | 74.05 | 75.33     |
> > | TENT         | 39.91 | 58.73 | 69.51 | 74.53     |
> > | TENT+ours    | 51.37 | 69.21 | 77.14 | 80.98     |

---

> > ### Author Response · Authors · 2023-11-23
> > **Response to comments (2/2)**
> >
> > > Generally speaking, the proposed method is limited to highly imbalanced datasets. It would be much more suitable if the authors studied the test time prior shift and compared the proposed method with counterparts [1-2] in that field.
> >
> >
> > We thank the reviewer for providing relevant papers. As pointed out by the reviewer, these papers primarily focus on the test time prior shift. The main difference of our method from these works is that our method does not directly tackle the test time prior shift but deals with the combination of covariate shift and label distribution shift. In particular, Logit Adjustment (LA)-like methods (including SADE [2]) apply an offset to each logit based on the corresponding class ratio between the training and test data to address the label distribution shift. In contrast, our method multiplies a class-dimensional square matrix to the logits to address the class-wise confusion patterns generated by the mix of covariate shift and class distribution shift.
> >
> >
> > To further verify this argument, we conducted additional experiments comparing BNAdapt+DART with BNAdapt+LA for CIFAR-10C-LT. Note that the performance of [2] is limited by BNAdapt+LA, since [2] uses the LA-like logit adjustment but without the exact knowledge of class distribution shift. In TTA setup, if one can effectively treat the covariate shift and label distribution shift separately, it will be enough to apply the BNAdapt+LA. However, we can observe that our method outperforms BNAdapt+LA by 2\% for $\rho=10$ and 7\% for $\rho=100$. These results emphasize the necessity of simultaneously addressing both covariate shift and label distribution shift—as does our method—to effectively resolve the complex TTA scenario where both shifts coexist.
> >
> > |              | $\rho=10$ | $\rho=100$ |
> > |--------------|--------|---------|
> > | NoAdapt      | 71.28 | 71.13  |
> > | BNAdapt      | 79.01 | 66.90  |
> > | BNAdapt+DART | 84.53 | 83.34  |
> > | NoAdapt+LA   | 72.99 | 76.69  |
> > | BNAdapt+LA   | 82.01 | 76.88  |

---

### Official Review · Reviewer_7ZMU · 2023-11-01

**Soundness:** 3 good
**Presentation:** 3 good
**Contribution:** 3 good
**Rating:** 6
**Confidence:** 4

**Summary:**

This study examines the impact of label distribution shifts on the test-time adaptation (TTA) methods. The authors first demonstrates how class distribution shifts degrade the performance of BNAdapt, a method that updates Batch Normalization statistics during test time. Particularly, the research found that as class imbalance increased, BNAdapt's performance worsened compared to a NoAdapt approach, which maintains the original training without any modification. The study also highlights consistent confusion patterns among classes during label distribution shifts. To mitigate performance degradation caused by these shifts, the research introduces a distribution shift-aware module that refines classifier predictions by adjusting for detected class distribution changes during test time.  In various benchmarks, DART consistently outperformed existing TTA methods, especially as class imbalance ratios increased

**Strengths:**

- DART introduces a interesting approach to address class imbalance, presenting a significant improvement over existing TTA methods.
- The comprehensive benchmarks validate DART's superior performance across a range of imbalance ratios.
- The main modules of the proposed method is demonstrated through various ablation studies.

**Weaknesses:**

- Using labeled data at intermediate time for training is a recently introduced protocol. However, compared to traditional TTA methods that cannot access labeled data, this approach may not be entirely fair. Test time adaptation, where the model learns directly during the testing phase, might be a more desirable direction.

- The distribution-aware shift matrix for refinement has been frequently employed in handling label-noise datasets (Natarajan et al., 2013; Patrini et al., 2017; Zhu et al., 2021). Although the authors argue that TTA and these tasks differ, TTA essentially involves adding noise to the original data. Therefore, the nature of the problem between TTA and handling label-noise is fundamentally similar. The authors need to further elucidate the methodological distinctions between their proposed approach and existing methods (Natarajan et al., 2013; Patrini et al., 2017; Zhu et al., 2021). Additional considerations should also be clearly addressed.

**Questions:**

- It would be beneficial if the authors provided a more detailed explanation or key intuition behind the use of averaged pseudo labels as inputs in the distribution shift-aware module during intermediate time.
- I wonder whether T_test is updated on every batch during test time, or if it is constructed just once across the entire test dataset.

**Details Of Ethics Concerns:**

No concern

---

> ### Author Response · Authors · 2023-11-16
> **Response to Reviewer 7ZMU (1/2)**
>
> Thank you for the constructive feedback. We hope the responses to the questions and weaknesses below resolve your concerns.
>
> > (W1) Using labeled data at intermediate time for training is a recently introduced protocol. However, compared to traditional TTA methods that cannot access labeled data, this approach may not be entirely fair. Test time adaptation, where the model learns directly during the testing phase, might be a more desirable direction.
>
> We agree with the reviewer’s point that the intermediate-time training requires additional cost compared to the traditional TTA, but we’d like to emphasize that the additional cost in training our module for DART is very low and such a low additional cost can bring a significant performance gain. In particular, the cost in training DART is low since (1) it utilizes the training dataset, not a novel auxiliary dataset and (2) it requires a short runtime (about 2 hours for CIFAR-10C). With this low additional cost, our method exhibits consistent and significant performance gains when combined with the traditional TTA methods as shown in Table 2 of the manuscript. For example, it shows performance gains of 5.52% and 16.44% on CIFAR-10C-LT of $\rho=10$ and $100$, respectively.
>
> > (W2) The distribution-shift aware matrix for refinement has been frequently employed in handling label-noise datasets (Natarajan et al., 2013; Patrini et al., 2017; Zhu et al., 2021). Although the authors argue that TTA and these tasks differ, TTA essentially involves adding noise to the original data. Therefore, the nature of the problem between TTA and handling label noise is fundamentally similar. The authors need to further elucidate the methodological distinctions between their proposed approach and existing methods (Natarajan et al., 2013; Patrini et al., 2017; Zhu et al., 2021). Additional considerations should also be clearly addressed.
>
> We’d like to first clarify that the nature of the two problems, Learning with Label Noise (LLN) and Test-Time Adaptation (TTA), are different since the causes of the class-wise confusion are different. In LLN, it is assumed that there exists a noise transition matrix $T$, which determines the label-flipping probability of a sample from one class to other classes. For LLN, two main strategies have been widely used in estimating $T$: 1) using anchor points, which are defined as the training examples that belong to a particular class almost surely, and 2) using the clusterability of nearest-neighbors of a training example belonging to the same true label class. LLN uses the empirical pseudo label distribution of the anchor points or nearest-neighbors to estimate $T$.
> For TTA, on the other hand, the misclassification occurs not based on a fixed label-flipping pattern, but from the combination of covariate shift and label distribution shift. To adjust the pre-trained model against the covariate shifts, most TTA methods apply the BN adaptation, which updates the Batch Norm statistics using the test batches. However, when there exists a label distribution shift in addition to the covariate shift, since the updated BN statistics follow the test label distribution, it induces bias in the classier (by pulling the decision boundary closer to the head classes and pushing the boundary farther from the tail classes as described in Appendix C “motivating toy example”). Thus, the resulting class-wise confusion pattern depends not only on the class-wise relationship in the embedding space but also on the classifier bias originated from the label distribution shift and the updated BN statistics. Such a classifier bias has not been a problem for LLN, where we don’t need to modify the BN statistics of the classifier at the test time.
>
> Our proposed method, DART, focuses on this new class-wise confusion pattern and is built upon the idea that if the module experiences various batches with diverse class distributions before the test time, it can develop the ability to refine inaccurate predictions resulting from label distribution shifts. Based on this intuition, we train a distribution shift-aware module during the intermediate time, by exposing several batches with diverse class distributions using the training datasets. As described in Equation (1) of the manuscript, the module is trained using the labeled training dataset to output a square matrix of the class dimension for prediction refinement. In this process, the module takes the averaged pseudo-label distribution as an input to learn the class-wise confusion pattern of the BN-adapted classifier depending on the label distribution shift.

---

> ### Author Response · Authors · 2023-11-16
> **Response to Reviewer 7ZMU (2/2)**
>
> > Continued response to (W2)
>
> In Appendix D, we also reported the experimental results showing that the traditional LLN method is not effective in TTA for refining the predictions. In particular, we considered HOC (Zhu et al., 2021), one of the recent LLN methods that is known to successfully estimate the noise transition matrix under LLN scenarios. We found that HOC failed to estimate the transition matrix for CIFAR-10C-LT with the label distribution shift of $\rho=100$. HOC estimates the transition matrix by using the empirical pseudo label distribution of nearest neighbors of each example. However, as observed in Figure 6, the nearest neighbors in the embedding space already have the same pseudo labels/predictions for the BN-adapted classifier, which makes it impossible to estimate a correct $T$ depending on the label distribution shift.
>
> > (Q1) It would be beneficial if the authors provided a more detailed explanation or key intuition behind the use of averaged pseudo labels as inputs in the distribution shift-aware module during intermediate time.
>
> As explained in our response to (W2 of the reviewer), the distribution shift-aware module is trained to output a square matrix of the class dimension, only using the averaged pseudo labels, to fix the misclassification of the BN-adapted model originated from the class distribution shift. We designed the module to take the averaged pseudo label distribution as an input since this is the only available information at the test time. In the intermediate time, the module experiences several batches of diverse class distribution shifts and is optimized to generate a square matrix depending on each pseudo label distribution that can be multiplied to the outputs of BN-adapted classifier to minimize the cross-entropy loss between the true labels of training examples and the modified predictions. Through this process, the module learns the misclassification pattern of the BN-adapted classifier, originated from the label distribution shift, just from the observation of the average model predictions (pseudo labels).
> To verify this argument, in Appendix E.2, we reported the experimental results of the iterative version of DART, where the DART takes the iteratively refined pseudo labels as input. Specifically, for $i \in \mathbb{N}$, $ T_i = g_\phi(\mathbb{E}\_{x \in D_{\text{test}}}[\text{softmax}(f_\theta(x) \Pi_{j=0}^{i-1}T_{j})]) $, where $T_0$ is set to an identity matrix. Thus, $T_1$ is the output of our original DART. In Table 7 of the manuscript, we reported the pseudo label accuracy as the iteration increases. We can observe that (1) the accuracy of the refined prediction is maximized with $T_1$, and (2) the accuracy gradually decreases as the number of iterations increases. This indicates that our distribution-shift aware module indeed requires the average model predictions (pseudo labels) of BN-adapted classifier (not a more refined test label distribution) as it is trained to fix the misclassification pattern of the BN-adapted classifier, just using the noisy pseudo labels of the model itself.
>
> >(Q2) I wonder whether T_test is updated on every batch during test time, or if it is constructed just once across the entire test dataset.
>
> $T_\text{test}$ is constructed just once across the entire test dataset using the averaged pseudo-label distribution of the whole test data as described in the last paragraph of Section 3.
> As an additional remark, we also considered DART for online TTA, assuming that each test data sample is encountered only once during test time, in Section 4.3. Since obtaining $T_\text{test}$ for the whole test data is unavailable for online TTA, we modified DART to take the averaged pseudo-label distribution of the first test batch to output $T_\text{test}$, which is then used throughout the entire test time. We summarized the experimental results of this variant of DART for online TTA in Table 4 (last row) and Appendix F. The results indicate that this online variant of DART performs similarly to the original DART but with a slight decrease in performance.
>
> However, for the online label distribution shift setup on ImageNet-C, the class distributions within test batches are significantly different. Thus we computed the square matrix $T$ for each test batch as described in Appendix A.5.

---

### Official Review · Reviewer_JDRC · 2023-11-03

**Soundness:** 2 fair
**Presentation:** 3 good
**Contribution:** 2 fair
**Rating:** 6
**Confidence:** 5

**Summary:**

This paper focuses on the poor performance of TTA caused by class distribution shifts. To address this, the authors propose to refines the predictions by focusing on class-wise confusion patterns. Extensive experimental results on CIFAR, PACS, ImageNet benchmarks demonstrate the effectiveness of the proposed method.

**Strengths:**

1.	The authors conduct a empirical analysis that class distribution shifts would harm the performance.
2.	The authors propose distribution shift-aware module to alleviate the test-time class distribution shifts.

**Weaknesses:**

1.	In Notation part, why do the test data have labels? In Section 2, the authors calculate the cross-entropy using the test labels $ CE(softmax(f_{\theta}(x)T), y) $. Is it a pseudo label?
2.	The recent work SAR[Towards Stable Test-time Adaptation in Dynamic Wild World] also considers the case with class distribution shifts. What is the advantage of the proposed method over SAR? I found that the authors prepared the imbalanced ImageNet-C dataset following SAR. However, I failed to find the empirical comparisons between the proposed method and SAR.
3.	In my understanding, the authors seek to train a distribution shift-aware module to generate a matrix for prediction refining. In this sense, the data to train such a module should be class-imbalanced. However, as mentioned in the paper, the dataset $D_{int}$ seems to be class-balanced.
4.	What is the computational cost to train a distribution shift-aware module in the “intermediate time”? Does it take a long time?
5.	In the case without class distribution shifts, what is the performance of the proposed method compared with existing methods? Better or worse?

**Questions:**

If the authors could address my concern, I would raise my scoring.

---

> ### Author Response · Authors · 2023-11-16
> **Response to Reviewer JDRC (1/2)**
>
> Thank you for your interest in our method. We hope that the responses to the questions resolve your concerns.
>
> > (W1) In Notation part, why do the test data have labels? In Section 2, the authors calculate the cross-entropy using the test labels $CE[\text{softmax}(f_\theta(x)T, y)]$. Is it a pseudo label?
>
> In Section 2, we first presented **an oracle attempt** using the test labels to verify the effectiveness of multiplying a square matrix $T$ to the logits in refining the predictions generated by the BN-adapted classifier. Through this oracle attempt, we showed in Table 1 of the manuscript that the prediction refinement effectively enhances the performance of the BN-adapted classifier. However, during the test time, we don’t have access to test data labels. Thus, in this work, we proposed a novel test-time adaptation (TTA) method DART, which estimates the square matrix $T$ just with the pseudo labels of the test data using the distribution shift-aware module, as presented in Section 3.
>
> > (W2) The recent work SAR also considers the case with class distribution shifts. What is the advantage of the proposed method over SAR? I found that the authors prepared the imbalanced ImageNet-C dataset following SAR. However, I failed to find the empirical comparisons between the proposed method and SAR.
>
> DART can be integrated as a plug-in method with any TTA methods that rely on pseudo-labels obtained from the classifiers, including SAR, since DART focuses on effectively modifying the inaccurate predictions/pseudo-labels caused by test-time label distribution shifts. Thus, DART can also be used with SAR, and we summarize the experimental results on CIFAR-10C-LT in Table R4 and ImageNet-C-imbalance in Table R5. We can observe that the performances of SAR are worse/better than those of TENT on CIFAR-10C-LT/ImageNet-C-imbalance, respectively. However, DART consistently improves the performance of the SAR in a similar way as it improved the performances of other TTA methods, since DART improves the accuracy of the initial pseudo-labels used for SAR.
>
> **R4. Average accuracy (%) on CIFAR-10C-LT**
> |                     | $\rho=10$                            | $\rho=100$      |
> |---------------------|--------------------------------------|-----------------|
> | NoAdapt             | 71.28                                | 71.13           |
> | BNAdapt             | 79.01                                | 66.9            |
> | BNAdapt+DART (ours) | 84.53 (+5.52)                        | 83.34 (+16.44)  |
> | TENT                | 83.02                                | 70.49           |
> | TENT+DART (ours)    | 85.13 (+2.11)                        | 88.56 (+18.07)  |
> | SAR                 | 79.76                                | 67.3            |
> | SAR+DART (ours)     | 84.90 (+5.14)                        | 83.56 (+16.26)  |
>
> **R5. Average accuracy (%) on ImageNet-C-imbalance**
> |              | $\alpha=1000$ | $\alpha=2000$ | $\alpha=5000$  |
> |--------------|---------------|---------------|----------------|
> | NoAdapt      | 18.15         | 18.16         | 18.16          |
> | BNAdapt      | 19.85         | 14.11         | 8.48           |
> | BNAdapt+ours | 25.18 (+5.33) | 20.48 (+6.37) | 14.82 (+6.34)  |
> | TENT         | 22.49         | 13.52         | 6.61           |
> | TENT+ours    | 26.18 (+3.69) | 18.51 (+4.99) | 11.17 (+4.56)  |
> | SAR          | 26.46         | 17.36         | 9.09           |
> | SAR+ours     | 32.49 (+6.03) | 23.38 (+6.02) | 12.9 (+3.81)   |
>
> > (W3) In my understanding, the authors seek to train a distribution shift-aware module to generate a matrix for prediction refining. In this sense, the data to train such a module should be class-imbalanced. However, as mentioned in the paper, the dataset D_int seems to be class-balanced.
>
> Our method exposes various batches of diverse class distributions to the distribution shift-aware module by sampling the balanced $D_\text{int}$ **using the Dirichlet sampling** as illustrated in Figure 3 of the manuscript. We make $D_\text{int}$ class-balanced to avoid unintentional bias when sampling through the **Dirichlet distribution**. For example, if the intermediate dataset of two classes (1,2) has class distribution [0.9, 0.1], the sampled batches from the dataset are likely to have many more samples from class 1 even when Dirichlet sampling is applied.

---

> ### Author Response · Authors · 2023-11-16
> **Response to Reviewer JDRC (2/2)**
>
> > (W4) What is the computational cost to train a distribution shift-aware module in the “intermediate time”? Does it take a long time?
>
> As reported in Appendix A.6, it takes only about 2 hours to train the distribution shift-aware module for CIFAR-10C. Since the pre-trained classifier weights are frozen and the distribution shift-aware module has a simple structure (2-layer MLP), training the module takes a much shorter time compared to the pre-training step. Additionally, the computational overhead is not significant since we train the module only once for each pre-trained classifier regardless of the test corruption types or label-distribution shifts as described in Section 3 “training of $g_\phi$”.
>
> > (W5) In the case without class distribution shifts, what is the performance of the proposed method compared with existing methods? Better or worse?
>
> Our method refines the predictions of the classifier by multiplying a square matrix $T$ to the logits, and this process adjusts the incorrect predictions mainly originated from the class distribution shift. Thus, without the class distribution shift, multiplying $T$ does not have a big impact in adjusting the classifier predictions. For the balanced CIFAR-10C ($\rho=1$), where there is no label distribution shift in test time, we report the experimental results of DART in Appendix E.5. We observe that DART-applied TTA methods show comparable or slightly worse performance than the original TTA methods. This is also attributed to the limited gain even with the oracle method (using true test labels to find the oracle square matrix $T$ that will be multiplied to the logits) as shown in Table 1 ($\rho=1$). Specifically, the oracle achieved only a marginal performance gain of 0.3% on average even with the labels of test data on balanced CIFAR-10C. Therefore, DART, which uses the same prediction modification scheme, can only achieve limited gains even when generating square matrices similar to the one with the oracle. We’d like to point out that similar trends of limited gain for balanced datasets were also reported for ODS (Zhou et al., ICML’23), another attempt to alleviate test-time class distribution shifts.

---

> > ### Comment · Reviewer_JDRC · 2023-11-21
> > **More Questions**
> >
> > Thanks for the authors' responses. The responses have addressed most of my concerns.
> >
> > But I still have some questions about the third answer.
> >
> > The authors state "if the intermediate dataset of two classes (1,2) has class distribution [0.9, 0.1], the sampled batches from the dataset are likely to have many more samples from class 1 even when Dirichlet sampling is applied." I understand there would be more samples from class 1 in this case, but does it have a negative influence while training the distribution shift-aware module? In the inference stage, the test sample may be very imbalanced like this. So using such imbalanced data to train the module is appropriate in my understanding.

---

> > > ### Author Response · Authors · 2023-11-21
> > > **Response to the additional question**
> > >
> > > We thank the reviewer for the additional question.
> > >
> > > We'd like to highlight that we consider a more general situation where the test label distribution can be of any form, even including the imbalanced distribution of the reverse order from that of the training dataset (or the intermediate dataset if it is imbalanced). Thus, during the intermediate time, it is better to expose batches of as diverse class distributions as possible to the distribution shift-aware module for making it to learn the ability to adjust the output depending on the test label distribution. In the example we mentioned, where the intermediate dataset of two classes (1,2) have class distribution of [0.9, 0.1], if we know in advance that the test samples will also follow the similar class imbalance (having more class-1 samples), it would be beneficial to use such an imbalanced intermediate dataset as the reviewer pointed out. However, for the reverse case (encountering more test samples from class 2), using such an imbalanced intermediate dataset may prevent the module from experiencing the reversely-ordered class imbalance. Thus, we use the balanced intermediate dataset, and try to generate batches of as diverse class distributions as possible, including imbalanced distributions towards each of the classes, using the Dirichlet sampling. As illustrated in the probability simplex in Fig. 3, this sampling indeed covers diverse imbalanced distributions, making our module to experience diverse distributions during the intermediate time.

---

> > > > ### Comment · Reviewer_JDRC · 2023-11-22
> > > > **Thanks for the reponses**
> > > >
> > > > The explanations provided in this version are exceptionally clear and comprehensive. I hope the authors can incorporate these explanations into the revised manuscript for further clarity and completeness.
> > > >
> > > > All of my concerns have been effectively addressed, leading me to adjust my evaluation to a 'weak accept'.

---

### Author Response · Authors · 2023-11-17
**The revised paper is uploaded**

We have uploaded our revised paper with the following modifications:

* (Appendix B.3) We summarize an additional related work, loss correction in learning with label noise (LLN), and describe the distinction between our proposed method DART.
* (Appendix D) We add more details on analyzing experimental results under the test-time distribution shift of HOC (Zhu et al., 2021), one of the recent LLN methods that is known to successfully estimate the noise transition matrix in LLN setting.
* (Appendix E.6) We summarize the sensitivity analysis on the structure of the distribution shift-aware module and the test batch size.
* (Appendix E.7) We provide a discussion on the combination of DART and SAR, and summarize the performance when SAR and DART are integrated on CIFAR-10C-LT and ImageNet-C-imbalance.
* (Section 4.2 & Appendix F) We update Table 21-23, which reports the experimental results on ImageNet-C-imbalance by adding experimental results of SAR and DART-applied SAR. We correct the numbers of SAR+ours in Table 3 since there is a minor issue in the code. We apologize for the inconvenience.

---

### Author Response · Authors · 2023-11-21
**A gentle reminder to Reviewers**

Thank you for your time and efforts in reviewing our work.

We kindly remind you to check the author response if you have not already. We would appreciate it if you could let us know whether our response successfully addresses your concerns and questions. A statement would be of great help to us.

Thank you.

Authors.

---

### Meta-Review · Area_Chair_Zq4G · 2023-12-04

**Metareview:**

This paper proposes a test-time adaptation approach after the training phase has ended that incorporates confusion patterns into the test-time training. I think this is a clever idea, especially for label shift problems (although the paper also presents results for label + covariate shift problems). The main concern of the reviewers was the efficacy of the method for long-tail/imbalanced distributions. There, the improvement appeared to be marginal. I have personally read the paper, and I would also have liked it to have a clearer explanation of the label vs covariate shift benefits. After all, similar approaches have shown to be successful on label shift problems already (without covariate shifts, see "RLSBench Domain adaptation under relaxed label shift"). Overall the paper is well written and tells a compelling story, I would encourage the authors to resubmit, clarifying the role on the type of shift in the method's performance.

**Justification For Why Not Higher Score:**

In more detail, I would like a discussion of the method on long-tailed vs uniform data sets exhibiting label vs covariate shift vs both. Right now, it is not clear where the method will truly shine. From the current experiments, it looks like it mostly helps with label shift, where other similar approaches also exist.

**Justification For Why Not Lower Score:**

NA

---

### Decision · Program_Chairs · 2024-01-16

Reject